# An Agentic Framework with LLMs for Solving Complex Vehicle Routing Problems

**Ni Zhang[1], Zhiguang Cao[1,\*], Jianan Zhou[2], Cong Zhang[2], Yew-Soon Ong[2]**

[1]School of Computing and Information Systems, Singapore Management University, Singapore

[2] College of Computing and Data Science, Nanyang Technological University, Singapore

`ni.zhang.2025@phdcs.smu.edu.sg, zgcao@smu.edu.sg`
`jianan004@e.ntu.edu.sg, cong.zhang92@gmail.com, asysong@ntu.edu.sg`

## Abstract

Complex vehicle routing problems (VRPs) remain a fundamental challenge, demanding substantial expert effort for intent interpretation and algorithm design. While large language models (LLMs) offer a promising path toward automation, current approaches still rely on external intervention, which restrict autonomy and often lead to execution errors and low solution feasibility. To address these challenges, we propose an Agentic Framework with LLMs (AFL) for solving complex vehicle routing problems, achieving *full automation* from problem instance to solution. AFL directly extracts knowledge from raw inputs and enables *self-contained* code generation without handcrafted modules or external solvers. To improve trustworthiness, AFL decomposes the overall pipeline into three manageable subtasks and employs four specialized agents whose coordinated interactions enforce cross-functional consistency and logical soundness. Extensive experiments on 60 complex VRPs, ranging from standard benchmarks to practical variants, validate the effectiveness and generality of our framework, showing comparable performance against meticulously designed algorithms. Notably, it substantially outperforms existing LLM-based baselines in both code reliability and solution feasibility, achieving rates close to 100% on the evaluated benchmarks.

## 1 Introduction

Vehicle routing problems (VRPs) are fundamental to industrial and commercial applications such as logistics (Bochtis & Sørensen, 2010; Konstantakopoulos et al., 2022) and transportation (Cattaruzza et al., 2017; Zhang et al., 2022), yet they remain challenging to solve due to their diverse variants with intricate real-world constraints. Traditional approaches (Furnon & Perron; Helsgaun, 2017; Vidal, 2022; Wouda et al., 2024) often require substantial expert effort, either to translate problem statements into mathematical formulations or to design specialized algorithms. Although recent neural solvers (Kool et al., 2018; Kwon et al., 2020) alleviate the dependence on domain knowledge, they still require a certain degree of manual adaptation to address more complex VRPs.

More recently, large language models (LLMs) (Zhao et al., 2023), with their strong natural language understanding and code generation capabilities (Ma et al., 2026), may offer a promising avenue for automation, reducing reliance on manual effort and enabling flexible solver development across diverse VRP variants. Some early attempts (Yang et al., 2024; Liu et al., 2024b) directly prompt LLMs to generate solutions but fall short in terms of solution optimality and feasibility. Other approaches (Romera-Paredes et al., 2024) explore the use of LLMs to generate programming code as a proxy for addressing challenges in VRP optimization, which can be broadly categorized into two directions. The first direction centers on *evolving basic heuristics* tailored for conventional VRPs, with representative examples including EoH (Liu et al., 2024a) and ReEvo (Ye et al., 2024). In contrast, the second direction emphasizes *developing general frameworks* capable of handling diverse VRP variants, making it more practical and application-oriented.

Early research efforts have started to address this challenging second direction. Their workflow generally comprises two phases: *framework-design*, in which the architecture and generation strategy

---
*corresponding author

Table 1: Comparison of representative LLM-based approaches for VRPs.

| | ARS (Li et al., 2025a) | DRoC (Jiang et al., 2025b) | SGE (Iklassov et al., 2024) | **AFL** (This Work) |
|---|---|---|---|---|
| Complex VRPs | ✓ | ✓ | ✗ | ✓ |
| Self-Containment§ | ✗ | ✗ | ✓ | ✓ |
| Full Automation† | ✗ | ✗ | ✗ | ✓ |
| High Trustworthiness* | ✗ | ✗ | ✗ | ✓ |

§ *LLMs produce complete code without relying on handcrafted modules or external solvers during framework-design.*
† *The entire workflow proceeds from raw input to final solution without human intervention during framework-execution.*
* *Achieving high code reliability and solution feasibility (e.g., $\geq 95\%$).*

are specified (see Section 3), and *framework-execution*, in which the resulting framework is deployed to solve diverse problem instances. ARS (Li et al., 2025a) constructs constraint-checking functions by retrieving and adapting templates from a predefined constraint library, while DRoC (Jiang et al., 2025b) employs a retrieval-augmented generation (RAG) strategy to produce code that invokes OR-Tools (Furnon & Perron) for problem solving. Although both ARS and DRoC can handle complex VRPs, these module-level generation methods are not self-contained, depending on handcrafted code modules or external solvers during framework-design, and not fully automated, as they still require human involvement to extract instance-specific information during framework-execution. This dependence may introduce misalignment between LLM-generated code and external systems, which can result in execution errors and reduced solution feasibility. In contrast, SGE (Iklassov et al., 2024) achieves self-containment but is limited to relatively simple problems like the Traveling Salesman Problem (TSP), as it lacks effective mechanisms for handling complex constraints and fails to provide full automation or reliable code and solution validity. In this paper, as summarized in Table 1, we address these limitations by proposing a general framework of collaborative LLM-empowered agents that can tackle complex VRPs with self-containment, full automation, and high trustworthiness in both code and solutions.

We introduce an **A**gentic **F**ramework with **L**LMs (AFL) that solves complex VRPs end-to-end, from problem instance to solution. Specifically, it derives domain knowledge directly from instance inputs and leverages this knowledge to guide code generation. To enhance the feasibility and reliability of the generated code and the resulting VRP solutions under complex constraints, the pipeline is decomposed into three tractable subtasks: *problem description*, *code generation*, and *solution derivation*, each handled by multiple LLM agents tailored to their tasks. In total, we design four specialized agents, including *generation agent*, *judgment agent*, *revision agent*, and *error analysis agent*, collaborating to ensure cross-functional consistency, logical soundness, and constraint satisfaction. The overview of AFL is presented in Fig. 1. Our main contributions are summarized as follows.

1) Conceptually, we position LLMs as knowledgeable developers of self-contained frameworks for solving complex VRPs, achieving full automation from problem instance to solution without reliance on handcrafted modules or external solvers.

2) Methodologically, we propose AFL, an agentic LLM framework that decomposes the inherently intractable pipeline into three manageable subtasks and employs four specialized agents to collaboratively enhance trustworthiness in both code and solutions.

3) Experimentally, we evaluate AFL on 60 VRPs, comprising 48 representative VRPs from the literature, 8 complex electric VRPs from practical scenarios, and 4 classical VRPs in broader settings. Extensive results demonstrate the effectiveness and generality of our framework, showing competitive performance against carefully tailored algorithms while delivering superior code reliability and solution feasibility compared to existing LLM-based approaches.

## 2 PRELIMINARIES

The VRP is a fundamental combinatorial optimization task. It seeks a set of minimum-cost routes that allow a fleet of vehicles to serve geographically distributed customers while satisfying practical constraints such as vehicle capacity, route length, or customer time windows. Classic variants include the Capacitated VRP (CVRP), where each vehicle has a fixed capacity limit; the VRP with Time Windows (VRPTW), where every customer must be served within a specific time interval; and

Table 2: Constraint descriptions and corresponding VRPLib-format fields.

| Constraint | VRPLib Field | Description |
|---|---|---|
| Capacity (C) | CAPACITY DEMAND_SECTION | Each vehicle has a maximum load capacity, and each customer is associated with a demand that must be satisfied without exceeding this capacity. |
| Duration Limit (L) | DISTANCE_LIMIT | Each vehicle route is constrained by a maximum travel distance, and the total distance of any route must not exceed this limit. |
| Time Windows (TW) | TIME_WINDOW_SECTION SERVICE_TIME_SECTION | Each customer must be served within a specified time interval, and service times must be included in the schedule to maintain feasibility. |
| Open Route (O) | DEPOT_SECTION | Vehicles may not be required to return to the depot after serving their assigned customers, relaxing the standard closed-route assumption. |
| Electric Vehicle (E) | FUEL_CAPACITY FUEL_CONSUMPTION_RATE REFUEL_RATE STATION_SECTION | Electric vehicles are constrained by limited battery capacity; they consume energy during travel and refuel at recharging stations. |
| Multi Depot (MD) | DEPOT_SECTION | Multiple depots are defined, allowing vehicles to start and/or end at different depots, enabling flexible resource allocation across regions. |
| Backhaul (B) | DEMAND_SECTION | Each route includes both deliveries (linehauls) and pickups (backhauls), where pickups occur after all deliveries. |
| Mixture Backhaul (MB) | DEMAND_SECTION | Linehaul and backhaul customers appear in mixed order within the same route. |

the Electric VRP (EVRP), which incorporates battery capacity and recharging requirements. These formulations capture diverse real-world delivery, ride-sharing, and service-dispatch applications. A detailed introduction to each variant considered in this paper is provided in Appendix B.

To represent benchmark instances in a consistent way, we adopt the VRPLIB format (Uchoa et al., 2017), a plain-text specification similar to TSPLIB (Reinelt, 1991). A VRPLIB file begins with general information such as the instance name and an optional comment, followed by key sections specifying the problem type, edge weight type, dimension, and the coordinates of each location. Additional sections may cover parameters such as vehicle capacity, customer demands (with positive values for linehaul and negative for backhaul), distance limits, depot IDs, time windows, service times, and energy-related data for electric vehicles (e.g., fuel capacity, consumption rate, refueling rate, and charging station locations).

The mapping between constraints and their corresponding VRPLIB fields is summarized in Table 2, with further details on each field provided in Table 8 in the Appendix. Our AFL directly takes VRPLIB-format instances as input. We also evaluate AFL on JSON and CSV formats to demonstrate its robustness to different data representations. The results are reported in Section C.11 and Table 17.

## 3 METHODOLOGY

In this section, we introduce AFL, an agentic LLM framework for solving complex VRPs by structuring the pipeline into three subtasks: *problem description*, *code generation*, and *solution derivation*. Within these subtasks, specialized agents, including the *generation agent (GA)*, *judgment agent (JA)*, *revision agent (RA)*, and *error analysis agent (EAA)*, collaborate to fulfill their respective roles, ultimately enhancing the trustworthiness of both the generated code and the derived solutions under the constraints of the given problem instance.

The overview of AFL is presented in Fig. 1. Specifically, given a VRP instance $\mathcal{G}$, the system first generates a problem description $\mathcal{D}(\mathcal{G})$ through the collaborative operation of the GA, JA, and RA. This problem description is then used to query the buffer which stores previously tested problem codes, to check whether relevant code has been previously stored. If such code exists, the workflow proceeds directly to the solution derivation stage. Otherwise, the GA progressively generates the required functions one by one, while the JA and RA iteratively evaluate and refine the code until it

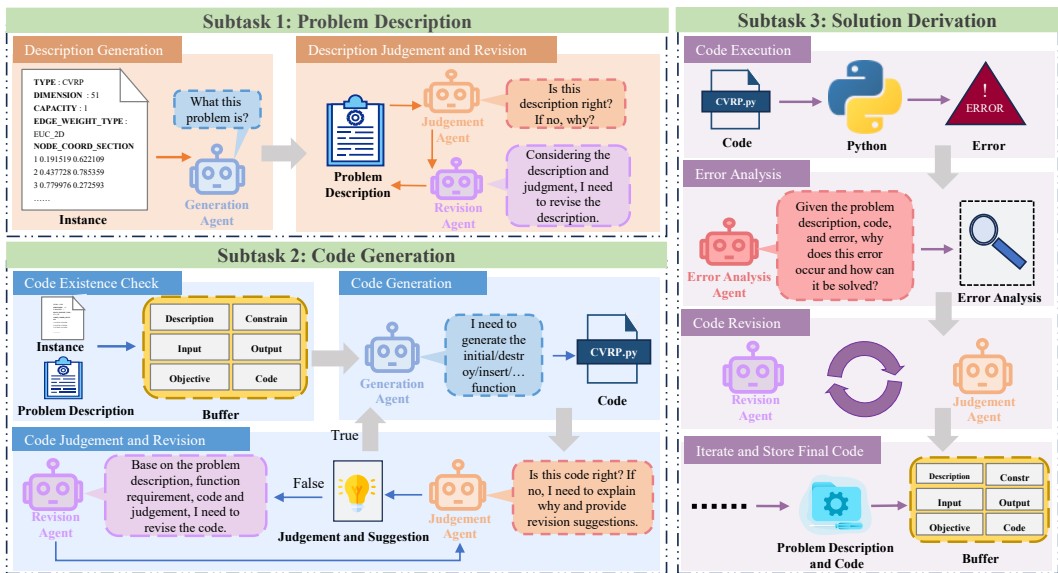

Figure 1: Overview of an agentic framework with LLMs for solving complex VRPs.

meets all requirements and constraints. Once a complete implementation is produced, it is executed to derive a solution. If execution errors occur, the EAA diagnoses their causes and provides explanations and suggestions, which the RA and JA use to revise the code. This iterative process continues until the code passes validation and produces a feasible solution. Finally, the corresponding problem description and code are stored in the buffer for future reuse. Examples of the agents' prompts and outputs for each subtask are provided in Appendix D. In the following, we present each specialized agent and pipeline stage in detail.

## 3.1 SPECIALIZED AGENT

**Generation Agents (GA)** are responsible for producing descriptions and code. In the problem description subtask, they generate a description $\mathcal{D}(\mathcal{G})$ for the input VRPLib-format instance $\mathcal{G}$. In the code generation subtask, they generate function code $\mathcal{C}(\mathcal{G}, \mathcal{D}(\mathcal{G}), \mathcal{P}(f))$ in an end-to-end manner, guided by the instance, the generated description, and the specific prompts $\mathcal{P}(f)$ associated with function $f$. The resulting description and code are then forwarded to the JA for evaluation.

**Judgment Agents (JA)** evaluate the validity of the generated description and code. In the problem description subtask, they verify whether $\mathcal{D}(\mathcal{G})$ aligns with the instance context. In the code generation and solution derivation subtasks, they further assess whether the generated or revised code satisfies the prompt requirements and is free from syntactic and logical errors. If the judgment is positive, the description or code is accepted and the process advances to the next step. Otherwise, the JA provides explanations of identified issues along with suggestions for resolution by the RA.

**Revision Agents (RA)** refine both the description and the code. Description revision is guided by the JA's feedback and the instance context, while code revision additionally leverages the previously generated description. After each revision, the updated description or code is returned to the JA for re-evaluation, and this process continues until a positive judgment is reached.

**Error Analysis Agents (EAA)** operate exclusively in the solution derivation subtask, where they analyze the causes of errors during code execution and provide suggestions for resolving them. The analysis is then passed to the RA for code revision.

## 3.2 SUBTASK 1: PROBLEM DESCRIPTION

**Description Generation.** Given a VRPLib-format instance $\mathcal{G}$, our framework automatically extracts domain knowledge from the instance context without human intervention, offering a user-friendly interface for problem setup. The VRPLib format is a widely adopted benchmark specification for VRPs, defining essential elements such as the problem type, number of nodes, node coordinates,

depot ID, and various constraint-related parameters, as summarized in Table 2. Based on this, the GA generates the problem description for the given instance $\mathcal{D}(\mathcal{G}) = \{P, S, K, X, Y, Z\}$. A detailed example is provided in Appendix D.1. Here, we define the components of $\mathcal{D}(\mathcal{G})$ as follows:

1) $P$ specifies the *type of problem* (e.g., CVRP, VRPTW, ECVRPTW). It is inferred from the problem type and the constraint-related parameters defined in the instance context, and it determines the name of the generated code file (e.g., CVRP.py).

2) $S$ denotes the *textual description* of the instance's problem type. It is provided to the code generation subtask to inform the agents about the problem definition.

3) $K$ represents the set of *constraints* along with their explanations. These are derived from the constraint-related parameters and problem type specified in the instance context. In addition, $K$ includes visit and depot constraints, which are supplementary requirements automatically analyzed and inferred by the GA. Within the code generation subtask, $K$ guides the agents in embedding these constraints into function design, thereby enhancing the solution feasibility.

4) $X$ denotes the *required input* for solving the given instance. In the code generation subtask, it specifies the information that must be read from the instance and enforces consistency by requiring input variable names to match those in $X$, thereby reducing potential errors. For example, in CVRP, $X$ includes node coordinates, depot ID, customer demands, and vehicle capacity.

5) $Y$ refers to the *expected output*. For instance, in CVRP, the solver should produce a set of vehicle routes, each starting and ending at the depot, visiting every customer exactly once, ensuring that no vehicle route exceeds capacity and that all demands are satisfied. Moreover, the returned route should represent the best feasible solution among the candidates.

6) $Z$ represents the *objective function*, such as minimizing the total travel distance, which is further used in constructing the cost function code.

**Description Judgment and Revision.** After the GA generates the above problem description $\mathcal{D}(\mathcal{G})$, the JA evaluates its correctness. The evaluation checks: (i) whether any component of $\mathcal{D}(\mathcal{G})$ conflicts with the instance, (ii) whether the components are internally consistent, and (iii) whether the input definition $X$ is properly specified in the instance context. If a conflict is detected, the instance context serves as the reference standard. If no issues are found, the output is set to TRUE, the problem description subtask terminates, and the process advances to the next code generation subtask. Otherwise, the output is set to FALSE, accompanied by explanations of the negative judgment and suggestions for the RA to make correction. The RA then revises $\mathcal{D}(\mathcal{G})$ based on the JA's feedback and the instance context. The revised description is returned to the JA for re-evaluation, and this iterative process continues until the JA confirms that $\mathcal{D}(\mathcal{G})$ is correct. This iterative procedure improves the accuracy of $\mathcal{D}(\mathcal{G})$, as demonstrated by the ablation study in Section 4.5. The problem description subtask provides the essential information required for code generation and enforces unified naming conventions and constraints, which must remain consistent throughout the entire pipeline.

### 3.3 SUBTASK 2: CODE GENERATION

We adopt a unified destroy-insert heuristic for solving VRPs, as it offers greater flexibility than others and can handle complex, practical problem variants. The code generation subtask consists of interdependent functions: $read\_vrp$, $distance$, $cost$, $initial$, $destroy$, $insert$, $validate$, and $main$, which together form a complete VRP solver. Generating the full solver code, however, is challenging, as it requires maintaining consistency across multiple functions while satisfying all requirements. To address this, the GA produces the functions sequentially, with each building upon the previously generated code to ensure correctness and reduce the burden on the LLM. In addition, the JA and RA iteratively refine the code by correcting unmet requirements, syntactic errors, and logical inconsistencies after each function is generated. We describe each step in detail below.

**Code Generation.** The code structure of the problem-solving workflow is shown in Fig. 2. We specify the role of each function to provide a structured foundation for guiding the GA in generating the corresponding code. Note that these functions are executed only in the solution derivation subtask and are fixed by the EAA if any runtime errors occur. First, $read\_vrp$ parses a VRPLib-format instance file into a structured dictionary containing all required fields specified by the input $X \in \mathcal{D}(\mathcal{G})$, ensuring that each variable in $X$ is accurately extracted from the instance context $\mathcal{G}$. Next, $distance$ computes the distance matrix from the node coordinates. $initial$ constructs

a solution using a greedy strategy that respects constraints in $K$. The feasibility of the solution is verified by $validate$. $cost$ evaluates the objective value of a given solution according to the objective function $Z$. To enable iterative improvement, $destroy$ removes a subset of customers from the current solution, following the strategy described in Appendix C.2 and Algorithm 1.

Then, the $insert$ function reinserts the removed customers into feasible positions while minimizing the additional cost. If no feasible insertion exists, a new vehicle is assigned to serve these customers in compliance with constraints in $K$. At each improvement step, the feasibility of the resulting solution is verified by $validate$ to ensure that every constraint in $K$ is satisfied. In the event of a constraint violation, the function must raise an error, thereby aiding the EAA in debugging. Finally, $main$ orchestrates the entire workflow, encompassing initialization, iterative improvement, and overall solution management, as illustrated in Fig. 2. In the initialization phase, an initial feasible solution is generated, while in the improvement phase ($T$ steps in total), the solution is iteratively refined through destruction, insertion, validation, and cost evaluation, with new solutions accepted according to the simulated annealing criterion (see Appendix C.3).

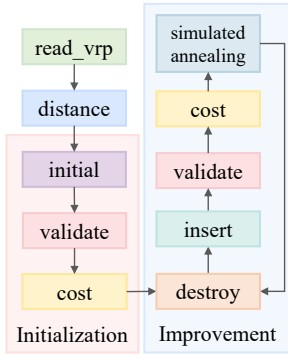

Figure 2: Code structure.

**Code Judgment and Revision.** For each function generated by the GA, the JA assesses the correctness of the code produced thus far, checking compliance with the requirements and detecting any syntactic or logical errors. If issues are identified, the RA revises the code based on the JA's feedback and the instance context. The revised code is then returned to the $JA$ for re-evaluation, and this process is repeated until the $JA$ delivers a positive judgment. By validating and correcting each code segment before generating the next function, this mechanism reduces the burden on subsequent code generation and revision, improves efficiency, and enhances the reliability of the final solver implementation. Moreover, constraint considerations are enforced throughout the code generation process. The generated code is repeatedly checked to ensure that all constraints in $K$ are properly incorporated. This iterative enforcement helps the final solver produce solutions feasible with respect to the instance constraints.

### 3.4 SUBTASK 3: SOLUTION DERIVATION

The functions produced in the code generation subtask is not always executable, as constructing a full VRP solver is highly complex. Bugs may arise for several reasons: some stem from syntactic errors, others from logical flaws, and still others from unmet requirements, such as failing to incorporate certain constraints. Although we have designed strategies such as enforcing constraint considerations during code generation, guaranteeing the correctness of LLM-generated code remains non-trivial. To address this challenge and enhance the trustworthiness of the generated VRP solver, we leverage an EAA to identify the cause of errors and provide explanations along with suggestions for correction. Similar to the code generation subtask, the RA then modifies the code based on this feedback, after which the JA evaluates the revision. If the code remains unsatisfactory, the RA further adjusts it according to the JA's feedback, and this process repeats until the JA delivers a positive judgment. The revised code is then re-executed to obtain a feasible solution. Eventually, the model stores the problem description $\mathcal{D}(\mathcal{G})$ together with the corresponding generated code in the buffer. If the same problem is encountered again, the framework can directly reuse the stored code, thereby improving efficiency and avoiding redundant computation.

## 4 EXPERIMENT

We first evaluate AFL against traditional and neural approaches on 48 standard benchmarks incorporating common constraints such as capacity (C), duration limit (L), time window (TW), open route (O), multi depot (MD), backhaul (B), and mixture backhaul (MB), which are widely used to assess traditional algorithms. We then extend the evaluation to 8 practical electric (E) VRPs, which remain challenging for traditional solvers. Next, we benchmark AFL against LLM-based approaches, assessing code reliability, solution feasibility, and overall performance. We also conduct ablation studies to examine the effectiveness of our agentic design. In addition, we report comparisons under

Table 3: Comparison results on standard benchmarks. More results are shown in Table 11.

| | | n=50 | | | n=100 | | | | n=50 | | | n=100 | | |
|---|---|---|---|---|---|---|---|---|---|---|---|---|---|---|
| | | Obj. | Gap (%) | Time (m) | Obj. | Gap (%) | Time (m) | | Obj. | Gap (%) | Time (m) | Obj. | Gap (%) | Time (m) |
| HGS-PyVRP | CVRP | 10.37 | – | 10.40 | 15.62 | – | 20.80 | CVRPL | 10.59 | – | 10.40 | 15.77 | – | 20.80 |
| OR-Tools | | 10.57 | 1.91 | 10.40 | 16.28 | 4.18 | 20.80 | | 10.83 | 2.34 | 10.40 | 16.47 | 5.30 | 20.80 |
| RF-POMO | | 10.51 | 1.31 | 0.03 | 15.91 | 1.83 | 0.12 | | 10.75 | 1.52 | 0.02 | 16.11 | 2.17 | 0.10 |
| AFL ($T$=500) | | 10.89 | 5.01 | 0.18 | 16.66 | 6.66 | 0.32 | | 11.35 | 7.18 | 0.33 | 17.36 | 10.08 | 1.03 |
| AFL ($T$=2000) | | 10.70 | 3.18 | 0.46 | 16.22 | 3.84 | 1.02 | | 11.25 | 6.23 | 1.13 | 17.03 | 7.99 | 7.35 |
| AFL ($T$=10000) | | **10.59** | **2.12** | 2.10 | **15.99** | **2.38** | 4.38 | | 11.18 | 5.57 | 6.93 | 16.84 | 6.79 | 25.48 |
| HGS-PyVRP | CVRPTW | 16.03 | – | 10.40 | 25.42 | – | 20.80 | OCVRP | 6.51 | – | 10.40 | 9.73 | – | 20.80 |
| OR-Tools | | 16.08 | 0.35 | 10.40 | 25.81 | 1.51 | 20.80 | | 6.55 | 0.69 | 10.40 | 10.00 | 2.73 | 20.80 |
| RF-POMO | | 16.37 | 2.09 | 0.02 | 26.34 | 3.58 | 0.12 | | 6.70 | 2.90 | 0.02 | 10.18 | 4.66 | 0.10 |
| AFL ($T$=500) | | **16.49** | **2.87** | 0.59 | 26.60 | 4.64 | 2.15 | | 6.79 | 4.30 | 0.31 | 10.37 | 6.58 | 0.4 |
| AFL ($T$=2000) | | **16.28** | **1.56** | 2.26 | **26.02** | **2.36** | 8.92 | | **6.69** | **2.76** | 0.66 | 10.11 | 3.91 | 1.16 |
| AFL ($T$=10000) | | **16.19** | **0.99** | 9.31 | 25.79 | 1.46 | 38.45 | | **6.64** | **2.00** | 2.20 | **9.99** | **2.67** | 5.52 |
| HGS-PyVRP | CVRPLTW | 16.36 | – | 10.40 | 25.76 | – | 20.80 | OCVRPL | 6.51 | – | 10.40 | 9.72 | – | 20.80 |
| OR-Tools | | 16.44 | 0.50 | 10.40 | 26.26 | 1.90 | 20.80 | | 6.55 | 0.67 | 10.40 | 10.00 | 2.79 | 20.80 |
| RF-POMO | | 16.75 | 2.38 | 0.02 | 26.78 | 3.95 | 0.12 | | 6.70 | 2.95 | 0.02 | 10.18 | 4.66 | 0.1 |
| AFL ($T$=500) | | 17.07 | 4.34 | 0.61 | 27.60 | 7.14 | 1.95 | | 6.81 | 4.61 | 0.43 | 10.37 | 6.68 | 1.03 |
| AFL ($T$=2000) | | **16.78** | **2.57** | 1.95 | 26.96 | 4.66 | 7.21 | | **6.71** | **3.07** | 1.21 | 10.15 | 4.42 | 3.68 |
| AFL ($T$=10000) | | **16.61** | **1.53** | 8.90 | 26.56 | 3.11 | 35.33 | | **6.64** | **1.99** | 5.65 | **9.99** | **2.78** | 16.27 |
| HGS-PyVRP | OCVRPTW | 10.51 | – | 10.40 | 16.93 | – | 20.80 | OCVRPLTW | 10.51 | – | 10.40 | 16.93 | – | 20.80 |
| OR-Tools | | 10.52 | 0.08 | 10.40 | 17.03 | 0.58 | 20.80 | | 10.50 | 0.11 | 10.40 | 17.02 | 0.73 | 20.80 |
| RF-POMO | | 10.66 | 1.38 | 0.02 | 17.39 | 2.72 | 0.12 | | 10.66 | 1.38 | 0.02 | 17.39 | 2.73 | 0.12 |
| AFL ($T$=500) | | **10.64** | **1.24** | 0.67 | **17.36** | **2.54** | 0.75 | | **10.74** | **2.12** | 1.05 | 17.61 | 4.02 | 3.63 |
| AFL ($T$=2000) | | **10.57** | **0.57** | 2.86 | **17.14** | **1.24** | 10.63 | | **10.64** | **1.24** | 3.76 | **17.32** | **2.30** | 16.75 |
| AFL ($T$=10000) | | **10.55** | **0.38** | 11.15 | **17.04** | **0.65** | 49.05 | | **10.58** | **0.67** | 17.27 | **17.19** | **1.54** | 70.31 |
| HGS-PyVRP | CVRPB | 9.69 | – | 10.40 | 14.38 | – | 20.80 | CVRPBL | 10.19 | – | 10.40 | 14.78 | – | 20.80 |
| OR-Tools | | 9.80 | 1.16 | 10.40 | 14.93 | 3.85 | 20.80 | | 10.33 | 1.39 | 10.40 | 15.43 | 4.34 | 20.80 |
| RF-POMO | | 10.00 | 3.17 | 0.02 | 15.02 | 4.47 | 0.10 | | 10.59 | 3.94 | 0.02 | 15.63 | 5.70 | 0.10 |
| AFL ($T$=500) | | 10.24 | 5.68 | 2.62 | 15.48 | 7.65 | 5.56 | | 10.87 | 6.67 | 0.31 | 16.18 | 9.47 | 2.15 |
| AFL ($T$=2000) | | 10.09 | 4.13 | 6.51 | 15.07 | 4.80 | 14.47 | | 10.64 | 4.42 | 3.31 | 15.71 | 6.29 | 4.80 |
| AFL ($T$=10000) | | **9.95** | **2.27** | 31.54 | **14.74** | **2.50** | 70.59 | | 10.52 | 3.24 | 11.06 | 15.41 | 4.26 | 20.18 |
| HGS-PyVRP | CVRPBTW | 18.29 | – | 10.40 | 29.47 | – | 20.80 | OCVRPB | 6.90 | – | 10.40 | 10.34 | – | 20.80 |
| OR-Tools | | 18.37 | 0.38 | 10.40 | 29.95 | 1.60 | 20.80 | | 6.93 | 0.21 | 10.40 | 10.58 | 2.32 | 20.80 |
| RF-POMO | | 18.60 | 1.67 | 0.02 | 30.34 | 2.96 | 0.12 | | 7.09 | 2.69 | 0.02 | 10.84 | 4.82 | 0.12 |
| AFL ($T$=500) | | 19.06 | 4.21 | 0.99 | 31.50 | 6.89 | 3.75 | | 7.57 | 9.71 | 0.72 | 11.67 | 12.86 | 3.50 |
| AFL ($T$=2000) | | **18.75** | **2.52** | 1.76 | 30.63 | 3.94 | 17.20 | | 7.40 | 7.24 | 2.64 | 11.34 | 9.67 | 16.30 |
| AFL ($T$=10000) | | **18.61** | **1.75** | 14.88 | **30.17** | **2.38** | 70.69 | | 7.30 | 5.80 | 7.97 | 11.12 | 7.54 | 56.08 |
| HGS-PyVRP | CVRPBLTW | 18.36 | – | 10.40 | 29.03 | – | 20.80 | OCVRPBL | 6.90 | – | 10.40 | 10.34 | – | 20.80 |
| OR-Tools | | 18.42 | 0.33 | 10.40 | 29.83 | 2.77 | 20.80 | | 6.93 | 0.39 | 10.40 | 10.58 | 2.36 | 20.80 |
| RF-POMO | | 18.94 | 1.85 | 1.00 | 30.80 | 3.28 | 0.12 | | 7.09 | 2.69 | 1.00 | 10.84 | 4.83 | 0.12 |
| AFL ($T$=500) | | 19.09 | 3.98 | 0.85 | 31.31 | 7.85 | 3.02 | | 7.16 | 3.77 | 1.05 | 10.95 | 5.90 | 4.57 |
| AFL ($T$=2000) | | **18.87** | **2.78** | 2.53 | 30.64 | 5.55 | 5.75 | | **7.05** | **2.17** | 4.02 | 10.72 | 3.68 | 8.72 |
| AFL ($T$=10000) | | **18.78** | **2.29** | 20.19 | 30.33 | 4.48 | 69.12 | | **7.03** | **1.74** | 23.50 | **10.58** | **2.32** | 76.06 |
| HGS-PyVRP | OCVRPBTW | 11.67 | – | 10.40 | 19.16 | – | 20.80 | OCVRPBLTW | 11.67 | – | 10.40 | 19.16 | – | 20.80 |
| OR-Tools | | 11.68 | 0.11 | 10.40 | 19.30 | 0.76 | 20.80 | | 11.68 | 0.11 | 10.40 | 19.31 | 0.77 | 20.80 |
| RF-POMO | | 11.80 | 1.15 | 1.00 | 19.61 | 2.34 | 0.12 | | 11.81 | 1.16 | 1.00 | 19.61 | 2.34 | 0.13 |
| AFL ($T$=500) | | **11.80** | **1.11** | 0.68 | **19.67** | **2.66** | 1.95 | | **11.81** | **1.20** | 0.57 | **19.67** | **2.66** | 4.65 |
| AFL ($T$=2000) | | **11.73** | **0.51** | 2.26 | **19.41** | **1.30** | 15.71 | | **11.75** | **0.69** | 4.92 | **19.41** | **1.30** | 18.15 |
| AFL ($T$=10000) | | **11.71** | **0.34** | 26.32 | **19.29** | **0.68** | 83.60 | | **11.71** | **0.34** | 10.75 | **19.29** | **0.68** | 39.73 |

**Note:** The term *obj.* denotes the objective value, *gap* represents the relative gap with respect to HGS-PyVRP, and *time* indicates the total runtime. For all three metrics, lower values are better. Bold numbers highlight our cases where the gap from the SOTA is within 3%, which is considered acceptable given that our framework is fully automated and self-contained.

different prompt strategies in the main text. We further evaluate AFL's robustness across different LLM backbones, and these results are presented in Section C.10 and Table C.9 in the Appendix. Finally, we evaluate 4 additional open benchmarks, including TSP, ATSP, ACVRP, and SOP, to demonstrate the broad applicability of our framework. All experiments were conducted via the OpenAI API using GPT-4.1 (OpenAI, 2024) on a server equipped with an AMD EPYC 7702P CPU and 64 GB of RAM, without GPU acceleration. All results shown below are derived using the same heuristic for the same variant on a given instance. This setting is more practical for real-world applications and significantly reduces code-generation overhead. The code and dataset is available at https://github.com/ZHANG-NI/AFL.git

Table 4: Comparison results on practical benchmarks.

| | | Small Instances | | | Large Instances | | | | Small Instances | | | Large Instances | | |
|---|---|---|---|---|---|---|---|---|---|---|---|---|---|---|
| | | Obj. | Gap (%) | Time (m) | Obj. | Gap (%) | Time (m) | | Obj. | Gap (%) | Time (m) | Obj. | Gap (%) | Time (m) |
| ACO | ECVRP | 270.68 | 0.00 | 0.23 | 1384.15 | 0.00 | 6.89 | ECVRPL | 341.74 | 0.00 | 0.24 | 952.07 | 0.00 | 7.92 |
| Greedy | | 309.63 | 14.39 | 0.05 | 1407.97 | 1.72 | 0.13 | | 714.34 | 109.03 | 0.06 | 7298.83 | 666.63 | 0.20 |
| AFL (T=500) | | **266.21** | **-1.65** | 0.18 | **1145.51** | **-17.24** | 0.26 | | **276.95** | **-18.96** | 0.18 | **885.15** | **-7.03** | 0.43 |
| AFL (T=2000) | | **264.21** | **-2.39** | 0.24 | **1100.11** | **-20.52** | 0.54 | | **276.94** | **-18.96** | 0.25 | **866.87** | **-8.95** | 1.22 |
| AFL (T=10000) | | **263.33** | **-2.72** | 0.51 | **1045.74** | **-24.45** | 2.19 | | **276.94** | **-18.96** | 0.53 | **861.05** | **-9.56** | 4.95 |
| ACO | ECVRPTW | 323.88 | 0.00 | 0.25 | 1129.67 | 0.00 | 7.00 | EOCVRP | 179.30 | 0.00 | 0.22 | 796.97 | 0.00 | 6.81 |
| Greedy | | 379.88 | 17.29 | 0.06 | 1868.64 | 65.41 | 0.22 | | 197.03 | 9.89 | 0.06 | 866.02 | 8.66 | 0.20 |
| AFL (T=500) | | **306.50** | **-5.37** | 0.18 | **1101.14** | **-2.53** | 0.40 | | **166.76** | **-6.99** | 0.19 | **632.73** | **-20.61** | 0.61 |
| AFL (T=2000) | | **300.95** | **-7.08** | 0.23 | **1071.22** | **-5.17** | 1.17 | | **166.02** | **-7.41** | 0.40 | **628.34** | **-21.16** | 1.90 |
| AFL (T=10000) | | **300.23** | **-7.30** | 0.51 | **1044.63** | **-7.53** | 4.52 | | **165.41** | **-7.75** | 1.20 | **623.36** | **-21.78** | 10.82 |
| ACO | ECVRPLTW | 419.85 | 0.00 | 0.44 | 2842.17 | 0.00 | 13.88 | EOCVRPL | 204.87 | 0.00 | 0.34 | 759.24 | 0.00 | 10.50 |
| Greedy | | 434.29 | 3.44 | 0.09 | 3039.87 | 6.96 | 0.34 | | 221.34 | 8.04 | 0.08 | 896.92 | 18.13 | 0.30 |
| AFL (T=500) | | **388.36** | **-7.50** | 0.21 | **2729.73** | **-3.96** | 0.50 | | **173.43** | **-15.35** | 0.21 | **726.34** | **-4.33** | 0.54 |
| AFL (T=2000) | | **386.84** | **-7.86** | 0.30 | **2612.83** | **-8.07** | 1.54 | | **172.90** | **-15.61** | 0.36 | **693.71** | **-8.63** | 1.50 |
| AFL (T=10000) | | **381.84** | **-9.05** | 0.87 | **2527.87** | **-11.06** | 7.07 | | **172.53** | **-15.79** | 1.12 | **669.43** | **-11.83** | 9.29 |
| ACO | EOCVRPTW | 223.91 | 0.00 | 0.39 | 1139.12 | 0.00 | 11.98 | EOCVRPLTW | 205.77 | 0.00 | 27.55 | 1184.48 | 0.00 | 14.12 |
| Greedy | | 241.22 | 7.73 | 0.08 | 1219.73 | 7.08 | 0.33 | | 222.89 | 8.32 | 0.10 | 1230.85 | 3.91 | 0.37 |
| AFL (T=500) | | **183.26** | **-18.15** | 0.19 | **856.13** | **-24.84** | 0.43 | | **181.74** | **-11.68** | 0.20 | **785.53** | **-33.68** | 0.48 |
| AFL (T=2000) | | **183.11** | **-18.22** | 0.24 | **835.03** | **-26.70** | 1.20 | | **181.43** | **-11.83** | 0.26 | **777.02** | **-34.40** | 1.26 |
| AFL (T=10000) | | **182.88** | **-18.32** | 0.49 | **815.64** | **-28.40** | 4.75 | | **181.27** | **-11.91** | 0.53 | **775.03** | **-34.57** | 5.84 |

## 4.1 COMPARISON ON STANDARD BENCHMARK

We compare AFL with the traditional solvers HGS (implemented in PyVRP) (Vidal, 2022; Wouda et al., 2024) and OR-Tools (Furnon & Perron), as well as the neural solver RF-POMO (Berto et al., 2025b). The experimental settings and testing data follow Berto et al. (2025b), including 1,000 instances for each problem. Note that *our objective is not to surpass SOTA solvers on conventional VRPs, which reflect decades of expert effort, but to develop a fully automated and self-contained framework for tackling complex VRPs*. Therefore, in the comparison shown in Table 3, we regard a 3% relative gap with respect to SOTA solvers as an acceptable criterion. We report the results of AFL after 500, 2,000, and 10,000 iterations of solution improvement. The runtime of AFL is measured as the sum of the problem description and solution derivation phases. The code generation phase, analogous to the training phase of a neural solver, is excluded, since once the solver code is produced, it can be reused across instances without incurring repeated generation overhead, and the runtime analysis of code generation phase is shown in Section C.4 in Appendix. Results are partly shown in Table 3, with the rest provided in Table 11 in the Appendix due to space limitations.

AFL automatically generates a complete VRP solver without any manual intervention. As shown in Table 3, it achieves a relative gap within 3% of the SOTA HGS on most benchmark problems, demonstrating competitive performance. It is worth noting that the reported runtimes exhibit stochastic variation: more complex problems do not necessarily result in longer execution times. For example, the runtime on OCVRPL is shorter than that on CVRPL. This variability arises from the LLM-based code generation process, where the model may occasionally produce implementations (e.g., sorting) with higher algorithmic complexity, leading to longer runtimes.

## 4.2 COMPARISON ON PRACTICAL BENCHMARK

Traditional solvers like HGS (Vidal, 2022; Wouda et al., 2024) and OR-Tools (Furnon & Perron) are inherently constrained by their internal implementations and cannot be directly adapted to new problem settings without substantial modifications to the core codebase, whereas AFL can naturally accommodate practical VRPs. To demonstrate this, we conduct experiments on a widely used benchmark for ECVRPTW (Schneider et al., 2014), a representative variant of the electric vehicle routing problem (EVRP). Specifically, the dataset contains 36 small instances with 5, 10, and 15 customers, as well as 56 large instances with 100 customers. To further assess generality, we extend this benchmark to 7 additional EVRP variants, namely ECVRP, ECVRPL, EOVRPL, EOCVRP, EOCVRPTW, ECVRPLTW, and EOCVRPLTW, enabling a more comprehensive evaluation across

Table 5: RER and SR.

|      | RER ↓  | SR ↑   |
|------|--------|--------|
| SGE  | 94.1%  | 5.9%   |
| DRoC | 82.4%  | 17.6%  |
| AFL  | **0%** | **100%** |

Table 6: Gap comparison on benchmark instances.

|      | TSPLib | | | CVRPLib | | | CVRPL | |
|------|--------|---------|---------|---------|---------|----------|------|------|
|      | 50–200 | 200–500 | 500–1000 | 100–200 | 200–500 | 500–1000 | 50 | 100 |
| SGE  | 109.59% | 287.53% | 660.36% | – | – | – | – | – |
| DRoC | 3.02% | 3.96% | 4.22% | 3.93% | 8.35% | – | 6.80% | 8.31% |
| ReEvo | 5.18% | 9.13% | 14.78% | 8.77% | 14.88% | 19.81% | – | – |
| AFL  | **1.28%** | **2.68%** | **2.98%** | **1.93%** | **5.20%** | **6.66%** | **5.57%** | **6.79%** |

diverse and challenging problem settings. Given the intrinsic difficulty of these problems and the lack of directly applicable advanced solvers, we adopt ACO and Greedy as baselines, as both are widely recognized flexible heuristics for complex VRPs. For ACO, the number of improvement steps is fixed at 500. The results in Table 4 demonstrate the consistent effectiveness of AFL. Although ACO is executed with 500 improvement steps, our framework attains better objective values in shorter runtimes. These empirical findings highlight the superiority of AFL on complex and practical VRPs, where traditional solvers often face limitations.

## 4.3 COMPARISON WITH LLM-BASED SOLVER

We compare AFL in trustworthiness and performance with representative LLM-based approaches for diverse VRPs (above 16 variants plus TSP), namely SGE (Iklassov et al., 2024) and DRoC (Jiang et al., 2025b), while excluding ARS (Li et al., 2025a) due to the unavailability of its source code.

To assess trustworthiness, we report the *Runtime Error Rate (RER)*, which measures the percentage of generated code that executes with errors, and the *Success Rate (SR)*, which measures the percentage of generated code that produces feasible solutions. As summarized in Table 5 and detailed in Appendix C.6, SGE is limited to solving only TSP, attaining an RER of 94.1% and an SR of 5.9%. DRoC extends to TSP, CVRP, and VRPL, with an RER of 82.4% and an SR of 17.6%. In contrast, AFL successfully handles all 17 tested VRP variants, reaching 0% RER and 100% SR, highlighting its superior code reliability and solution feasibility.

To assess performance, we further evaluate AFL on the problem classes (i.e., TSP, CVRP, and CVRPL) solvable by SGE and DRoC. Specifically, we conduct experiments on TSPLib (Reinelt, 1991), a real-world TSP benchmark containing 50 instances with sizes ranging from 50 to 1,000 customers, and on CVRPLib (Uchoa et al., 2017), a real-world CVRP benchmark containing 100 instances with sizes ranging from 100 to 1,000 customers. We further report the results of ReEvo (Ye et al., 2024) on TSPLib and CVRPLib, as both TSP and CVRP are solved in the original paper, to provide a broader assessment. For CVRPL, we adopt the same benchmark setting as in Section 4.1. The results are shown in Table 6, where gaps for TSPLib and CVRPLib are reported relative to their best-known solutions, while CVRPL gaps are measured against HGS. DRoC is able to solve CVRPLib instances only with fewer than 500 customers within the 10-hour time limit. Across all evaluated benchmarks, AFL consistently outperforms both SGE, DRoC, and ReEvo.

## 4.4 COMPARISON WITH PROMOTING STRATEGY

To evaluate the effectiveness of our model, we compare AFL with several classical prompting strategies, including standard prompting (Brown et al., 2020), self-refine (Madaan et al., 2023), self-debug (Chen et al., 2023), self-verification (Weng et al., 2022), and chain-of-thought (CoT) prompting (Wei et al., 2022). Table 9 presents the results, where the gaps are measured with respect to HGS-PyVRP. All strategies employ the same step-by-step function generation process for a fair comparison. The results show that AFL achieves the lowest runtime error rate and the highest success rate, as well as the best overall solution quality.

## 4.5 ABLATION STUDY

To study the necessity of the judgement agent (JA) and revision agent (RA), we run ablation experiments by removing them from AFL. The results are shown in Fig. 3. Without JA and RA, the framework frequently produces incorrect problem descriptions and invalid code. With RA included, the framework becomes more robust, yielding more accurate problem descriptions and executable

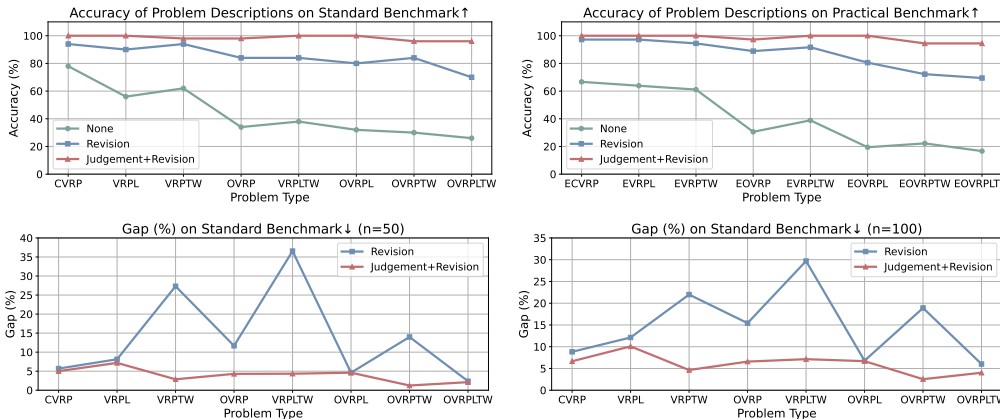

Figure 3: Ablation studies on the JA and RA.

code. With both JA and RA, the accuracy of problem description reaches almost 100%, and the framework produces reliable code and feasible solutions. This improvement arises because these agents ensures that all operator requirements are adequately considered during the code generation. These results verify our agentic design, showing that JA and RA are crucial for maintaining accurate problem descriptions and ensuring the trustworthiness of the generated code and derived solutions.

## 4.6 BROAD APPLICABILITY

We further evaluate AFL on four additional open benchmarks: TSP, ATSP, ACVRP, and SOP. Results for TSP are presented in Table 6. ATSP and ACVRP capture asymmetric routing, which frequently arises in real-world applications, while SOP introduces precedence-constrained path planning, another realistic and challenging setting. All three variants are formulated using non-Euclidean distance metrics. Specifically, the ATSP benchmark (Johnson & McGeoch, 1997) contains 18 instances ranging from 17 to 443 nodes. The ACVRP dataset (Helsgaun, 2017) provides 120 capacity-constrained cases with asymmetric distance matrices covering 16 to 200 customers. The SOP benchmark (Renaud et al., 1996) includes 39 precedence-constrained instances with sizes between 9 and 380 nodes. Experimental results for ATSP, ACVRP, and SOP are presented in Table 13, Table 14, and Table 15 in the Appendix, respectively. By achieving competitive performance across these diverse datasets, AFL demonstrates broad applicability, indicating that our framework has the potential to be extended to a wider range of problem variants.

## 5 CONCLUSION

This paper introduces AFL, an agentic LLM-based framework for solving complex vehicle routing problems (VRPs). Unlike prior approaches that depend on human intervention or predefined modules, AFL achieves self-containment and full automation by extracting domain knowledge directly from raw inputs and generating executable code and feasible solutions end-to-end. By decomposing the pipeline into three tractable subtasks and coordinating four specialized agents, AFL substantially improves code reliability and solution feasibility. Extensive experiments on 60 standard and practical VRP variants demonstrate the effectiveness, applicability, and trustworthiness of AFL.

The main limitation lies in performance, which do not yet surpass state-of-the-art solvers specifically designed for well-studied problems such as CVRP, a trade-off we consider acceptable given AFL's automation and generality. As future work, we plan to incorporate strategies such as evolutionary search to guide code generation and further enhance both code quality and search efficiency. Overall, our work highlights the potential of agentic LLMs as a general and trustworthy paradigm for solving complex combinatorial optimization problems with minimal domain knowledge, paving the way for more autonomous and adaptive optimization frameworks and democratizing access to advanced optimization techniques for non-expert users.

ACKNOWLEDGMENTS AND DISCLOSURE OF FUNDING

This research is supported by the National Research Foundation, Singapore under the AI Singapore Programme (AISG Award No: AISG3-RP-2022-031, AISG3-RPGV-2025-017).

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

APPENDIX

# A RELATED WORK

## A.1 ML FOR VRPS

Existing neural approaches to solving VRPs generally follow two directions: learning to construct and learning to improve them. In the construction paradigm, a model either directly generates a feasible solution (Vinyals et al., 2015; Bello et al., 2017) or produces a probability heatmap (Joshi et al., 2021; Sun & Yang, 2023) from which a solution is sampled. AM (Kool et al., 2018) first introduces attention mechanisms for VRPs, and POMO (Kwon et al., 2020) subsequently exploits multiple optima to enhance solution quality, inspiring a series of extensions and refinements (Drakulic et al., 2023; Luo et al., 2023; Berto et al., 2025a; Zhang & Cao, 2025). Recently, increasing attention has been devoted to improving generalization (Joshi et al., 2022; Zhou et al., 2023; Gao et al., 2024) and scalability (Li et al., 2021; Ouyang et al., 2025), addressing complex constraints (Bi et al., 2024; Li et al., 2025b), and handling multiple VRP variants within a single framework (Zhou et al., 2024; Berto et al., 2025b; Pan et al., 2025). As a representative work, Lin et al. (2024) propose a cross-problem learning framework for VRPs, leveraging a pre-trained Transformer backbone and a modular architecture to enable scalable and transferable adaptation to complex variants. However, these efforts still fall short of practical deployment. In the improvement paradigm, an initial solution is iteratively refined using learned local-search (Wu et al., 2021; Ma et al., 2023) or destroy–repair strategies (Hottung & Tierney, 2022) to progressively reduce cost and enhance solution quality.

## A.2 LLM FOR VRPS

One line of work uses LLMs to directly generate or improve VRP solutions. For example, OPRO (Yang et al., 2024) attempts to construct solutions outright with an LLM, LEMA (Liu et al., 2024b) leverages the LLM to perform the genetic search itself, and Jiang et al. (2025a) finetune LLM to construct in the end to end manner. However, they fall short in terms of solution quality and feasibility. A notable work in this line is DRAGON (Chen et al., 2026), a pioneering study providing the first compelling evidence that LLM-based agents can directly generate high-quality solutions for large-scale COPs, marking a significant advance in LLM-driven decision-making. Another line of research applies LLMs to generate code for VRPs, which can be broadly categorized into two directions: evolving basic heuristics for conventional VRPs and developing general frameworks for complex VRPs.

In *evolving basic heuristics*, LLMs are employed to iteratively evolve a simple or existing heuristic within fixed templates or established solvers for conventional VRPs. Early work such as EOH (Liu et al., 2024a) adopts a population-based framework with fixed templates for heuristic evolution. ReEvo (Ye et al., 2024) further combines evolutionary search with LLM reflections to provide verbal feedback and enhance search efficiency. More recently, HSEvo (Dat et al., 2025) and MCTS-AHD (Zheng et al., 2025) have advanced this line of research. Zhu et al. (2026) uses evolution framework to bridge synthetic and real routing problems. However, these approaches remain tailored to specific problems, limiting their generality across VRP variants with practical constraints.

In *developing general frameworks*, LLMs are leveraged as knowledgeable developers to generate function modules with distinct roles, enabling the framework to address diverse and complex VRPs. In ARS (Li et al., 2025a) and DRoC (Jiang et al., 2025b), LLMs generate specific functions to adapt frameworks to different variants, typically relying on handcrafted modules or external solvers. However, aligning LLM-generated code with these components remains challenging. In ARS, the integration between the LLM and the base algorithm is weak: the handcraft improvement operator and the constraint validation functions generated by the LLM are decoupled, and feasibility can only be verified after optimization, often leading to infeasible solutions. In DRoC, the LLM produces code that invokes OR-Tools without genuine knowledge of its internals, relying instead on external retrieval, which may undermine code reliability and reduce solution feasibility. In contrast, SGE (Iklassov et al., 2024) eliminates the need for predefined modules or solvers by directly generating code end-to-end. However, due to the inherent complexity of full code generation and the absence of effective constraint-handling mechanisms, its applicability is restricted to TSP. Moreover, it still depends on handcrafted extraction of instance information (e.g., inputs and constraints), and its overall performance remains uncompetitive.

Table 7: Constraint composition of VRP variants.

| VRP Variant | Capacity (C) | Open Route (O) | Backhaul (B) | Mixed Backhaul (MB) | Duration Limit (L) | Time Windows (TW) | Multi depot (MD) | Electric Vehicle (E) |
|---|---|---|---|---|---|---|---|---|
| CVRP | ✓ | | | | | | | |
| OCVRP | ✓ | ✓ | | | | | | |
| CVRPB | ✓ | | ✓ | | | | | |
| CVRPL | ✓ | | | | ✓ | | | |
| CVRPTW | ✓ | | | | | ✓ | | |
| OCVRPTW | ✓ | ✓ | | | | ✓ | | |
| OCVRPB | ✓ | ✓ | ✓ | | | | | |
| OCVRPL | ✓ | ✓ | | | ✓ | | | |
| CVRPBL | ✓ | | ✓ | | ✓ | | | |
| CVRPBTW | ✓ | | ✓ | | | ✓ | | |
| CVRPLTW | ✓ | | | | ✓ | ✓ | | |
| OCVRPBL | ✓ | ✓ | ✓ | | ✓ | | | |
| OCVRPBTW | ✓ | ✓ | ✓ | | | ✓ | | |
| OCVRPLTW | ✓ | ✓ | | | ✓ | ✓ | | |
| CVRPBLTW | ✓ | | ✓ | | ✓ | ✓ | | |
| OCVRPBLTW | ✓ | ✓ | ✓ | | ✓ | ✓ | | |
| CVRPMB | ✓ | | | ✓ | | | | |
| OCVRPMB | ✓ | ✓ | | ✓ | | | | |
| CVRPMBL | ✓ | | | ✓ | ✓ | | | |
| CVRPMBTW | ✓ | | | ✓ | | ✓ | | |
| OCVRPMBL | ✓ | ✓ | | ✓ | ✓ | | | |
| OCVRPMBTW | ✓ | ✓ | | ✓ | | ✓ | | |
| CVRPMBLTW | ✓ | | | ✓ | ✓ | ✓ | | |
| OCVRPMBLTW | ✓ | ✓ | | ✓ | ✓ | ✓ | | |
| MDCVRP | ✓ | | | | | | ✓ | |
| MDOCVRP | ✓ | ✓ | | | | | ✓ | |
| MDCVRPB | ✓ | | ✓ | | | | ✓ | |
| MDCVRPL | ✓ | | | | ✓ | | ✓ | |
| MDCVRPTW | ✓ | | | | | ✓ | ✓ | |
| MDOCVRPB | ✓ | ✓ | ✓ | | | | ✓ | |
| MDOCVRPL | ✓ | ✓ | | | ✓ | | ✓ | |
| MDOCVRPTW | ✓ | ✓ | | | | ✓ | ✓ | |
| MDOCVRPBL | ✓ | ✓ | ✓ | | ✓ | | ✓ | |
| MDOCVRPBTW | ✓ | ✓ | ✓ | | | ✓ | ✓ | |
| MDOCVRPLTW | ✓ | ✓ | | | ✓ | ✓ | ✓ | |
| MDCVRPMB | ✓ | | | ✓ | | | ✓ | |
| MDCVRPMBL | ✓ | | | ✓ | ✓ | | ✓ | |
| MDCVRPMBTW | ✓ | | | ✓ | | ✓ | ✓ | |
| MDOCVRPMB | ✓ | ✓ | | ✓ | | | ✓ | |
| MDOCVRPMBL | ✓ | ✓ | | ✓ | ✓ | | ✓ | |
| MDOCVRPMBTW | ✓ | ✓ | | ✓ | | ✓ | ✓ | |
| MDCVRPMBLTW | ✓ | | | ✓ | ✓ | ✓ | ✓ | |
| MDOCVRPMBLTW | ✓ | ✓ | | ✓ | ✓ | ✓ | ✓ | |
| ECVRP | ✓ | | | | | | | ✓ |
| ECVRPL | ✓ | | | | ✓ | | | ✓ |
| ECVRPTW | ✓ | | | | | ✓ | | ✓ |
| EOCVRP | ✓ | ✓ | | | | | | ✓ |
| ECVRPLTW | ✓ | | | | ✓ | ✓ | | ✓ |
| EOCVRPL | ✓ | ✓ | | | ✓ | | | ✓ |
| EOCVRPTW | ✓ | ✓ | | | | ✓ | | ✓ |
| EOCVRPLTW | ✓ | ✓ | | | ✓ | ✓ | | ✓ |

In this paper, we address these limitations by proposing a general framework of collaborative LLM-empowered agents that can tackle complex VRPs with self-containment, full automation, and high trustworthiness in both code and solutions.

## B PROBLEM STATEMENT

The connections between different VRP variants and their associated constraints are illustrated in Table 7. We further provide detailed descriptions of several representative variants as follows.

1. CVRP

- **Problem Type:** CVRP.
- **Description:** The Capacitated Vehicle Routing Problem (CVRP) involves determining optimal routes for a fleet of vehicles to deliver goods to a set of customers while minimizing total distance traveled and ensuring that vehicle capacity is not exceeded.
- **Constraints:** *Capacity* – the total demand on any route cannot exceed vehicle capacity. *Visit* – each customer is visited exactly once. *Depot* – every route starts and ends at the depot.

- **Input:** *depot*, *node_coordinates*, *demands*, *capacity*.
- **Output:** A set of vehicle routes, each beginning and ending at the depot, visiting every customer exactly once while respecting capacity constraints.
- **Objective:** Minimize the total travel distance.

2. CVRPL

- **Problem Type:** CVRPL.
- **Description:** The Capacitated Vehicle Routing Problem with a Distance Limit (CVRPL) involves optimizing routes for a fleet of vehicles with a fixed capacity, ensuring deliveries are made while adhering to a constraint on the maximum route distance, while all routes must start and end at a common depot.
- **Constraints:** *Capacity* – vehicles have limited capacity for carrying goods. *Distance Limit* – the total travel distance of each route must not exceed the specified maximum. *Visit* – each customer is visited exactly once. *Depot* – all routes must start and end at the fixed depot.
- **Input:** *depot*, *node_coordinates*, *demands*, *capacity*, *distance_limit*.
- **Output:** A set of feasible vehicle routes, each beginning and ending at the depot, visiting every customer exactly once while respecting both capacity and maximum-distance constraints.
- **Objective:** Minimize the total travel distance.

3. CVRPTW

- **Problem Type:** CVRPTW.
- **Description:** The problem is about optimizing delivery routes for a fleet of vehicles to serve a set of customers, considering time windows and vehicle capacity constraints. Each customer must be visited within a specific time frame and vehicles have limited capacity for deliveries.
- **Constraints:** *Capacity* – vehicles have limited capacity for deliveries. *Time Windows* – customers must be served within specified time intervals. *Visit* – every customer is visited exactly once. *Depot* – every route starts and ends at the depot.
- **Input:** *depot*, *node_coordinates*, *demands*, *capacity*, *service_times*, *time_windows*.
- **Output:** A set of vehicle routes, each beginning and ending at the depot, visiting every customer exactly once while satisfying vehicle capacity and customer time-window constraints.
- **Objective:** Minimize the total travel distance.

4. OCVRP

- **Problem Type:** OCVRP.
- **Description:** The Open Capacitated Vehicle Routing Problem (OCVRP) involves determining the optimal routes for a fleet of vehicles with limited capacity that start at a depot and must deliver goods to a set of customers without the obligation to return to the depot.
- **Constraints:** *Capacity* – vehicles have limited capacity. *Open Route* – vehicles are not required to return to the depot after completing service. *Visit* – each customer may only be visited once. *Depot* – every route must start at the depot.
- **Input:** *depot*, *node_coordinates*, *demands*, *capacity*.
- **Output:** A set of open vehicle routes, each starting at the depot and ending at a customer location, visiting every customer exactly once while satisfying the vehicle-capacity constraint.
- **Objective:** Minimize the total travel distance.

5. CVRPLTW

- **Problem Type:** CVRPLTW.
- **Description:** The problem is a Capacitated Vehicle Routing Problem with a Distance Limit and Time Windows (CVRPLTW), where a fleet of vehicles is tasked with delivering goods to a set of customers, each with specific demand and time windows, while respecting the vehicles' capacity and route limitations.
- **Constraints:** *Capacity* – vehicles have limited capacity. *Distance Limit* – each route has a maximum distance. *Time Windows* – customers must be served within specified time intervals. *Visit* – every customer is visited exactly once. *Depot* – every route starts and ends at the depot.
- **Input:** *depot*, *node_coordinates*, *demands*, *capacity*, *service_times*, *time_windows*, *distance_limit*.
- **Output:** A set of feasible vehicle routes, each beginning and ending at the depot, visiting every customer exactly once while satisfying capacity, distance-limit, and time-window constraints.
- **Objective:** Minimize the total travel distance.

6. OCVRPL

- **Problem Type:** OCVRPL.
- **Description:** The problem is a variant of the Open Capacitated Vehicle Routing Problem with a Distance Limit (OCVRPL), where vehicles must deliver goods to various customers while adhering to a capacity limitation and a maximum route distance, without the requirement to return to a depot.
- **Constraints:** *Capacity* – vehicles have limited capacity for the amount of goods they can transport. *Distance Limit* – each route has a maximum distance that vehicles must not exceed. *Open Route* – vehicles do not return to the depot after their delivery routes. *Visit* – each customer is visited only once. *Depot* – routes must start at the depot, specifically at the designated location.
- **Input:** *depot*, *node_coordinates*, *demands*, *capacity*, *distance_limit*.
- **Output:** A set of open vehicle routes, each starting at the depot and ending at a customer location, visiting every customer exactly once while satisfying capacity and distance-limit constraints.
- **Objective:** Minimize the total travel distance.

7. OCVRPTW

- **Problem Type:** OCVRPTW.
- **Description:** The Open Capacitated Vehicle Routing Problem with Time Windows (OCVRPTW) involves determining optimal routes for a fleet of vehicles with limited capacity that service a set of customers with specific time windows, without the requirement for vehicles to return to the depot after completing their deliveries.
- **Constraints:** *Capacity* – the total demand on any route cannot exceed vehicle capacity. *Time Windows* – customers must be served within specified time intervals. *Open Route* – vehicles do not return to the depot. *Visit* – each customer may only be visited once. *Depot* – every route must start at the depot.
- **Input:** *depot*, *node_coordinates*, *demands*, *capacity*, *service_times*, *time_windows*.
- **Output:** A set of feasible open vehicle routes, each starting at the depot and ending at a customer location, visiting every customer exactly once while satisfying capacity, service-time, and time-window constraints.
- **Objective:** Minimize the total travel distance.

8. OCVRPLTW

- **Problem Type:** OCVRPLTW.

- **Description:** The Open Capacitated Vehicle Routing Problem with Distance Limit and Time Windows (OCVRPLTW) requires planning optimal routes for a fleet of vehicles with limited carrying capacity to serve customers within specified time windows, while each route must also satisfy a maximum travel-distance limit and vehicles are not required to return to the depot after their final delivery.

- **Constraints:** *Capacity* – the total demand on any route cannot exceed vehicle capacity. *Distance Limit* – each route's total travel distance must not exceed the specified maximum. *Time Windows* – customers must be served within their given time intervals. *Open Route* – vehicles do not return to the depot after completing service. *Visit* – each customer is visited exactly once. *Depot* – every route must start at the depot.

- **Input:** *depot*, *node_coordinates*, *demands*, *capacity*, *service_times*, *time_windows*, *distance_limit*.

- **Output:** A set of feasible open vehicle routes, each starting at the depot and ending at a customer location, visiting every customer exactly once while satisfying capacity, time-window, and distance-limit constraints.

- **Objective:** Minimize the total travel distance.

9. ECVRP

- **Problem Type:** ECVRP.

- **Description:** Electric Capacitated Vehicle Routing Problem (ECVRP) involves determining optimal routes for a fleet of electric vehicles to serve a set of customer demands, considering vehicle load capacity and battery (fuel) constraints, where recharging is available at designated charging stations. Each route starts and ends at the depot and each customer is to be visited exactly once.

- **Constraints:** *Electricity* – Each vehicle has a limited battery, consumes energy proportional to distance, and may recharge only at designated charging stations, with charging time affecting feasibility. *Capacity* – Each vehicle has a maximum load capacity. *Depot* – Each route must start and end at the depot. *Visit* - Each customer must be visited exactly once.

- **Input:** *depot*, *node_coordinates*, *demands*, *capacity*, *fuel_capacity*, *fuel_consumption_rate*, *refuel_rate*, *stations*.

- **Output:** A set of feasible vehicle routes, each starting and ending at the depot, visiting each customer exactly once, specifying the visiting order and any charging station stops, such that vehicle capacity and battery constraints are satisfied.

- **Objective:** Minimize the total travel distance.

10. ECVRPL

- **Problem Type:** ECVRPL.

- **Description:** Electric Vehicle Capacitated Vehicle Routing Problem with distance limit: find the set of routes for electric vehicles, starting and ending at the depot, that serve all customers without exceeding vehicle capacity, electric fuel capacity, and an explicit route distance limit. Electricity is managed with fuel consumption, battery recharging at stations, and recharging time affects route scheduling.

- **Constraints:** *Electricity* – electric vehicles have limited battery, fuel consumption rate, designated charging stations, and recharging time which constrain feasible routes. *Capacity* – vehicles have a maximum load they can carry at one time. *Distance Limit* –each route cannot exceed a maximum distance. *Visit* – each customer is visited exactly once. *Depot* – all routes must start and end at the depot.

- **Input:** *depot*, *node_coordinates*, *demands*, *capacity*, *distance_limit*, *fuel_capacity*, *fuel_consumption_rate*, *refuel_rate*, *stations*.

- **Output:** a set of vehicle routes, each starting and ending at the depot, visiting every customer exactly once while satisfying capacity, distance-limit, and electric-vehicle energy constraints, including recharging stops if required.

- **Objective:** Minimize the total travel distance.

11. ECVRPTW

- **Problem Type:** ECVRPTW.
- **Description:** the Electric Capacitated Vehicle Routing Problem with Time Windows (ECVRPTW) constructs least-cost routes for a fleet of electric vehicles, each starting and ending at the depot, to serve all customers exactly once within specified time windows, considering both vehicle load capacity and electric battery (fuel) constraints; vehicles may recharge at designated charging stations, and service at each customer takes a specified time.
- **Constraints:** *Electricity* – Electric vehicles have limited battery capacity, consume energy proportional to travel distance, may recharge at specified stations, and recharging duration depends on refuel rate. *Time Windows* – Customers must be served within predetermined time intervals. *Capacity* – Each vehicle has limited load capacity. *Visit* – Each customer is visited exactly once. *Depot* – Every route must start and end at the depot.
- **Input:** *depot*, *node_coordinates*, *demands*, *capacity*, *service_times*, *time_windows*, *fuel_capacity*, *fuel_consumption_rate*, *refuel_rate*, *stations*.
- **Output:** A set of feasible vehicle routes, each starting and ending at the depot, visiting every customer exactly once while satisfying capacity, time-window, and electric-vehicle energy constraints, including necessary recharging stops.
- **Objective:** Minimize the total travel distance.

12. EOCVRP

- **Problem Type:** EOCVRP.
- **Description:** the Electric Open Capacitated Vehicle Routing Problem plans routes for a fleet of electric vehicles, where each vehicle starts at the depot, serves customer demands, may recharge at designated charging stations, does not need to return to the depot (open route), and respects vehicle capacity and battery limits. Each customer is visited exactly once.
- **Constraints:** *Electricity* – vehicles have limited battery capacity, consume energy with travel, and may recharge at stations, considering recharging time. *Capacity* – vehicle loads cannot exceed their capacity. *Open Route* – vehicles are not required to return to the depot. *Distance Limit* – each route has a maximum allowed travel distance. *Visit* – each customer must be visited exactly once. *Depot* – each route starts at the depot.
- **Input:** *depot*, *node_coordinates*, *demands*, *capacity*, *distance_limit*, *fuel_capacity*, *fuel_consumption_rate*, *refuel_rate*, *stations*.
- **Output:** a set of feasible open vehicle routes starting at the depot, each serving a subset of customers exactly once, possibly using charging stations, and ending at any node, such that all routes meet capacity, battery, and problem constraints
- **Objective:** Minimize the total travel distance.

13. ECVRPLTW

- **Problem Type:** ECVRPLTW.
- **Description:** Electric Capacitated Vehicle Routing Problem with Time Windows (ECVRPTW) involves designing routes for a fleet of electric vehicles to deliver goods to customers, respecting vehicle capacity limits, electric battery/range constraints, maximum route distance, and customer-specific time windows. Each customer must be visited exactly once, and all routes must start and end at the depot. Vehicles may need to recharge at designated stations as part of their routes.
- **Constraints:** *Electricity* – electric vehicles have limited battery (fuel) capacity, must recharge at stations as needed, and recharging consumes time. *Capacity* – each vehicle has a limited payload capacity. *Distance Limit* – each route has a maximum distance. *Time Windows* – customers must be served within given time intervals. *Visit* – each customer can be visited only once. *Depot* – each route must start and end at the depot.

- **Input:** *depot*, *node_coordinates*, *demands*, *capacity*, *service_times*, *time_windows*, *distance_limit*, *fuel_capacity*, *fuel_consumption_rate*, *refuel_rate*, *stations*.

- **Output:** A set of feasible vehicle routes, each represented as an ordered list of nodes (customers, stations, and depot), with associated service start times and charging operations, where each route starts and ends at the depot, all customers are visited exactly once within their time windows, and all constraints on capacity, time window, fuel, route length, and recharging are satisfied.

- **Objective:** Minimize the total travel distance.

14. EOCVRPL

- **Problem Type:** EOCVRPL.

- **Description:** the Electric Vehicle Open Capacitated Vehicle Routing Problem involves electric vehicles with limited load and battery capacity, serving customers starting from the depot without needing to return (open route). Vehicles can recharge at charging stations, and each customer is visited at most once.",

- **Constraints:** *Electricity* – Vehicles have a limited electric battery, consumed proportionally to travel; vehicles may recharge at designated stations. *Capacity* – Vehicles have a maximum load they can carry. *Open Route* – Vehicles do not need to return to the depot after serving customers. *Distance Limit* – Each route has a maximum allowable distance. *Visit* – Each customer is visited at most once. *Depot* – Routes start at the depot..

- **Input:** *depot*, *node_coordinates*, *demands*, *capacity*, *distance_limit*, *fuel_capacity*, *fuel_consumption_rate*, *refuel_rate*, *stations*.

- **Output:** A set of feasible open vehicle routes starting from the depot, where each route serves a subset of customers, may include recharging stops at stations as needed, and satisfies all capacity, distance, fuel, and visit constraints, ensuring each customer is visited at most once.

- **Objective:** Minimize the total travel distance.

15. EOCVRPTW

- **Problem Type:** EOCVRPTW.

- **Description:** Electric Open Capacitated Vehicle Routing Problem with Time Windows: The goal is to design minimum-cost routes for a fleet of electric vehicles that start from a depot, serve each customer exactly once within specified time windows, while observing vehicle capacity, electric battery limitations (with possible recharging at stations), and vehicles are not required to return to the depot.

- **Constraints:** *Electricity* – Vehicles have limited battery, consume energy with distance, and may recharge at designated stations. *Capacity* – Each vehicle has a fixed carrying capacity. *Time Windows* – Customers must be serviced within specific time intervals. *Open Route* – Vehicles do not return to the depot after serving customers. *Visit* – Each customer must be visited exactly once. *Depot* – Routes start from the depot.

- **Input:** *depot*, *node_coordinates*, *demands*, *capacity*, *service_times*, *time_windows*, *fuel_capacity*, *fuel_consumption_rate*, *refuel_rate*, *stations*.

- **Output:** a set of feasible vehicle routes where each route starts at the depot, visits each customer exactly once within their specified time windows and service times, respects vehicle capacity and battery constraints with possible recharging at stations, and does not return to the depot.

- **Objective:** Minimize the total travel distance.

16. EOCVRPLTW

- **Problem Type:** EOCVRPLTW.

- **Description:** Electric Open Capacitated Vehicle Routing Problem with Distance Limit and Time Windows. In this problem, a fleet of electric vehicles with limited cargo and battery

capacity must serve customers, each within specified distance limit and time windows. Vehicles start at the depot but do not return (open routes), must visit each customer exactly once, can recharge only at designated charging stations, and each route has a maximum allowable distance.

- **Constraints:** *Electricity* – Electric vehicles have limited battery (fuel) capacity, fuel consumed proportionally to distance, can recharge at charging stations, and recharging takes time. *Capacity* – Vehicles have limited cargo capacity. *Distance Limit* – Each route has a maximum allowed distance. *Time Windows* – Service at each customer must begin within its given time interval. *Open Route* – Vehicles do not return to depot after serving customers. *Visit* – Each customer must be visited exactly once. *Depot* – every route must start at the depot.

- **Input:** *depot*, *node_coordinates*, *demands*, *capacity*, *service_times*, *time_windows*, *distance_limit*, *fuel_capacity*, *fuel_consumption_rate*, *refuel_rate*, *stations*.

- **Output:** A set of feasible open vehicle routes, each starting at the depot, visiting every customer exactly once within their specified time windows, not exceeding vehicle cargo capacity, route distance limits, or battery constraints (with recharging at stations as needed), and ensuring vehicles do not return to the depot.

- **Objective:** Minimize the total travel distance.

17. TSP

- **Problem Type:** TSP.

- **Description:** The Symmetric Traveling Salesman Problem is to find the shortest possible route that visits each node (drilling location) exactly once and returns to the starting point, typically used for optimizing routes such as drilling or circuit board manufacturing.

- **Constraints:** *Visit* – each node (location) must be visited exactly once.

- **Input:** *node_coordinates*.

- **Output:** A single closed tour that begins and ends at one node and visits every node exactly once.

- **Objective:** Minimize the total travel distance.

18. ATSP

- **Problem Type:** ATSP.

- **Description:** The Asymmetric Traveling Salesman Problem (ATSP) generalizes the classical TSP by allowing the travel cost from node $i$ to node $j$ to differ from the cost from $j$ to $i$. The goal is to find the shortest Hamiltonian cycle that visits every node exactly once and returns to the starting node when edge weights are direction-dependent.

- **Constraints:** *Visit* – each node must be visited exactly once. *Depot* – the tour must start and end at the same node. *Asymmetry* – travel costs or distances are not necessarily symmetric.

- **Input:** *edge_weight_matrix*.

- **Output:** A single directed Hamiltonian cycle that begins and ends at one node and visits every node exactly once while respecting the asymmetric cost structure.

- **Objective:** Minimize the total travel cost.

19. ACVRP

- **Problem Type:** ACVRP.

- **Description:** Asymmetric Capacitated Vehicle Routing Problem (ACVRP): a fleet of vehicles with limited carrying capacity must start and end at a central depot and serve every customer exactly once, while accounting for direction-dependent travel costs.

- **Constraints:** *Capacity* – the total demand on any route cannot exceed the vehicle capacity. *Asymmetry* – travel costs between two nodes are not necessarily equal in both directions. *Visit* – each customer is visited exactly once. *Depot* – every route must start and end at the depot.

- **Input:** *depot*, *edge_weight_matrix*, *demands*, *capacity*.
- **Output:** A set of vehicle routes, each starting and ending at the depot, visiting every customer exactly once while satisfying vehicle-capacity constraints and accounting for direction-dependent travel costs.
- **Objective:** Minimize the total travel cost.

20. SOP

- **Problem Type:** SOP.
- **Description:** The Sequential Ordering Problem (SOP) is to find a minimum-cost Hamiltonian path that visits each node exactly once and respects given precedence constraints (some nodes must be visited before others), subject to forbidden arcs.
- **Constraints:** *Precedence* – Certain nodes must be visited before others, according to the instance's precedence relations. *Forbidden Arcs* – Some node-to-node connections are infeasible and cannot be used. *Visit* – each node is visited exactly once (Hamiltonian path).
- **Input:** *edge_weight_matrix*, *precedence_constraints*, *forbidden_arcs*.
- **Output:** A minimum-cost Hamiltonian path visiting each node exactly once that satisfies all precedence constraints and forbidden arc constraints.
- **Objective:** Minimize the total travel cost of the path.

## C    SUPPLEMENTARY METHODOLOGY AND EXPERIMENTAL DETAILS

### C.1    VRPLIB COMPONENT

Table 8 summarizes the main components of a VRPLIB instance file, presenting each required or optional field along with its description and an illustrative example. The "VRPLib Field" column lists the standard sections of a vehicle-routing problem instance (e.g., NAME/COMMENT, TYPE, DIMENSION). The Description column explains the meaning of each section, while the Example column provides typical syntax drawn from a sample instance (for example, TYPE : CVRP, CAPACITY : 200), showing the exact formatting used in VRPLIB files.

### C.2    DESTROY STRATEGY

As the algorithm shown in Algorithm 1, Given a solution $S$, distance matrix $Dis$, and ratio $\rho$, the algorithm first determines the number of nodes $n_{rm}$ to remove. A random customer $c$ is sampled, and a candidate list $L$ is formed by sorting all customers by distance to $c$. Iteratively, a candidate $u$ is selected, and if its route has not been destroyed, a contiguous subsequence $Q$ including $u$ is randomly chosen and removed. The subsequence $Q$ is added to the removed set $R$, and the corresponding route is updated and marked as destroyed. This process continues until $n_{rm}$ customers are removed, yielding the residual solution $S'$ and the removed set $R$.

### C.3    SIMULATED ANNEALING CRITERION

The *Simulated Annealing (SA) criterion* is a stochastic acceptance rule inspired by the physical annealing process, where a material is gradually cooled to reach a low–energy crystalline state. In VRPs scenarios, it provides a mechanism to escape local minima by occasionally accepting solutions that are worse than the current one.

Given a current solution with cost $E_{\text{current}}$ and a candidate solution with cost $E_{\text{new}}$, the move is accepted if $E_{\text{new}} \leq E_{\text{current}}$; otherwise it is accepted with probability

$$P = \exp\left(-\frac{E_{\text{new}} - E_{\text{current}}}{T}\right),$$

where $T$ is a temperature parameter that decreases according to a cooling schedule. In our implementation, $T$ is defined as

$$T = \frac{\text{iteration} - \text{step} + 1}{10},$$

where iteration denotes the total number of iterations and step is the current iteration index.

Table 8: Main sections of a VRPLIB instance file with descriptions and example.

| VRPLib Field | Description | Example |
|---|---|---|
| NAME / COMMENT | Optional. Includes the instance name and optional comments that can provide additional context for AFL to understand the related problem. | `NAME : A-n32-k5`
`COMMENT : 32-node`
`Capacitated Vehicle`
`Routing Problem`
`instance.` |
| TYPE | Optional. Specifies the declared problem type (e.g., CVRP, VRPTW, EVRP), serving only as a reference for constraint extraction; the actual problem characteristics must be determined from the complete instance. | `TYPE : CVRP` |
| DIMENSION | Required. Total number of nodes including both customers and depot(s). | `DIMENSION : 33` |
| EDGE WEIGHT TYPE | Required. Specifies how inter-node distances are given: a metric (e.g., EUC 2D) or an explicit matrix. | `EDGE WEIGHT TYPE : EUC`
`2D` |
| NODE COORD SECTION | Required. Lists coordinates of each node (typically planar Euclidean). | `NODE COORD SECTION:`
`1 45 68`
`2 37 52` |
| DEMAND SECTION | Required for instances with a capacity (C) constraint. Specifies the demand of each customer node, typically measured in weight or quantity units, where positive values denote linehaul customers (delivery) and negative values denote backhaul customers (pickup). | `DEMAND SECTION:`
`1 0`
`2 15`
`3 -10` |
| CAPACITY | Required for capacity-constrained (C) problems. Defines the maximum load (e.g., weight or volume) each vehicle can carry. | `CAPACITY : 200` |
| DEPOT SECTION | Required for instances with depot constraint. Identifies the depot node(s) where vehicles start and optionally end their routes. | `DEPOT SECTION:`
`1`
`-1` |
| DISTANCE LIMIT | Required for instances with open route (O) constraint. Specifies the maximum travel distance allowed for each route. | `DISTANCE LIMIT : 500` |
| TIME WINDOW SECTION | Required for instances with time window (TW) constraint. Provides earliest and latest allowable arrival times for each customer. | `TIME WINDOW SECTION:`
`1 0 100`
`2 20 80` |
| SERVICE TIME SECTION | Required for instances with time window (TW) constraint. Service time required at each customer node; combined with travel time to satisfy time-window constraints. | `SERVICE TIME SECTION:`
`1 10`
`2 15` |
| FUEL CAPACITY | Required for instances with electric vehicle (E) constraint. Maximum energy or battery capacity for each vehicle in electric-vehicle routing problems. | `FUEL CAPACITY : 300` |
| FUEL CONSUMPTION RATE | Required for instances with electric vehicle (E) constraint. Energy consumed per distance unit. | `FUEL CONSUMPTION RATE :`
`1.0` |
| REFUEL RATE | Required for instances with electric vehicle (E) constraint. Charging or refueling rate per time unit at stations. | `REFUEL RATE : 2.0` |
| STATION SECTION | Required for instances with electric vehicle (E) constraint. Identifies charging or refueling station nodes where vehicles can refuel. | `STATION SECTION:`
`34`
`35` |

---

**Algorithm 1** Destroy Strategy

---

1: **Input:** solution $S$ (set of routes), distance matrix $Dis$, ratio $\rho$
2: **Output:** removed nodes $R$, destroyed solution $S'$
3: $R \leftarrow \emptyset, S' \leftarrow S, n_{rm} \leftarrow \lfloor |C| \cdot \rho \rfloor$        ▷ $C$: set of customers
4: $c \sim \text{Uniform}(C)$           ▷ random sample center
5: $L \leftarrow \text{sort}(C, Dis(c, \cdot))$       ▷ sort candidate list by distance to $c$
6: **while** $|R| < n_{rm}$ and $L \neq \emptyset$ **do**
7:   $u \leftarrow \text{next}(L); r \leftarrow \text{route}(u, S')$
8:   **if** $r$ already destroyed **then**
9:    **continue**
10:   **end if**
11:   $Q \leftarrow \text{subseq}(r, u)$     ▷ random select contiguous subsequence including $u$
12:   $r \leftarrow r \setminus Q, R \leftarrow R \cup Q$     ▷ update $r$ by excluding $Q$ and extend $R$ with $Q$
13: **end while**
14: **return** $(R, S')$

---

Table 9: Gap comparison across different prompting strategies.

| | CVRP | | VRPL | | VRPTW | | OVRP | | VRPLTW | | OVRPL | | OVRPTW | | OVRPLTW | |
|---|---|---|---|---|---|---|---|---|---|---|---|---|---|---|---|---|
| | 50 | 100 | 50 | 100 | 50 | 100 | 50 | 100 | 50 | 100 | 50 | 100 | 50 | 100 | 50 | 100 |
| Standard Prompt | 11.76% | 12.42% | – | – | 5.11% | 8.14% | 10.60% | 13.77% | 4.58% | 7.38% | 14.59% | 23.56% | – | – | – | – |
| Self-Refine | 5.69% | 8.83% | 8.13% | 12.12% | – | – | – | – | – | – | 4.61% | 6.79% | 13.99% | 18.90% | 2.37% | 6.02% |
| Self-Debug | 11.76% | 12.42% | – | – | 5.11% | 8.14% | 10.60% | 13.77% | 4.58% | 7.38% | 14.59% | 23.56% | 43.39% | 51.39% | 12.57% | 14.37% |
| Self-Verification | 7.61% | 9.47% | – | – | 6.80% | 8.64% | 5.71% | 9.25% | – | – | – | – | 3.54% | 5.96% | – | – |
| COT | 8.39% | 12.04% | 7.37% | 10.51% | – | – | 56.84% | 93.83% | – | – | 62.06% | 105.86% | – | – | – | – |
| AFL | **5.01%** | **6.66%** | **7.18%** | **10.08%** | **2.87%** | **4.64%** | **4.30%** | **6.58%** | **4.34%** | **7.14%** | **4.61%** | **6.68%** | **1.24%** | **2.54%** | **2.12%** | **4.02%** |

**Note:** – indicates cases where no runnable code or feasible solution could be obtained.

## C.4 CODE GENERATION TIME ANALYSIS

At first, we provide the runtime breakdown during testing phrase in Table 10. These results, as noted in Lines 350–351, measure the combined runtime for Problem Description and Solution Derivation over 1,000 test instances using the already generated codes.

A detailed runtime breakdown of solver generation phase for the three subtasks: Problem Description, Code Generation, and Solution Derivation, across eight representative VRP variants, are shown in Table 10. These results reflect the time required to generate executable solver code and obtain a feasible solution for a single instance, where the generated code can be reused in the same VRP variant. As shown in Table 10, the code can be generated within approximately 30 minutes, which is notably shorter than the time typically required for human expert design, highlighting the model's strong efficiency in producing high-quality, executable, and feasible solver code within a short time.

In the Figure. 4, we explore the trade-off between solution quality and different number of solution refinement iterations.

## C.5 COMPARISON ON 48 STANDARD BENCHMARK

The results of the all 48 standard VRP variants are presented in Table 11, derived from 1,000 instances with 100 nodes each. AFL achieves results within 3% of the SOTA baseline HGS-PyVRP in most problem settings.

## C.6 DETAILS OF COMPARISON OF CODE RELIABILITY AND SOLUTION FEASIBILITY

**Runtime Error Rate (RER).** Let $V_{\text{err}}$ be the number of generated programs that terminate with a runtime failure. The *Runtime Error Rate* is calculated as

$$\text{RER} = \frac{V_{\text{err}}}{V} \times 100\%,$$

where $V$ is the total number of generated programs across all VRP variants. A high RER indicates a large proportion of solutions that fail to execute due to logical flaws, or syntax mistakes.

Table 10: Runtime Analysis Across Testing Phase and Solver Generation Phase.

| Phase | Component | CVRP | CVRPL | CVRPTW | OCVRP | CVRPLTW | OCVRPL | OCVRPTW | OCVRPLTW |
|---|---|---|---|---|---|---|---|---|---|
| **Testing** | Problem Description | 0.10 | 0.12 | 0.11 | 0.10 | 0.15 | 0.17 | 0.15 | 0.20 |
| | Solution Derivation | 2.10 | 6.93 | 9.31 | 2.20 | 8.90 | 5.65 | 11.15 | 17.27 |
| **Solver Generation** | Problem Description | 0.10 | 0.13 | 0.12 | 0.10 | 0.16 | 0.16 | 0.15 | 0.20 |
| | Code Generation | 12.28 | 12.88 | 18.38 | 23.94 | 27.43 | 28.20 | 24.85 | 30.50 |
| | Solution Derivation | 0.003 | 1.05 | 0.009 | 0.005 | 1.10 | 1.21 | 0.010 | 1.42 |

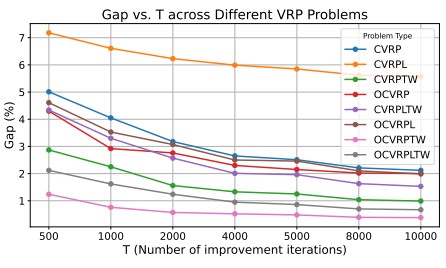 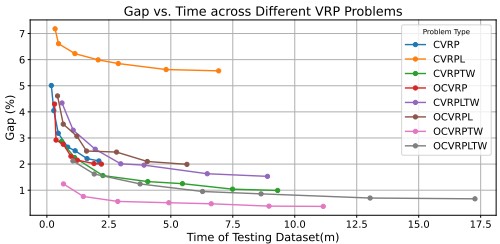

Figure 4: Comparison of solution quality across (a) different numbers of improvement iterations $T$ and (b) different testing time, evaluated over multiple VRP variants.

**Success Rate (SR).** Denote by $V_{\text{succ}}$ the number of generated programs that successfully produce an feasible solution for the target VRP instance. The *Success Rate* is given by

$$\text{SR} = \frac{V_{\text{succ}}}{V} \times 100\%.$$

This metric measures the percentage of generated programs that both execute without errors and produce a solution feasible with respect to the constraints of the input instance.

Table 12 compares the code reliability and solution feasibility of three solvers, SGE, DRoC, and AFL, across a broad range of VRP variants. Because SGE and DRoC can generate runtime-error-free code only for relatively simple problems where the constraints are easy to satisfy, SGE succeeds solely on TSP, while DRoC performs reliably on TSP, CVRP, and CVRPL, producing code that both compiles correctly and passes feasibility checks. In contrast, our model AFL consistently produces executable code and achieves feasibility verification across all listed variants. In this table, a ✓indicates that the solver achieves both Code Reliability and Solution Feasibility, whereas a ✗ denotes failure to meet one or both of these criteria.

## C.7 COMPARISON ON ATSP BENCKMARK

We evaluate our model on the ATSP benchmark (Johnson & McGeoch, 1997), which contains 18 instances with optimal solutions ranging from 17 to 443 nodes, to assess its applicability. The Asymmetric Traveling Salesman Problem (ATSP) is a variant of the classical TSP in which the distance from customer $i$ to customer $j$ may differ from the distance from customer $j$ to customer $i$, making the problem more challenging and representative of real-world routing scenarios such as one-way street networks or asymmetric transportation costs. The results, presented in Table 13, show that our model achieves consistently favorable performance on this task.

## C.8 COMPARISON ON ACVRP BENCKMARK

We further evaluate our model on the ACVRP benchmark (Helsgaun, 2017), which provides 120 capacity-constrained instances with asymmetric distance matrices and customer sizes ranging from 16 to 200. The Asymmetric Capacitated Vehicle Routing Problem (ACVRP) extends the classical VRP by allowing the travel cost from location $i$ to $j$ to differ from that of the reverse direction, while also imposing vehicle-capacity constraints. This combination of asymmetric travel costs and capacity limits makes ACVRP a challenging and practical testbed, reflecting real distribution networks where one-way streets or differing traffic conditions create directional cost differences. The results,

Table 11: Comparison results on 48 standard 100-node benchmark instances.

| | Obj. | Gap(%) | Obj. | Gap(%) | Obj. | Gap(%) | Obj. | Gap(%) | Obj. | Gap(%) | Obj. | Gap(%) |
|---|---|---|---|---|---|---|---|---|---|---|---|---|
| | CVRP | | CVRPL | | CVRPTW | | OCVRP | | CVRPLTW | | OCVRPL | |
| HGS-PyVRP | 15.62 | – | 15.77 | – | 25.42 | – | 9.73 | – | 25.76 | – | 9.72 | – |
| OR-Tools | 16.28 | 4.18 | 16.47 | 5.30 | 25.81 | 1.51 | 10.00 | 2.73 | 26.26 | 1.90 | 10.00 | 2.79 |
| RF-POMO | 15.91 | 1.83 | 16.11 | 2.17 | 26.34 | 3.58 | 10.18 | 4.66 | 26.78 | 3.95 | 10.18 | 4.66 |
| AFL ($T$=500) | 16.66 | 6.66 | 17.36 | 10.08 | 26.60 | 4.64 | 10.37 | 6.58 | 27.60 | 7.14 | 10.37 | 6.68 |
| AFL ($T$=2000) | 16.22 | 3.84 | 17.03 | 7.99 | 26.02 | 2.36 | 10.11 | 3.91 | 26.96 | 4.66 | 10.15 | 4.42 |
| AFL ($T$=10000) | **15.99** | **2.38** | 16.84 | 6.79 | **25.79** | **1.46** | **9.99** | **2.67** | 26.56 | 3.11 | **9.99** | **2.78** |
| | OCVRPTW | | OCVRPLTW | | CVRPB | | CVRPBL | | CVRPBTW | | OCVRPB | |
| HGS-PyVRP | 16.93 | – | 16.93 | – | 14.38 | – | 14.78 | – | 29.47 | – | 10.34 | – |
| OR-Tools | 17.03 | 0.58 | 17.02 | 0.73 | 14.93 | 3.85 | 15.43 | 4.34 | 29.95 | 1.60 | 10.58 | 2.32 |
| RF-POMO | 17.39 | 2.72 | 17.39 | 2.73 | 15.02 | 4.47 | 15.63 | 5.70 | 30.34 | 2.96 | 10.84 | 4.82 |
| AFL ($T$=500) | 17.36 | 2.54 | 17.61 | 4.02 | 15.48 | 7.65 | 16.18 | 9.47 | 31.50 | 6.89 | 11.67 | 12.86 |
| AFL ($T$=2000) | 17.14 | 1.24 | 17.32 | 2.30 | 15.07 | 4.80 | 15.71 | 6.29 | 30.63 | 3.94 | 11.34 | 9.67 |
| AFL ($T$=10000) | **17.04** | **0.65** | **17.19** | **1.54** | **14.74** | **2.50** | 15.41 | 4.26 | 30.17 | 2.38 | 11.12 | 7.54 |
| | CVRPBLTW | | OCVRPBL | | OCVRPBTW | | OCVRPBLTW | | MDCVRP | | MDCVRPL | |
| HGS-PyVRP | 29.03 | – | 10.34 | – | 19.16 | – | 19.16 | – | 11.89 | – | 11.90 | – |
| OR-Tools | 29.83 | 2.77 | 10.58 | 2.36 | 19.30 | 0.76 | 19.31 | 0.77 | 15.52 | 5.27 | 12.52 | 5.24 |
| RF-POMO | 30.80 | 3.28 | 10.84 | 4.83 | 19.61 | 2.34 | 19.61 | 2.34 | 15.11 | 30.10 | 16.25 | 37.19 |
| AFL ($T$=500) | 31.31 | 7.85 | 10.95 | 5.90 | 19.67 | 2.66 | 19.67 | 2.66 | 12.95 | 8.92 | 12.89 | 8.32 |
| AFL ($T$=2000) | 30.64 | 5.55 | 10.72 | 3.68 | 19.41 | 1.30 | 19.41 | 1.30 | 12.56 | 5.63 | 12.54 | 5.38 |
| AFL ($T$=10000) | 30.33 | 4.48 | **10.58** | **2.32** | **19.29** | **0.68** | **19.29** | **0.68** | 12.39 | 4.21 | 12.36 | 3.87 |
| | MDCVRPTW | | MDOCVRP | | MDCVRPLTW | | MDOCVRPL | | MDOCVRPTW | | MDOCVRPLTW | |
| HGS-PyVRP | 19.33 | – | 7.97 | – | 19.35 | – | 7.97 | – | 13.00 | – | 13.00 | – |
| OR-Tools | 19.62 | 1.55 | 8.16 | 2.33 | 19.66 | 1.58 | 8.16 | 2.33 | 13.09 | 0.74 | 13.09 | 0.70 |
| RF-POMO | 26.60 | 38.16 | 10.23 | 28.52 | 27.04 | 40.22 | 10.23 | 28.55 | 17.54 | 35.43 | 17.54 | 35.43 |
| AFL ($T$=500) | 20.33 | 5.17 | 8.35 | 4.77 | 20.91 | 8.06 | 8.36 | 4.89 | **13.28** | **2.12** | **13.28** | **2.15** |
| AFL ($T$=2000) | **19.85** | **2.69** | 8.21 | 3.01 | 20.52 | 6.05 | **8.20** | **2.97** | **13.14** | **1.05** | **13.14** | **1.08** |
| AFL ($T$=10000) | **19.61** | **1.45** | 8.14 | 2.12 | 20.28 | 4.81 | **8.13** | **2.01** | **13.07** | **0.50** | **13.07** | **0.54** |
| | MDCVRPB | | MDCVRPBL | | MDCVRPBTW | | MDOCVRPB | | MDCVRPBLTW | | MDOCVRPBL | |
| HGS-PyVRP | 11.64 | – | 11.68 | – | 22.03 | – | 8.69 | – | 22.06 | – | 8.69 | – |
| OR-Tools | 12.22 | 5.01 | 12.22 | 4.62 | 22.40 | 1.69 | 8.87 | 2.33 | 22.43 | 1.70 | 8.87 | 2.13 |
| RF-POMO | 15.11 | 30.10 | 15.71 | 34.80 | 30.46 | 38.80 | 10.89 | 25.48 | 30.97 | 41.00 | 10.89 | 25.49 |
| AFL ($T$=500) | 12.54 | 7.73 | 12.81 | 9.67 | 23.49 | 6.27 | 9.03 | 3.91 | 23.56 | 6.80 | 9.03 | 3.91 |
| AFL ($T$=2000) | 12.25 | 5.24 | 12.47 | 6.76 | 22.59 | 3.54 | **8.91** | **2.53** | 23.12 | 4.81 | **8.91** | **2.50** |
| AFL ($T$=10000) | 12.18 | 4.63 | 12.26 | 4.97 | **22.57** | **2.45** | **8.84** | **1.73** | 22.85 | 3.58 | **8.83** | **1.61** |
| | MDOCVRPBTW | | MDOCVRPBLTW | | CVRPMB | | CVRPMBL | | CVRPMBTW | | OCVRPMB | |
| HGS-PyVRP | 12.96 | – | 12.96 | – | 13.54 | – | 13.78 | – | 25.51 | – | 9.01 | – |
| OR-Tools | 14.49 | 11.79 | 14.49 | 11.79 | 14.93 | 10.27 | 15.42 | 11.90 | 29.97 | 17.48 | 10.59 | 17.54 |
| RF-POMO | 18.69 | 44.70 | 18.69 | 44.69 | 14.98 | 10.90 | 15.29 | 11.12 | 28.53 | 11.94 | 10.84 | 20.31 |
| AFL ($T$=500) | 13.40 | 3.40 | **13.22** | **2.01** | 14.81 | 9.38 | 14.62 | 6.09 | 27.38 | 7.33 | 9.38 | 4.11 |
| AFL ($T$=2000) | **13.26** | **2.31** | **13.09** | **1.00** | 14.38 | 6.20 | 14.18 | 2.90 | 26.71 | 4.70 | **9.23** | **2.43** |
| AFL ($T$=10000) | **13.20** | **1.85** | **13.04** | **0.62** | 14.17 | 4.65 | **13.91** | **0.94** | 26.40 | 3.49 | **9.14** | **1.43** |
| | CVRPMBLTW | | OCVRPMBL | | OCVRPMBTW | | OCVRPMBLTW | | MDCVRPMB | | MDCVRPMBL | |
| HGS-PyVRP | 25.85 | – | 9.01 | – | 16.97 | – | 16.97 | – | 10.68 | – | 10.71 | – |
| OR-Tools | 30.44 | 17.76 | 10.59 | 17.54 | 19.31 | 13.78 | 19.31 | 13.78 | 12.22 | 14.37 | 12.23 | 14.23 |
| RF-POMO | 28.89 | 11.89 | 10.84 | 20.32 | 18.62 | 9.72 | 18.62 | 9.71 | 15.09 | 41.78 | 15.37 | 44.05 |
| AFL ($T$=500) | 27.46 | 6.23 | 9.37 | 4.00 | **17.40** | **2.53** | **17.39** | **2.47** | 11.62 | 8.80 | 11.45 | 6.91 |
| AFL ($T$=2000) | **26.62** | **2.98** | **9.23** | **2.45** | **17.17** | **1.18** | **17.18** | **1.24** | 11.29 | 5.71 | 11.17 | 4.30 |
| AFL ($T$=10000) | **26.26** | **1.59** | **9.14** | **1.45** | **17.08** | **0.62** | **17.08** | **0.65** | 11.15 | 4.40 | **11.02** | **2.89** |
| | MDCVRPMBTW | | MDOCVRPMB | | MDCVRPLTW | | MDOCVRPMBL | | MDOCVRPMBTW | | MDOCVRPMBLTW | |
| HGS-PyVRP | 19.29 | – | 7.66 | – | 19.35 | – | 7.66 | – | 12.96 | – | 12.96 | – |
| OR-Tools | 22.39 | 16.12 | 8.88 | 15.83 | 19.66 | 1.58 | 8.87 | 15.78 | 14.49 | 11.79 | 14.49 | 11.79 |
| RF-POMO | 28.68 | 49.27 | 10.90 | 42.41 | 27.04 | 40.22 | 10.90 | 42.37 | 18.69 | 44.70 | 18.69 | 44.69 |
| AFL ($T$=500) | 20.62 | 6.89 | 7.91 | 3.26 | 20.91 | 8.06 | **7.89** | **3.00** | 13.40 | 3.40 | **13.22** | **2.01** |
| AFL ($T$=2000) | 20.12 | 4.30 | **7.80** | **1.83** | 20.52 | 6.05 | **7.80** | **1.82** | **13.26** | **2.31** | **13.09** | **1.00** |
| AFL ($T$=10000) | 19.89 | 3.11 | **7.74** | **1.04** | 20.28 | 4.81 | **7.75** | **1.17** | **13.20** | **1.85** | **13.04** | **0.62** |

**Note:** *Obj.* denotes the objective value, and *Gap(%)* represents the relative gap w.r.t. HGS-PyVRP. Lower values indicate better performance. Bold numbers denote cases where the gap from SOTA is within 3%, which is considered acceptable for a fully automated and self-contained framework.

summarized in Table 14, demonstrate that our model maintains strong code reliability and solution feasibility across all ACVRP instances.

Table 12: Comparison of Code Reliability and Solution Feasibility.

| Solver | TSP | CVRP | VRPL | VRPTW | OVRP | VRPLTW | OVRPL | OVRPTW | OVRPLTW |
|--------|-----|------|------|-------|------|--------|-------|--------|---------|
| SGE | ✓ | × | × | × | × | × | × | × | × |
| DRoC | ✓ | ✓ | ✓ | × | × | × | × | × | × |
| AFL | ✓ | ✓ | ✓ | ✓ | ✓ | ✓ | ✓ | ✓ | ✓ |

| Solver | ECVRP | ECVRPL | ECVRPTW | EOCVRP | ECVRPLTW | EOCVRPL | EOCVRPTW | EOCVRPLTW |
|--------|-------|--------|---------|--------|----------|---------|----------|-----------|
| SGE | × | × | × | × | × | × | × | × |
| DRoC | × | × | × | × | × | × | × | × |
| AFL | ✓ | ✓ | ✓ | ✓ | ✓ | ✓ | ✓ | ✓ |

Table 13: Comparison of AFL and Greedy on ATSP benchmark.

| Instance | ft53 | ft70 | ftv33 | ftv35 | ftv38 | ftv44 | ftv47 | ftv55 | ftv64 |
|----------|------|------|-------|-------|-------|-------|-------|-------|-------|
| Best | 6905 | 38673 | 1286 | 1473 | 1530 | 1613 | 1776 | 1608 | 1839 |
| Greedy Obj. | 8816 | 43524 | 1637 | 1817 | 1765 | 1898 | 2353 | 2163 | 2262 |
| Greedy Gap (%) | 27.68 | 12.54 | 27.29 | 23.35 | 15.36 | 17.67 | 32.49 | 34.51 | 23.00 |
| AFL Obj. | **7480** | **39519** | **1340** | **1500** | **1570** | **1657** | **1790** | **1630** | **1882** |
| AFL Gap (%) | **8.33** | **2.19** | **4.20** | **1.83** | **2.61** | **2.73** | **0.79** | **1.37** | **2.34** |

| Instance | ftv70 | ftv170 | kro124p | p43 | rbg323 | rbg358 | rbg403 | rbg443 | ry48p |
|----------|-------|--------|---------|-----|--------|--------|--------|--------|-------|
| Best | 1950 | 2755 | 36230 | 5620 | 1326 | 1163 | 2465 | 2720 | 14422 |
| Greedy Obj. | 2359 | 3887 | 45092 | 5688 | 1742 | 1794 | 3552 | 3876 | 16215 |
| Greedy Gap (%) | 20.97 | 41.09 | 24.46 | 1.21 | 31.37 | 54.26 | 44.10 | 42.50 | 12.43 |
| AFL Obj. | **2031** | **2964** | **37987** | **5620** | **1358** | **1172** | **2510** | **2780** | **14958** |
| AFL Gap (%) | **4.15** | **7.59** | **4.85** | **0.00** | **2.41** | **0.77** | **1.83** | **2.21** | **3.72** |

## C.9    COMPARISON ON SOP BENCKMARK

We also evaluate our model on the SOP benchmark (Renaud et al., 1996), which contains 39 instances with optimal solutions and sizes ranging from 9 to 380 nodes. The Sequential Ordering Problem (SOP) is a generalization of the Traveling Salesman Problem that introduces precedence constraints, requiring certain nodes to be visited before others. This added ordering requirement captures practical scenarios such as production sequencing and logistics scheduling, where tasks must follow a specified order. The results, presented in Table 15, show that our model effectively handles these precedence constraints while maintaining high solution quality.

## C.10    AFL'S SENSITIVITY TO DIFFERENT LLMS

We tested AFL with two additional LLMs: Claude-Sonnet-4-20250514 and GPT-4o. The results are shown in Table C.9. These experiments help us understand how different LLMs affect AFL's performance and stability.

**Solution Quality.** From the results, we see that the strength of the LLM does influence solution quality. More powerful models generate more accurate and complete heuristics, which leads to better objective values. In our tests, the overall performance ranking is: Claude-Sonnet-4 > GPT-4.1 > GPT-4o. Although the final performance changes with different LLMs, AFL improves how well each model implements the required heuristics. As shown in Table 9, AFL performs better than both the standard prompt and alternative prompt strategies, showing that AFL can reliably boost a model's ability to produce working heuristics.

**Code Reliability.** Most importantly, code reliability stays stable across all LLMs. No matter which model we use, AFL always generates executable code and feasible solutions. This shows that the multi-agent design keeps the whole pipeline stable even when the LLM is not very strong, proving that our framework is robust and widely applicable.

## C.11    EXPERIMENT ON DIFFERENT INPUT FORMAT

As VRPLIB is the standard format in the VRP domain, we further evaluated AFL on JSON and CSV inputs to assess its robustness to alternative data representations commonly used in industrial

Table 14: Comparison of AFL and Greedy on 120 ACVRP instances.

| Instance | A-G-100-1 | A-G-100-2 | A-G-100-3 | A-G-100-4 | A-G-100-5 | A-G-100-6 | A-G-100-7 | A-G-100-8 | A-G-100-9 | A-G-100-10 | A-G-100-11 | A-G-100-12 |
|---|---|---|---|---|---|---|---|---|---|---|---|---|
| Best Obj. | 2139 | 1722 | 2550 | 1273 | 1878 | 1540 | 1493 | 1467 | 2232 | 1631 | 1992 | 2057 |
| Greedy Gap (%) | 68.68 | 42.92 | 57.41 | 74.47 | 77.80 | 46.69 | 58.07 | 72.12 | 55.06 | 51.07 | 38.10 | 28.83 |
| AFL Gap (%) | **13.04** | **7.03** | **9.41** | **13.98** | **31.36** | **20.78** | **11.32** | **11.32** | **10.35** | **8.52** | **13.45** | **3.94** |

| Instance | A-G-100-13 | A-G-100-14 | A-G-100-15 | A-G-100-16 | A-G-100-17 | A-G-100-18 | A-G-100-19 | A-G-100-20 | A-G-150-1 | A-G-150-2 | A-G-150-3 | A-G-150-4 |
|---|---|---|---|---|---|---|---|---|---|---|---|---|
| Best Obj. | 2885 | 2101 | 1208 | 928 | 928 | 2427 | 2055 | 1929 | 1308 | 913 | 1619 | 930 |
| Greedy Gap (%) | 27.35 | 61.78 | 34.87 | 121.69 | 127.16 | 25.18 | 67.40 | 30.53 | 100.15 | 145.89 | 41.51 | 77.42 |
| AFL Gap (%) | **3.60** | **16.56** | **18.28** | **26.41** | **28.99** | **0.00** | **9.54** | **16.69** | **16.44** | **19.06** | **25.63** | **41.51** |

| Instance | A-G-150-5 | A-G-150-6 | A-G-150-7 | A-G-150-8 | A-G-150-9 | A-G-150-10 | A-G-150-11 | A-G-150-12 | A-G-150-13 | A-G-150-14 | A-G-150-15 | A-G-150-16 |
|---|---|---|---|---|---|---|---|---|---|---|---|---|
| Best Obj. | 1203 | 1138 | 1110 | 897 | 1525 | 892 | 1132 | 1187 | 1619 | 1490 | 1095 | 749 |
| Greedy Gap (%) | 79.88 | 92.71 | 93.51 | 130.21 | 71.28 | 80.61 | 100.09 | 72.03 | 70.72 | 49.19 | 125.11 | 87.85 |
| AFL Gap (%) | **20.20** | **25.57** | **13.15** | **36.23** | **24.59** | **15.25** | **23.85** | **27.46** | **14.76** | **7.85** | **33.88** | **36.98** |

| Instance | A-G-150-17 | A-G-150-18 | A-G-150-19 | A-G-150-20 | A-G-200-1 | A-G-200-2 | A-G-200-3 | A-G-200-4 | A-G-200-5 | A-G-200-6 | A-G-200-7 | A-G-200-8 |
|---|---|---|---|---|---|---|---|---|---|---|---|---|
| Best Obj. | 717 | 1610 | 1274 | 1177 | 1137 | 913 | 1492 | 819 | 1048 | 990 | 1016 | 814 |
| Greedy Gap (%) | 53.84 | 53.66 | 46.00 | 54.89 | 48.90 | 202.74 | 7.31 | 63.37 | 64.22 | 114.55 | 92.03 | 109.95 |
| AFL Gap (%) | **0.00** | **10.19** | **32.10** | **5.01** | **10.99** | **18.73** | **15.55** | **27.59** | **17.18** | **17.58** | **28.94** | **0.00** |

| Instance | A-G-200-9 | A-G-200-10 | A-G-200-11 | A-G-200-12 | A-G-200-13 | A-G-200-14 | A-G-200-15 | A-G-200-16 | A-G-200-17 | A-G-200-18 | A-G-200-19 | A-G-200-20 |
|---|---|---|---|---|---|---|---|---|---|---|---|---|
| Best Obj. | 1438 | 816 | 1039 | 1082 | 1514 | 1341 | 949 | 662 | 668 | 1522 | 1188 | 1035 |
| Greedy Gap (%) | 72.67 | 189.95 | 152.26 | 102.31 | 58.52 | 116.93 | 172.71 | 167.52 | 254.04 | 65.90 | 99.41 | 55.46 |
| AFL Gap (%) | **12.52** | **33.95** | **26.56** | **13.31** | **4.89** | **29.23** | **4.11** | **24.62** | **37.43** | **0.53** | **10.35** | **13.33** |

| Instance | A-U-4-1 | A-U-4-2 | A-U-4-3 | A-U-4-4 | A-U-4-5 | A-U-4-6 | A-U-4-7 | A-U-4-8 | A-U-4-9 | A-U-4-10 | A-U-4-11 | A-U-4-12 |
|---|---|---|---|---|---|---|---|---|---|---|---|---|
| Best Obj. | 1671 | 1108 | 1937 | 994 | 1447 | 1251 | 1142 | 1043 | 1826 | 1150 | 1523 | 1524 |
| Greedy Gap (%) | 75.10 | 42.60 | 54.93 | 93.16 | 74.91 | 78.02 | 109.81 | 90.32 | 35.32 | 140.35 | 42.88 | 111.15 |
| AFL Gap (%) | **20.35** | **10.74** | **0.00** | **5.53** | **19.42** | **42.13** | **23.82** | **41.61** | **10.46** | **27.65** | **19.63** | **25.33** |

| Instance | A-U-4-13 | A-U-4-14 | A-U-4-15 | A-U-4-16 | A-U-4-17 | A-U-4-18 | A-U-4-19 | A-U-4-20 | A-U-6-1 | A-U-6-2 | A-U-6-3 | A-U-6-4 |
|---|---|---|---|---|---|---|---|---|---|---|---|---|
| Best Obj. | 2096 | 1716 | 1290 | 995 | 825 | 1986 | 1622 | 1453 | 1205 | 913 | 1492 | 819 |
| Greedy Gap(%) | 36.12 | 82.81 | 94.57 | 44.12 | 87.27 | 34.44 | 70.28 | 53.68 | 64.23 | 66.81 | 57.98 | 37.73 |
| AFL Gap(%) | **13.74** | **14.34** | **35.74** | **9.05** | **24.97** | **6.70** | **8.45** | **9.02** | **26.22** | **12.27** | **7.71** | **28.94** |

| Instance | A-U-6-5 | A-U-6-6 | A-U-6-7 | A-U-6-8 | A-U-6-9 | A-U-6-10 | A-U-6-11 | A-U-6-12 | A-U-6-13 | A-U-6-14 | A-U-6-15 | A-U-6-16 |
|---|---|---|---|---|---|---|---|---|---|---|---|---|
| Best Obj. | 1086 | 990 | 1016 | 814 | 1438 | 816 | 1071 | 1111 | 1544 | 1341 | 949 | 662 |
| Greedy Gap (%) | 90.98 | 133.64 | 92.03 | 215.11 | 93.88 | 262.62 | 101.87 | 92.53 | 64.12 | 88.22 | 172.71 | 283.38 |
| AFL Gap (%) | **6.72** | **19.80** | **8.17** | **36.86** | **13.42** | **32.11** | **36.13** | **29.52** | **26.04** | **27.96** | **12.22** | **43.35** |

| Instance | A-U-6-17 | A-U-6-18 | A-U-6-19 | A-U-6-20 | A-U-8-1 | A-U-8-2 | A-U-8-3 | A-U-8-4 | A-U-8-5 | A-U-8-6 | A-U-8-7 | A-U-8-8 |
|---|---|---|---|---|---|---|---|---|---|---|---|---|
| Best Obj. | 668 | 1563 | 1193 | 1069 | 981 | 753 | 1159 | 737 | 832 | 923 | 871 | 691 |
| Greedy Gap (%) | 98.05 | 40.24 | 102.51 | 47.52 | 121.10 | 1771.31 | 72.04 | 62.14 | 116.23 | 127.63 | 135.02 | 147.76 |
| AFL Gap (%) | **0.00** | **1.41** | **41.16** | **18.71** | **36.70** | **29.22** | **32.18** | **31.89** | **26.20** | **17.77** | **52.93** | **27.93** |

| Instance | A-U-8-9 | A-U-8-10 | A-U-8-11 | A-U-8-12 | A-U-8-13 | A-U-8-14 | A-U-8-15 | A-U-8-16 | A-U-8-17 | A-U-8-18 | A-U-8-19 | A-U-8-20 |
|---|---|---|---|---|---|---|---|---|---|---|---|---|
| Best Obj. | 1295 | 716 | 900 | 827 | 1091 | 1178 | 682 | 580 | 591 | 1298 | 1015 | 854 |
| Greedy Gap (%) | 53.51 | 73.46 | 116.50 | 187.30 | 81.58 | 125.64 | 250.59 | 72.41 | 184.43 | 45.61 | 93.00 | 133.02 |
| AFL Gap (%) | **4.48** | **35.47** | **29.50** | **35.31** | **26.67** | **15.20** | **0.00** | **15.69** | **35.03** | **7.86** | **14.09** | **9.84** |

Table 15: Comparison of AFL and Greedy on SOP instances.

| Instance | ESC12 | ESC25 | ESC47 | ESC63 | ESC78 | br17.10 | br17.12 | ft53.1 | ft53.2 | ft53.3 |
|---|---|---|---|---|---|---|---|---|---|---|
| Best Obj. | 1675 | 1681 | 1288 | 62 | 18230 | 55 | 55 | 7531 | 8062 | 10262 |
| Greedy Gap (%) | 21.43 | 99.88 | 198.37 | 22.58 | 100.00 | 43.64 | 43.64 | 38.15 | 56.98 | 47.41 |
| AFL Gap (%) | **3.16** | **12.14** | **84.32** | **0.00** | **0.00** | **5.45** | **5.45** | **5.11** | **1.28** | **12.45** |

| Instance | ft53.4 | ft70.1 | ft70.2 | ft70.3 | ft70.4 | kro124p.1 | kro124p.2 | kro124p.3 | kro124p.4 | p43.1 |
|---|---|---|---|---|---|---|---|---|---|---|
| Best Obj. | 14425 | 39313 | 40419 | 42535 | 53530 | 39420 | 41336 | 49499 | 76103 | 28140 |
| Greedy Gap (%) | 28.59 | 17.16 | 19.64 | 22.41 | 100.00 | 33.37 | 39.64 | 56.10 | 29.33 | 5.29 |
| AFL Gap (%) | **0.93** | **4.16** | **4.89** | **7.03** | **0.88** | **8.66** | **10.29** | **7.28** | **1.69** | **0.23** |

| Instance | p43.2 | p43.3 | p43.4 | prob42 | prob100 | rbg048a | rbg050c | rbg109a | rbg150a | rbg174a |
|---|---|---|---|---|---|---|---|---|---|---|
| Best Obj. | 28480 | 28835 | 83005 | 243 | 1163 | 351 | 467 | 1038 | 1750 | 2033 |
| Greedy Gap (%) | 4.37 | 8.69 | 2.70 | 88.48 | 184.69 | 44.16 | 21.63 | 39.02 | 23.89 | 20.22 |
| AFL Gap (%) | **0.02** | **0.99** | **0.05** | **30.45** | **94.41** | **6.55** | **0.64** | **0.77** | **0.46** | **0.00** |

| Instance | rbg253a | rbg323a | rbg341a | rbg358a | rbg378a | ry48p.1 | ry48p.2 | ry48p.3 | ry48p.4 |
|---|---|---|---|---|---|---|---|---|---|
| Best Obj. | 2950 | 3140 | 2568 | 2545 | 2816 | 15805 | 16074 | 19490 | 31446 |
| Greedy Gap (%) | 20.61 | 28.41 | 47.43 | 61.49 | 45.92 | 42.32 | 30.09 | 40.29 | 30.94 |
| AFL Gap (%) | **0.00** | **1.21** | **3.08** | **2.67** | **2.66** | **5.90** | **6.07** | **12.72** | **0.73** |

pipelines and benchmark integrations. As shown in Table 17, the model consistently maintains strong performance and solution feasibility across different input formats.

## C.12 EXPERIMENT ON LARGE CVRP INSTANCES

AFL is capable of handling extremely large VRP instances. Our current approach generates a general-purpose heuristic from a small instance (e.g., a 50-node instance, which can be obtained by trimming the data of the target instance and serves as a minimal example containing all relevant information such as constraints, problem characteristics, and format specifications). The generated solver is then applied to large-scale cases. As shown in Table 18, AFL achieves competitive performance on CVRPLib-XXL (Gao et al., 2024) with more than 16,000 customers.

Table 16: Comparison results on different LLM.

| | | n=50 | | | n=100 | | | | n=50 | | | n=100 | | |
|---|---|---|---|---|---|---|---|---|---|---|---|---|---|---|
| | | Obj. | Gap (%) | Time (m) | Obj. | Gap (%) | Time (m) | | Obj. | Gap (%) | Time (m) | Obj. | Gap (%) | Time (m) |
| HGS-PyVRP | CVRP | 10.37 | – | 10.40 | 15.62 | – | 20.80 | CVRPL | 10.59 | – | 10.40 | 15.77 | – | 20.80 |
| GPT 4o | | 10.64 | 2.60 | 2.21 | 16.05 | 2.75 | 4.96 | | 11.01 | 3.97 | 13.06 | 16.64 | 5.52 | 34.25 |
| Claude 4 | | **10.54** | **1.65** | 2.60 | **15.90** | **1.79** | 5.12 | | **10.91** | **3.02** | 6.85 | **16.44** | **4.25** | **25.79** |
| GPT 4.1 | | 10.59 | 2.12 | 2.10 | 15.99 | 2.38 | 4.38 | | 11.18 | 5.57 | 6.93 | 16.84 | 6.79 | 25.48 |
| HGS-PyVRP | CVRPTW | 16.03 | – | 10.40 | 25.42 | – | 20.80 | OCVRP | 6.51 | – | 10.40 | 9.73 | – | 20.80 |
| GPT 4o | | 16.45 | 2.62 | 12.62 | 26.88 | 5.74 | 25.44 | | 6.87 | 5.53 | 3.07 | 10.36 | 6.47 | 7.38 |
| Claude 4 | | 16.28 | 1.56 | 8.50 | 26.19 | 3.03 | 33.78 | | **6.60** | **1.38** | 2.95 | **9.94** | **2.26** | 7.22 |
| GPT 4.1 | | **16.19** | **0.99** | 9.31 | **25.79** | **1.46** | 38.45 | | 6.64 | 2.00 | 2.20 | 9.99 | 2.67 | 5.52 |
| HGS-PyVRP | CVRPLTW | 16.36 | – | 10.40 | 25.76 | – | 20.80 | OCVRPL | 6.51 | – | 10.40 | 9.72 | – | 20.80 |
| GPT 4o | | 16.79 | 2.63 | 8.54 | 26.81 | 4.08 | 35.48 | | 6.81 | 4.61 | 7.13 | 10.33 | 6.28 | 13.42 |
| Claude 4 | | **16.55** | **1.16** | 7.32 | **26.50** | **2.87** | 30.94 | | **6.59** | **1.23** | 7.43 | **9.91** | **1.95** | 16.59 |
| GPT 4.1 | | 16.61 | 1.53 | 8.90 | 26.56 | 3.11 | 35.33 | | 6.64 | 1.99 | 5.65 | 9.99 | 2.78 | 16.27 |
| HGS-PyVRP | OCVRPTW | 10.51 | – | 10.40 | 16.93 | – | 20.80 | OCVRPLTW | 10.51 | – | 10.40 | 16.93 | – | 20.80 |
| GPT 4o | | 10.91 | 3.81 | 11.89 | 17.68 | 4.43 | 27.39 | | 10.88 | 3.52 | 11.29 | 17.66 | 4.31 | 58.89 |
| Claude 4 | | 10.63 | 1.14 | 14.89 | 17.24 | 1.83 | 35.95 | | 10.64 | 1.24 | 12.23 | 17.26 | 1.95 | 62.24 |
| GPT 4.1 | | **10.55** | **0.38** | 11.15 | **17.04** | **0.65** | 49.05 | | **10.58** | **0.67** | 17.27 | **17.19** | **1.54** | 70.31 |

Table 17: Combined Results on JSON and CSV Format Inputs.

| | | n=50 | | | n=100 | | | | n=50 | | | n=100 | | |
|---|---|---|---|---|---|---|---|---|---|---|---|---|---|---|
| | | Obj. | Gap (%) | Time (m) | Obj. | Gap (%) | Time (m) | | Obj. | Gap (%) | Time (m) | Obj. | Gap (%) | Time (m) |
| JSON | CVRP | 10.68 | 2.99 | 2.69 | 16.36 | 3.71 | 4.99 | CVRPL | 10.86 | 2.55 | 7.37 | 16.42 | 4.12 | 29.20 |
| CSV | | 10.69 | 3.09 | 2.61 | 16.40 | 4.99 | 5.50 | | 10.75 | 1.51 | 4.73 | 16.31 | 3.42 | 20.72 |
| JSON | CVRP TW | 16.26 | 1.43 | 12.97 | 26.17 | 2.95 | 21.22 | OCVRP | 6.71 | 3.07 | 2.07 | 10.22 | 5.04 | 5.38 |
| CSV | | 16.19 | 1.00 | 14.87 | 26.07 | 2.56 | 26.11 | | 6.68 | 2.61 | 2.80 | 10.14 | 4.21 | 6.17 |
| JSON | CVRPL TW | 16.63 | 1.65 | 8.02 | 26.52 | 2.95 | 31.64 | OCVRP L | 6.73 | 3.38 | 6.78 | 10.26 | 5.56 | 16.46 |
| CSV | | 16.61 | 1.52 | 8.99 | 26.48 | 2.80 | 32.10 | | 6.71 | 3.07 | 7.96 | 10.11 | 4.01 | 18.24 |
| JSON | OCVRP TW | 10.58 | 0.67 | 14.73 | 17.19 | 1.54 | 55.47 | OCVRP LTW | 10.55 | 0.38 | 19.62 | 17.15 | 1.30 | 82.31 |
| CSV | | 10.60 | 0.86 | 10.26 | 17.25 | 1.89 | 50.99 | | 10.56 | 0.48 | 16.65 | 17.16 | 1.36 | 69.19 |

During code generation, we also enforce a practical constraint: during solution derivation, the algorithm must execute within 10 seconds for the given instance. If this limit is exceeded, the system triggers a timeout, then the Revision Agent (RA) and Error Analysis Agent (EAA) are trigered to refine the algorithm for improved efficiency. We will add clarifications of this mechanism in the revised manuscript. Looking ahead, we will further enhance efficiency by exploring multi-objective evolutionary strategies to evolve the generated algorithms, jointly optimizing both performance and computational efficiency.

# D    EXAMPLES OF PROMPTS AND OUTPUTS

## D.1    SUBTASK 1: PROBLEM DESCRIPTION

In the problem description subtask, we divide the GA's work into two sequential phases to reduce its cognitive load and improve overall accuracy. In the first phase, the agent generates the description, constraints, and the specific problem type. In the second phase, it produces the input specification, expected output, and the optimization objective. After these two phases are completed, the JA evaluates the entire draft for consistency and correctness, and the RA subsequently refines each component as needed.

### D.1.1    GENERATION AGENT(GA)

**Prompt 1**:

> We need to solve a VRP instance. I will provide you with the instance. Please analyze it carefully. First, give a concise description of the problem type in [ ], explaining what the problem is about.

Table 18: Performance of AFL on CVRPLib-XXL

| Gap(%) | L1(3k) | L2(4k) | A1(6k) | A2(7k) | G1(10k) | G2(11k) | B1(15k) | B2(16k) |
|--------|--------|--------|--------|--------|---------|---------|---------|---------|
| POMO | 75.30 | 78.16 | 112.27 | 159.22 | – | – | – | – |
| LEHD | 14.04 | 26.30 | 18.90 | 26.40 | 27.23 | 38.45 | 35.94 | 40.76 |
| Our | **8.12** | **13.34** | **8.60** | **13.62** | **8.02** | **14.85** | **7.01** | **16.56** |

Second, identify its constraints and list them clearly in numbered format (1), 2), 3), ...) within [ ]. For each constraint, write both the abbreviation (if any) and a short explanation (e.g., 'Capacity (C): vehicles have limited capacity.'). Do not include instance-specific details like the exact number of nodes, vehicles, or capacity values.

When analyzing, be as comprehensive as possible: consider not only the common constraints but also more general ones, such as whether each customer can be visited multiple times, whether all routes must start and end at a depot, or other structural constraints that might apply. You may refer to the following typical constraint categories as guidance, but you are not limited to them:

- Electricity (E): electric vehicles are subject to fuel constraints. Each vehicle has a limited fuel capacity, fuel is consumed proportionally to the distance traveled related to fuel consumption rate, and vehicles must recharge at designated charging stations when necessary. Recharging consumes time related to the refuel rate and remaining fuel.

- Capacity (C): vehicles have limited capacity.

- Open Route (O): vehicles do not return to the depot.

- Backhaul (B): vehicles must handle linehaul (deliveries) first, then backhaul (pickups).

- Mixed (M): vehicles may mix deliveries and pickups in the same route.

- Duration Limit (L): each route has a maximum distance or time limit.

- Time Windows (TW): customers must be served within specified time intervals.

- Multi-depot (MD): there are multiple depots.

- Visit constraint (V): each customer may only be visited once.

- Depot constraint (D): routes must start and end at the depot.

Finally, write the standard problem type abbreviation (e.g., TSP, CVRP, CVRPL, VRPTW, PDP, OVRP, MDVRP, ECVRP) enclosed in " ".

Check again whether the constraints fit the specific name; the Visit constraint should be present.

Output exactly three lines in this order:

1) [problem description]

2) [constraints]

3) "specific name"

Below is the .vrp instance content: `<vrp text>`

**Output 1:**

1) [The Capacitated Vehicle Routing Problem (CVRP) involves determining optimal routes for a fleet of vehicles to deliver goods to a set of customers while minimizing total distance traveled and ensuring that vehicle capacity is not exceeded.]

2) [1) Capacity (C): the total demand on any route cannot exceed vehicle capacity. 2) Visit (V): each customer is visited exactly once. 3) Depot (D): every route starts and ends at the depot.]

3) "CVRP"

**Prompt 2:**

We need to design an algorithm for the following VRP instance. The details of the instance are: `<Output1>`.

Based on this description and the instance contents, please specify:

First, list the essential input elements an algorithm would require from the instance. Element names must not contain spaces; use underscores _ instead.

Second, describe precisely what the algorithm should output (e.g., a best feasible set of vehicle routes that satisfy all listed constraints).

Third, describe clearly the optimization objective (e.g., minimize total travel distance, minimize fleet size, minimize lateness).

Important: Each of the three answers (input, output, objective) must be enclosed in [ ] as shown. Do not include instance-specific details like the exact number of nodes, vehicles, or capacity values. Also verify that every listed input is actually provided by the instance content.

Output exactly three lines in this order:

4) [input]

5) [output]

6) [objective]

Below is the .vrp instance content: `<vrp text>`

**Output 2:**

4) [depot, node_coordinates, demands, vehicle_capacity]

5) [A set of vehicle routes, each beginning and ending at the depot, visiting every customer exactly once while respecting capacity constraints.]

6) [Minimize the total travel distance.]

D.1.2 JUDGMENT AGENT (JA)

**Prompt:**

You are a VRP expert. I will give you:
1) The original .vrp file content
2) GPT's first answer (problem description + constraints + specific name)
3) GPT's second answer (input, output, objective)

Your task is to judge correctness:

- For the first answer: check whether the problem description, listed constraints, and specific name are consistent with .vrp file contents. Specifically, check for contradictions in the following pairs:

• problem description vs. .vrp file

---

- constraints vs. .vrp file

- specific name vs. .vrp file

- problem description vs. constraints

- problem description vs. specific name

- constraints vs. specific name

If any contradictions exist, treat the .vrp file as the ground truth and mark it as incorrect. If everything is consistent, mark it as correct. Check again whether the constraints fit the specific name and problem description. If correct, return 'True' with a short explanation. If wrong, return'False' with a short explanation.

- For the second answer: check if input, output, and optimization objective are valid and consistent with the VRP instance and the constraints. Input must correspond only to elements explicitly defined in the instance file. Every input listed must be directly obtainable from the instance. Input element names must not contain spaces; use underscores _. Output must clearly describe feasible vehicle routes respecting all constraints. The optimization objective must align with VRP goals.

If any contradictions exist, treat the .vrp file as the ground truth and mark it as incorrect.If everything is consistent, mark it as correct.If correct, return 'True' with a short explanation. If wrong, return 'False' with a short explanation.

Output format must be exactly 4 lines:

1) right1: True/False

2) jud1: explanation

3) right2: True/False

4) jud2: explanation

Here is the VRP file: `<vrp_text>`

Here is GPT's first answer: `<problem_description_1>`

Here is GPT's second answer: `<problem_description_2>`

---

**Output:**

---

1) right1: True

2) jud1: The first answer accurately reflects the .vrp file, providing a clear problem description and consistent constraints (Capacity, Visit, Depot) along with the correct specific name, with no contradictions.

3) right2: False

4) jud2: The second answer includes input or objective elements that are not fully supported by the CVRP instance—specifically, it omits capacity from the input—so it is not fully consistent with the file's available data and constraints.

---

### D.1.3 REVISION AGENTS (RA)

The formats of Output 1 and Output 2 are identical to those in Section D.1.1.

**Prompt 1:**

We need to solve a VRP instance. I will provide you with the instance. Please analyze it carefully.

Step 1: Give a concise description of the problem type in [ ], explaining what the problem is about.

Step 2: Identify its constraints and list them clearly in numbered format (1), 2), 3), ...) within [ ]. For each constraint, write both the abbreviation (if any) and a short explanation (e.g., 'Capacity (C): vehicles have limited capacity.'). Do not include instance-specific details like the exact number of nodes, vehicles, or capacity values. Consider whether customers may be visited once or multiple times, and whether all routes must start/end at a depot (unless open routes are specified).

Reference constraint categories (not exhaustive):

- Electricity (E): electric vehicles are subject to fuel constraints. Each vehicle has a limited fuel capacity, fuel is consumed proportionally to the distance traveled related to fuel consumption rate, and vehicles must recharge at designated charging stations when necessary. Recharging consumes time related to the refuel rate and remaining fuel.

- Capacity (C): vehicles have limited capacity.

- Open Route (O): vehicles do not return to the depot.

- Backhaul (B): deliveries first, then pickups.

- Mixed (M): deliveries and pickups can be mixed.

- Distance Limit (L): each route has a maximum distance or time limit.

- Time Windows (TW): customers must be served within specific time intervals.

- Multi-depot (MD): multiple depots instead of one.

- Visit constraint (V): whether each customer can be visited only once.

- Depot constraint (D): routes must start and end at the depot.

Step 3: Write the standard problem type abbreviation (e.g., TSP, CVRP, CVRPL) enclosed in " ". Ensure the abbreviation is consistent with the constraints.

Check again that the constraints fit the specific name, and include the Visit constraint.

Here is your previous answer: `<ans>`

However, there were some issues identified: `<jud>`

Now, please correct your answer strictly according to the rules above.

Output format (exactly three lines):

1) [problem description]

2) [constraints]

3) "specific name"

Below is the .vrp instance content: `<vrp text>`

**Prompt 2:**

We need to design an algorithm for the following VRP instance. The details of the instance are: `<Output 1>`.

Based on this description and the instance contents, please provide:

Step 1: List the essential elements an algorithm would require from the instance, and the list must include depot.

Step 2: Describe precisely what the algorithm should output (e.g., a set of feasible vehicle routes that satisfy all listed constraints).

Step 3: Describe clearly the optimization objective (e.g., minimize total travel distance, minimize fleet size, minimize lateness).

Important rules:

- Each of the three answers (input, output, objective) must be enclosed in [ ] exactly as shown.

- Do not include instance-specific details like the exact number of nodes, vehicles, or capacity values.

- Step 1 element names must not contain spaces; use underscores _ instead.

Here is your previous answer: `<ans>`

Issues identified in that answer: `<jud>`

Now, please correct your answer strictly according to the rules above.

Final output format (exactly three lines):

4) [input]

5) [output]

6) [objective]

Below is the .vrp instance content: `<vrp_text>`

## D.2 SUBTASK 2: CODE GENERATION

### D.2.1 GENERATION AGENT (GA)

**Prompt (Partly):**

Here is the code you generated before (for reference, please improve or extend it if needed): ``

We are working on a VRP problem instance:

Problem description: `<problem_desc>`

Constraints: `<constraints>`

Specific name: `<specific_name>`

Input definition: `<input_def>`

Output definition: `<output_def>`

Optimization objective: `<objective>`

Important rules:

- You are given the raw .vrp file content for context.

- Do not hardcode any instance-specific details such as the number of nodes, vehicle count, or node coordinates.

- Any functions generated must be general-purpose and reusable for any VRP instance.

Task: Generate a Python function named exactly 'read_vrp(path: str)':

- The function must read a .vrp file and parse its content strictly according to the items listed in the Input definition.

- Every element mentioned in the Input definition must be parsed and included. No additional fields may be added, and no required fields may be omitted.

- The function must return a dictionary where:

  • Keys exactly match the names in the Input definition, using underscores "_" instead of spaces.

  • Each value is returned as a list, even if it contains only a single element.

  • A section may end with "-1", "EOF", or the beginning of another section header (lines in all caps ending with "_SECTION") and the function must handle all of these cases.

- The implementation must be general-purpose; do not hardcode any instance-specific details.

Output format requirement:

Return only valid Python code inside a single code block marked as ```python ...   ``` with no extra text, explanations, or comments outside the code block.

If previous code is provided, include it (or an improved version) together with the newly generated function in the final output.

Ensure that all necessary import statements required for the code to run are included at the top of the code block.

**Output (Partly):**

```python
import re
import math
import random
import copy
import argparse
import sys
def read_vrp(path: str):
    with open(path, 'r') as f:
        lines = f.readlines()
    node_coord_dict = {}
    demand_dict = {}
    depot_ids = []
    dimension = None
    capacity = None
    in_node_coord_section = False
    in_demand_section = False
    in_depot_section = False
    for line in lines:
        line = line.strip()
        if not line or line == 'EOF':
            continue
        if line.startswith('DIMENSION'):
            dimension = int(re.search(r':\s*(\d+)', line).group(1))
            continue
        if line.startswith('CAPACITY'):
            capacity = float(re.search(r':\s*([\d.]+)', line).group(1)
                )
            continue
        if line == 'NODE_COORD_SECTION':
            in_node_coord_section = True
            in_demand_section = False
            in_depot_section = False
            continue
        if line == 'DEMAND_SECTION':
```

```python
34              in_node_coord_section = False
35              in_demand_section = True
36              in_depot_section = False
37              continue
38          if line == 'DEPOT_SECTION':
39              in_node_coord_section = False
40              in_demand_section = False
41              in_depot_section = True
42              continue
43          if in_node_coord_section:
44              tokens = line.split()
45              idx = int(tokens[0])
46              x = float(tokens[1])
47              y = float(tokens[2])
48              node_coord_dict[idx] = [x, y]
49          elif in_demand_section:
50              tokens = line.split()
51              idx = int(tokens[0])
52              d = float(tokens[1])
53              demand_dict[idx] = d
54          elif in_depot_section:
55              val = int(line)
56              if val == -1:
57                  continue
58              depot_ids.append(val)
59      if not depot_ids:
60          raise ValueError('Depot information missing in VRP file.')
61      if dimension is None:
62          raise ValueError('DIMENSION missing in VRP file.')
63      if capacity is None:
64          raise ValueError('CAPACITY missing in VRP file.')
65      all_ids = sorted(node_coord_dict.keys())
66      if len(all_ids) != dimension:
67          raise ValueError('Parsed node_coord_dict length does not match
                  DIMENSION.')
68      if len(demand_dict) != dimension:
69          raise ValueError('Parsed demand_dict length does not match
                  DIMENSION.')
70      depot_id = depot_ids[0]
71      if depot_id not in all_ids:
72          raise ValueError('Depot id not present in node_coord_dict.')
73      customer_ids = [nid for nid in all_ids if nid != depot_id]
74      node_id_list = [depot_id] + sorted(customer_ids)
75      node_coordinates = [node_coord_dict[nid] for nid in node_id_list]
76      demands = [demand_dict[nid] for nid in node_id_list]
77      return {
78          "depot": [depot_id],
79          "node_coordinates": node_coordinates,
80          "demands": demands,
81          "vehicle_capacity": [capacity]
82      }
```

Listing 1: Example of route feasibility check

### D.2.2 JUDGMENT AGENT (JA)

**Prompt (Partly):**

You are a strict Python code reviewer and VRP expert.

Here is the generated Python code: ``

We are working on a VRP problem instance:

Problem description: `<problem_desc>`

Constraints: `<constraints>`

Specific name: `<specific_name>`

Input definition: `<input_def>`

Output definition: `<output_def>`

Optimization objective: `<objective>`

Evaluation rules:

Only evaluate the given code snippet. Ignore any other functions or unrelated context.

Check if the code has syntax errors or logical bugs that would prevent execution.

For 'read_vrp':

• Ensure it extracts exactly the elements listed in the Input definition.

• No additional fields may be added, and no required fields may be omitted.

• Verify that each extracted element is explicitly available in the provided .vrp file content.

• Do not assume or fabricate values not present in the file.

• A section may end with -1, EOF, or the beginning of another section header (e.g., lines in all caps ending with _SECTION); it must handle all of these situations.

• Confirm that the function returns all required fields in a dictionary format, and each value must be returned as a list (array); keys must match the Input definition exactly, using underscores _ instead of spaces.

• The function 'read_vrp' must be contained in the code.

Use the provided .vrp file content as the only ground truth for evaluation. Do not invent or assume data that is not present in the instance.

Below is the .vrp instance content: `<vrp_text>`

Assume that the VRP file provides all elements in `<input_def>`.

Assume that this code is only for `<specific_name>`, `<problem_desc>`.

Assume that node IDs preserve the exact order given in the input .vrp file, are unique, and contain no duplicates.

Your task:

If the code is fully correct (no syntax errors, no logical bugs, all constraints satisfied, and fully consistent with the VRP rules), return right1: True and provide a brief explanation.

If the code has any issues (syntax bugs, logical errors, constraint violations, inconsistent naming, wrong input/output handling, or deviations from the specification), return right1: False and explain why. If wrong, you must also provide clear and concrete suggestions for how to fix or improve the code.

Important formatting rule: For easier parsing, the explanation or suggestions must be written in plain text on a single line, without using any line breaks (\n) or additional colons except the ones required in right: and jud:.

Output format must be exactly 2 lines:

right: True/False

jud: explanation and suggestions

**Output**:

> right: False
>
> jud: The code is invalid because the returned dictionary key "node_coordinate" does not match the required Input definition key "node_coordinates"; rename the key to "node_coordinates" to ensure consistency with the specified input definition.

### D.2.3  REVISION AGENT (RA)

**Prompt**:

> Here is the code you generated before: `<prev_code>`
>
> We are working on a VRP problem instance.
>
> Problem description: `<problem_desc>`
>
> Constraints: `<constraints>`
>
> Specific name: `<specific_name>`
>
> Input definition: `<input_def>`
>
> Output definition: `<output_def>`
>
> Optimization objective: `<objective>`
>
> Important rules:
>
> - You are given the raw .vrp file content below for context.
>
> - Do not hardcode any instance-specific details such as number of nodes, vehicle count, or node coordinates.
>
> - The functions you generate must be general-purpose and reusable for any VRP instance.
>
> Below is the .vrp instance content (for context only): `<vrp_text>`
>
> The code you generated previously has the following issues and revised suggestions: `<jud>`
>
> Please correct the code according to the issues above, without changing the number of functions, their names, or their signatures.
>
> Output format requirement: Return only valid Python code inside a single code block: ` ```python ... ``` ` with no extra text, explanations, or comments outside the code block.

The output format is identical to that described in Sec. D.2.1.

### D.3  SUBTASK 3: SOLUTION DERIVATION

### D.3.1  ERROR ANALYSIS AGENTS (EAA)

**Prompt:**

> You are a strict Python code reviewer and VRP expert.
>
> We are working on a VRP problem instance.
>
> Problem description: `<problem_desc>`
>
> Constraints: `<constraints>`
>
> Specific name: `<specific_name>`
>
> Input definition: `<input_def>`

Output definition: `<output_def>`

Optimization objective: `<objective>`

The code was generated using the following rules: `<all rules in generation code>`

Use the provided .vrp file content as the only ground truth for analysis. Do not invent or assume data that is not present in the instance.

Below is the .vrp instance content: `<vrp_text>`

The code execution failed and produced the following error message: `<error_msg>`

Your task: Analyse the error message and explain why the error happened. Provide clear and concrete suggestions for how to fix or improve the code.

Formatting rule: For easier parsing, your explanation and suggestions must be written strictly in plain text on a single line, without using any line breaks (\n) or additional colons except the required jud: prefix.

Output format must be exactly 1 line:

1) jud: explanation and suggestions

**Output:**

jud: The error occurred because the algorithm timed out while generating routes for the CVRP instance, indicating that the current implementations of the initial and insert functions have excessively high computational complexity; to fix this, simplify the construction strategy in initial by reducing unnecessary nested loops, and in insert replace exhaustive cost evaluations with a priority queue or incremental cost update to lower overall runtime.

The JA and RA are identical to those described in Section D.2.2 and Section D.2.3.

## E   THE USE OF LARGE LANGUAGE MODELS

In this study, large language models (LLMs) are not merely tools for polishing text but an integral component of the proposed framework. They serve as autonomous agents within our framework, while the core ideas and the manuscript itself were conceived, prepared, and finalized by the authors.

