# OpenReview forum: "An Agentic Framework with LLMs for Solving Complex Vehicle Routing Problems"
_ICLR.cc/2026/Conference — ICLR 2026 Poster_

### Official Review · Reviewer_KgjX · 2025-10-25

**Soundness:** 3
**Presentation:** 2
**Contribution:** 2
**Rating:** 4
**Confidence:** 4

**Summary:**

This paper proposes a self-contained, LLM-driven framework for solving complex vehicle routing problems (VRPs). The framework is evaluated on 20 VRP variants, demonstrating its effectiveness and generality.

**Strengths:**

1. The idea of designing a self-contained framework powered by LLMs to fully automate the process of solving VRPs is promising.
2. The framework aims to generalize across various VRP formulations, which is an ambitious and potentially impactful direction.

**Weaknesses:**

1. The paper states that its goal is not to surpass state-of-the-art (SOTA) solvers on conventional VRPs, but rather to develop an automated and self-contained framework for handling complex VRPs. However, prior works such as [1, 2] have already demonstrated that LLMs can automatically design heuristics that achieve or even surpass SOTA performance. Compared with those works, the main novelty of this paper appears to lie in increasing automation by accepting natural language task descriptions as input. While this is an interesting step toward fully automated problem solving, it is not entirely clear how significant this improvement is in practice. Given the current framework design, I am not convinced that the heuristics generated by the proposed method would outperform those produced in [1, 2].
2. In Tables 3 and 4, the paper reports results on eight VRPs. However, it is unclear whether the same generated heuristic is used across all problem instances of a given VRP type, or if a separate heuristic is generated for each instance. This distinction is important for assessing the generalization capability of the approach. The authors report performance under different numbers of iterations but do not provide the computational cost or runtime associated with these iterations. Such information is necessary to properly interpret the trade-off between performance and efficiency.
3. In Table 6, the paper presents results on different benchmarks with varying numbers of nodes. It is again unclear whether the same heuristic method is reused or newly generated for each benchmark setting. Clarification on this point is needed to understand how the proposed method scales and generalizes.

# References
[1] ReEvo: Large Language Models as Hyper-Heuristics with Reflective Evolution

[2] Generalizable Heuristic Generation Through Large Language Models with Meta-Optimization

**Questions:**

Check Weakness Section.

---

> ### Author Response · Authors · 2025-11-22
>
> Thank you for highlighting the promise of a fully self-contained LLM-powered framework for automated VRP solving, as well as the potential impact of targeting generalization across diverse VRP formulations. We appreciate your positive assessment and address your questions and concerns below.
>
> #### **W1. Overall Concern**
>
> Thank you for your thoughtful comment. In summary, 1）prior works [1, 2] rely on problem-specific, fully-designed heuristics, whereas our method adopts a general framework that balances generality and performance; 2) our contribution is an agentic framework capable of tackling a broad range of complex VRPs with self-containment, full automation, and high trustworthiness in both code reliability and solution feasibility, representing a practically significant advancement; 3) our framework is compatible with prior works [1, 2] and can be integrated with them to broaden their applicability. In the following, we would like to address this comment in detail from three perspectives.
>
> **a. On the claim that prior works [1, 2] already achieve or surpass SOTA performance on standard VRPs**
>
> Although prior works such as [1, 2] demonstrate that LLMs can evolve heuristics with strong performance on standard VRPs, they face several limitations:
> * Their approaches rely on **manually written base solvers**, with the LLM **modifying only a small component**, typically a single heuristic function.
> * They do not demonstrate the LLM's ability to **handle complex constraints**, nor can they **guarantee the executability of the generated code or the feasibility of the resulting solutions**.
>
> These are fundamental challenges for deploying LLM-based methods in practical settings, as both feasibility and complex-constraint handling are essential for reliable performance in real-world VRPs, and their absence may limit the applicability of such approaches.
>
> In contrast, our approach can handle **60 VRP variants** (including 40 additional variants beyond those in the original paper), covering **several complex formulations that even HGS cannot handle, while maintaining near-100% executability and feasibility.** To further demonstrate this capability, we provide additional results on more complex VRP variants in Table 12. More details of the result are shown in Table 3 in the main paper and Table 11 in the Appendix in the revised paper.

---

> > ### Author Response · Authors · 2025-11-22
> >
> > ##### Table 12. Comparison results on additional 40 standard 100-node benchmark instances.
> >
> > | Gap (%)         | CVRPB         | CVRPBL         | CVRPBTW        | OCVRPB        | CVRPBLTW       | OCVRPBL        | OCVRPBTW      | OCVRPBLTW      | MDCVRP          | MDCVRPL          |
> > | -------------- | ------------- | -------------- | -------------- | ------------- | -------------- | -------------- | ------------- | -------------- | --------------- | ---------------- |
> > | OR-Tools       | 3.85          | 3.43           | 1.60           | 2.32          | 2.77           | 2.36           | 0.76          | 0.77           | 5.27            | 5.24             |
> > | RF-POMO        | 4.47          | 5.70           | 2.96           | 4.82          | 3.28           | 4.83           | 2.34          | 2.34           | 30.10           | 37.19            |
> > | AFL (T=10000) | **2.50**      | 4.26           | 2.38           | 7.54          | 4.48           | **2.32**       | **0.68**      | **0.65**       | 4.21            | **3.87**         |
> > | **Gap (%)**     | **MDCVRPTW**  | **MDCVRPBL**   | **MDCVRPBTW**  | **MDOCVRPB**  | **MDCVRPBLTW** | **MDOCVRPBL**  | **MDCVRPB**   | **MDOCVRP**    | **MDCVRPLTW**   | **MDOCVRPLTW**   |
> > | OR-Tools       | 1.55          | 4.62           | 1.69           | 2.33          | 1.70           | 2.13           | 5.01          | 2.33           | 1.58            | 0.70             |
> > | RF-POMO        | 38.16         | 34.80          | 38.80          | 25.48         | 41.00          | 25.49          | 30.10         | 28.52          | 40.22           | 35.43            |
> > | AFL (T=10000) | **1.45**      | 4.97           | 2.45           | **1.73**      | 3.58           | **1.61**       | **4.63**      | **2.12**       | 4.81            | **0.54**         |
> > | **Gap (%)**     | **MDOCVRPTW** | **MDOCVRPL**   | **MDOCVRPBTW** | **CVRPMB**    | **CVRPMBL**    | **CVRPMBTW**   | **OCVRPMB**   | **CVRPMBLTW**  | **OCVRPMBL**    | **MDOCVRPBLTW**  |
> > | OR-Tools       | 0.74          | 2.33           | 11.79          | 10.27         | 11.90          | 17.48          | 17.54         | 17.76          | 17.54           | 11.79            |
> > | RF-POMO        | 35.43         | 28.55          | 44.70          | 10.90         | 11.12          | 11.94          | 20.31         | 11.89          | 20.32           | 44.69            |
> > | AFL (T=10000) | **0.51**      | **2.01**       | **1.85**       | **4.65**      | **0.94**       | **3.49**       | **1.43**      | **1.59**       | 1.45            | **0.62**         |
> > | **Gap (%)**     | **OCVRPMBTW** | **OCVRPMBLTW** | **MDCVRPMB**   | **MDCVRPMBL** | **MDCVRPMBTW** | **MDOCVRPLTW** | **MDCVRPLTW** | **MDOCVRPMBL** | **MDOCVRPMBTW** | **MDOCVRPMBLTW** |
> > | OR-Tools       | 13.78         | 13.78          | 14.37          | 14.23         | 16.12          | 15.83          | 1.58          | 15.78          | 11.79           | 11.79            |
> > | RF-POMO        | 9.72          | 9.71           | 41.78          | 44.05         | 49.27          | 42.41          | 40.22         | 42.37          | 44.70           | 44.69            |
> > | AFL (T=10000) | **0.62**      | **0.65**       | **4.40**       | **2.89**      | **3.11**       | **1.04**       | 4.81          | **1.17**       | **1.85**            | **0.62**         |

---

> > > ### Author Response · Authors · 2025-11-22
> > >
> > > **b. On the statement that our main novelty is accepting natural-language descriptions as input; and how significant this improvement is in practice**
> > >
> > > **Contribution.** The primary contribution of this paper is not merely natural-language input, but the design of **an agentic framework that delivers self-containment, full automation, and high trustworthiness in both code reliability and solution feasibility.** The coordinated operation of the Generation Agent (GA), Judgment Agent (JA), Revision Agent (RA), and Error Analysis Agent (EAA) is essential for ensuring correctness. Natural-language input is only one component. The architectural contribution lies in enabling LLMs to generate correct, verifiable, and fully executable solvers and feasible solutions for diverse VRP formulations. We additionally provide the comparision between our multi-agent framework with other agentic strategies in Table 8, to demonstrate the effectiveness of our design. We have incorporated these corresponding results into the revised paper, including Sections 4.4 and Table 7.
> > >
> > > **Practical Significance.** Fully automated problem solving aligns closely with real-world needs. In industrial settings, VRP instances typically involve numerous interacting constraints and often fall outside the scope of traditional solvers such as HGS unless substantial manual redesign is performed. In contrast, *AFL can directly handle practical, constraint-rich VRPs (e.g., Electric VRPs) by generating customized algorithms in under ~30 minutes, achieving near-100% code executability and solution feasibility without requiring extensive domain expertise.* Overall, we believe this represents a substantial practical advancement toward truly automated, scalable optimization systems.
> > >
> > > ##### Table 8. Gap Comparison across Different Agentic Strategies.
> > > *Note: "-" indicates cases where the code generated by the prompt strategy is not executable or the resulting routes are infeasible.*
> > > | Gap(%)                                | CVRP     | CVRPL      | CVRPTW    | OCVRP     | CVRPLTW   | OCVRPL    | OCVRPTW   | OCVRPLTW  |
> > > | ------------------------------------- | -------- | --------- | -------- | -------- | -------- | -------- | -------- | -------- |
> > > | Standard Prompt (50 Node Instance)    | 11.76    | -         | 5.11     | 10.60    | 4.58     | 14.59    | -        | -        |
> > > | Self-Refine (50 Node Instance)        | 5.69     | 8.13      | -        | -        | -        | 4.61     | 13.99    | 2.37     |
> > > | Self-Debug (50 Node Instance)         | 11.76    | -         | 5.11     | 10.60    | 4.58     | 14.59    | 43.39    | 12.57    |
> > > | Self-Verification (50 Node Instance)  | 7.61     | -         | 6.80     | 5.71     | -        | -        | 3.54     | -        |
> > > | AFL (50 Node Instance)                 | **5.01** | **7.18**  | **2.87** | **4.30** | **4.34** | **4.61** | **1.24** | **2.12** |
> > > | Standard Prompt (100 Node Instance)   | 12.42    | -         | 8.14     | 13.77    | 7.38     | 23.56    | -        | -        |
> > > | Self-Refine (100 Node Instance)       | 8.83     | 12.12     | -        | -        | -        | 6.79     | 18.90    | 6.02     |
> > > | Self-Debug (100 Node Instance)        | 12.42    | -         | 8.14     | 13.77    | 7.38     | 23.56    | 51.39    | 14.37    |
> > > | Self-Verification (100 Node Instance) | 9.47     | -         | 8.64     | 9.25     | -        | -        | 5.96     | -        |
> > > | AFL (100 Node Instance)                | **6.66** | **10.08** | **4.64** | **6.58** | **7.14** | **6.68** | **2.54** | **4.02** |

---

> > > > ### Author Response · Authors · 2025-11-22
> > > >
> > > > **c. On the concern that AFL’s heuristics may not outperform those from [1, 2]**
> > > >
> > > > As discussed in Appendix A.2, our work and the evolving-heuristics line of research (e.g., ReEvo [1], MOH [2]) belong to **different but complementary directions.** ReEvo and MOH focus on evolving basic heuristics, whereas AFL focuses on developing general frameworks capable of automatically solving a wide range of VRP variants. **Their emphasis is on algorithmic evolution (e.g., improving performance), whereas our emphasis is on fully automated solver generation for diverse VRP variants (e.g., enhancing generality).** Representative works in this direction include SGE (NeurIPS 2024) and DRoC (ICLR 2025), which, similar to AFL, aim to automatically solve different VRP formulations rather than evolve a single heuristic component. We include a table to clearly classify these two research directions.
> > > >
> > > > ##### Table 13. Comparison of Two Main Stems in LLM-Based Code Generation
> > > >
> > > > | Aspect               | Evolving Basic Heuristics           | Developing General Frameworks                       |
> > > > | -------------------- | ----------------------------------- | --------------------------------------------------- |
> > > > | Focus                | Evolve a single heuristic component | Handling complex constraints and solve complex VRPs |
> > > > | Written code         | one single function                 | Multiple functions or full solver                   |
> > > > | Scope            | Simple variants (e.g., TSP, CVRP)   | Complex VRP variants                                |
> > > > | Representative works | ReEvo (NeurIPS 2024), MOH (Arxiv)         | SGE (NeurIPS 2024), DRoC (ICLR 2025)                 |
> > > >
> > > > **These two research directions are orthogonal yet inherently complementary.** Techniques from evolving-basic-heuristics approaches can further refine the algorithms produced by general frameworks like AFL, while AFL can extend evolved heuristics to complex, constraint-rich VRP variants. This mutual complementarity highlights the broader potential of integrating both paradigms in future work. As an initial demonstration (Table 14), our framework can extend ReEvo’s results to additional VRP variants. This remains an early exploration, with substantial room for further improvement. In addition, to directly address the reviewer’s concern, we added a comparison against ReEvo (MoH is not officially published and its code is not public). As shown in Table 11, when compared with ReEvo, **AFL delivers competitive performance while offering substantially broader applicability.**
> > > > ##### Table 14. AFL Application on ReEvo
> > > >
> > > > | Gap(%)            | CVRP  | CVRPL  | CVRPTW | OCVRP  | CVRPLTW | OCVRPL | OCVRPTW | OCVRPLTW |
> > > > | ----------------- | ----- | ----- | ----- | ----- | ------ | ----- | ------ | ------- |
> > > > | 50 Node Instance  | 12.63 | 16.31 | 14.47 | 17.94 | 12.90  | 18.11 | 10.28  | 9.13    |
> > > > | 100 Node Instance | 18.82 | 16.68 | 16.98 | 18.99 | 24.62  | 27.65 | 14.20  | 16.09   |
> > > > ##### Table 11. Gap comparison on benchmark instances
> > > > *Note: "-" indicates cases where the code generated by the prompt strategy is not executable or the resulting routes are infeasible.*
> > > > | Gap (%) | TSPLib (50-200 Node) | TSPLib (200-500 Node) | TSPLib (500-1000 Node) | CVRPLib (100-200 Node) | CVRPLib (200-500 Node) | CVRPLib (500-1000 Node) | CVRPL (50 Node) | CVRPL (100 Node) |
> > > > | ------ | ------------------------- | -------------------------- | --------------------------- | -------------------------- | -------------------------- | --------------------------- | -------------- | --------------- |
> > > > | SGE    | 109.59                    | 287.53                     | 660.36                      | –                          | –                          | –                           | –              | –               |
> > > > | DRoC   | 3.02                      | 3.96                       | 4.22                        | 3.93                       | 8.35                       | –                           | 6.80           | 8.31            |
> > > > | ReEvo  | 5.18                      | 9.13                       | 14.78                       | 8.77                       | 14.88                      | 19.81                       | -              | -               |
> > > > | AFL    | **1.28**                  | **2.68**                   | **2.98**                    | **1.93**                   | **5.20**                   | **6.66**                    | **5.57**       | **6.79**        |

---

> ### Author Response · Authors · 2025-11-22
>
> #### **W2. Heuritic generation and Runtime associated with iterations**
>
> Thanks for your comment. We clarify that AFL generates **one heuristic per VRP variant based on a single representative instance**, which serves as a minimal example preserving all relevant information such as constraints, problem characteristics, and format specifications. The resulting solver (i.e., heuristic) can then be reused across all instances of that variant. For the results reported in Tables 3 and 4, we generate one solver per VRP type rather than per instance. **Once generated, the solver is applied directly to all test instances of that variant.**
>
> The iteration parameter *T* shown refers to the number of improvement steps that the heuristic performs during solution refinement. It does not represent regenerating the heuristic. In addition, the same heuristic is reused by the same VRP variant, so the runtime in the Table 3 and 4 of paper is the computational cost or runtime associated with different iterations. **We additionally provide Table 1.3 below and Figure.4 in the Appendix to explore the trade of performance and efficiency.**
>
> This clarification also highlights the generalization capability of AFL: a single solver generated from one representative instance is able to generalize effectively across all instances of the same VRP variant, which is essential for practical deployment and scalability.
> ##### Table 1.3 The Solution Derivation Time on Single Instance.
>
> | Time (s)     | CVRP | CVRPL | CVRPTW | OCVRP | CVRPLTW | OCVRPL | OCVRPTW | OCVRPLTW |
> | ----------- | ---- | ----- | ------ | ----- | ------- | ------ | ------- | -------- |
> | 50 (T=500)   | 0.08 | 0.19  | 0.45   | 0.20  | 0.43    | 0.26   | 0.50    | 0.82     |
> | 50 (T=2000)  | 0.35 | 0.96  | 2.05   | 0.54  | 1.72    | 1.01   | 2.60    | 3.42     |
> | 50 (T=10000) | 1.92 | 6.53  | 8.81   | 2.02  | 8.39    | 1.43   | 10.56   | 16.39    |
>
> #### **W3. Heuritic Generation with Different Benchmarks**
>
> Thanks for your comment. For the results in Table 6, the same heuristic generated for **a given VRP variant with 50 nodes** is reused across all benchmark settings, including those with different numbers of nodes. No new heuristic is generated for each benchmark. This design aligns with practical industrial needs, where a solver produced once can be repeatedly applied to instances of varying scales. As the results show, the reused heuristic continues to deliver strong performance even when applied to larger and more challenging benchmarks.
>
> We have added responses to W2 and W3 in the revised paper (Lines 370–372), along with our initial explanation in Lines 383–385, to improve clarity.
>
> ----
> Thank you for your insightful comments and the time devoted to reviewing our work. Should any part of the revised manuscript remain unclear or require additional justification, we would be more than happy to elaborate further. We greatly value your feedback and are open to addressing any additional points you may raise.

---

### Official Review · Reviewer_KZcf · 2025-10-27

**Soundness:** 3
**Presentation:** 2
**Contribution:** 2
**Rating:** 6
**Confidence:** 2

**Summary:**

This paper proposes a novel framework named AFL (Agentic Framework with LLMs), which aims to fully automate the solving of complex Vehicle Routing Problems (VRPs) using Large Language Models (LLMs). It decomposes the complex solving pipeline into three subtasks: problem description, code generation, and solution derivation. It also introduces four specialized LLM Agents—Generation, Critique, Revision, and Error Analysis—to collaborate, significantly improving the reliability of the generated code and the feasibility of the final solution.

**Strengths:**

1. AFL achieves full end-to-end automation, from the VRP instance file to the final solution, without requiring human intervention during execution or relying on external solvers or predefined code libraries.
2. Decomposing the complex task and introducing multiple collaborating LLM Agents with specialized roles (Generation, Critique, Revision, Error Analysis) is an effective strategy for enhancing the reliability of LLMs in complex programming and reasoning tasks. This approach is generalizable and could potentially be applied to other optimization problems.
3. The experiments not only include standard VRPs but also specifically test complex variants more common in real-world scenarios, such as Electric VRPs (EVRPs) with multiple combined constraints, demonstrating the framework's effectiveness and generality in handling complex constraints.

**Weaknesses:**

1. It is highly dependent on the powerful code generation, comprehension, and reasoning capabilities of the LLM used (GPT-4.1). How the framework performs on less capable, open-source LLMs, and its sensitivity to different LLMs, warrants further investigation.
2. Excessive time consumption is a potential issue, as shown in Table 3. How do the authors view this trade-off of sacrificing time for accuracy? In scenarios requiring rapid feedback, is this algorithm still viable?

Since I am not an expert in this field,  I don't know if the novelty of this kind of prompt engineering paper is sufficient.

**Questions:**

see in weakness

---

> ### Author Response · Authors · 2025-11-22
>
> We are grateful for your thoughtful and encouraging assessment of AFL, particularly regarding its end-to-end automation, the effectiveness of the multi-agent design, and its strong applicability to real-world VRP variants. Your comments and the directions suggested for further exploration are highly meaningful and constructive. Below, we provide detailed responses to the your questions and suggestions.
>
> #### **W1. How the framework performs on less capable, open-source LLMs, and its sensitivity to different LLMs, warrants further investigation.**
>
> We value your insightful suggestion, which is highly relevant for assessing the robustness of AFL across different LLM backbones. Following this recommendation, we conducted additional experiments using **Claude-Sonnet-4-20250514** (as Sonnet 3 API has been officially discontinued) and **GPT-4o**. The results, summarized in Tables 6 and 7, show the performance and reliability of AFL under different LLM configurations.
>
> Regarding the impact of varying LLM capabilities, our additional experiments (Tables 6 and 7) show that the underlying **model capacity does influence the solution quality** produced by AFL. Stronger LLMs tend to yield higher-fidelity implementations of the required heuristics, leading to better objective values. Overall, we observe an ordering of  Claude-Sonnet-4 > GPT-4.1  > GPT-4o in terms of overall performance, including code generation and understanding of human intent. Although different LLMs exhibit varying capabilities and thus achieve different performance levels, our framework can enhance the implementation of the required heuristics within a single LLM, as shown in Table 8. The results demonstrate that our method outperforms both standard prompting and alternative prompt strategies, indicating that our framework effectively strengthens an LLM’s ability to implement the targeted heuristics.
>
> In contrast, **code reliability is largely unaffected by the choice of LLM**. Across all tested models, AFL consistently produces executable code and feasible solutions. This demonstrates that the multi-agent design, effectively stabilizes the pipeline even when using weaker LLMs, validating the robustness and generality of our framework.
>
> These additional results and insights have been incorporated into Section 4.4 and Table 7 of the main paper, as well as Section C.10 and Section 16 of the Appendix in the revised version.
>
> ##### Table 6. AFL Performance on Claude-Sonnet-4-20250514.
> *Note: **Bold** numbers indicate results that surpass the GPT-4.1 baseline used in the main paper.*
> |                              | CVRP (obj.)       | Gap (%)     | Time (m)     | CVRPL (obj.)      | Gap (%)     | Time (m)     | CVRPTW (obj.)      | Gap (%)     | Time (m)     | OCVRP (obj.)        | Gap (%)     | Time (m)     |
> | ---------------------------- | ---------------- | ---------- | ----------- | --------------- | ---------- | ----------- | ---------------- | ---------- | ----------- | ----------------- | ---------- | ----------- |
> | 50 Node Instance (*T*=10000)  | **10.54**        | **1.64**   | 2.60        | **10.91**       | **3.02**   | 6.85        | 16.28            | 1.56       | 8.50        | **6.60**          | **1.38**   | 2.95        |
> | 100 Node Instance (*T*=10000) | **15.90**        | **1.79**   | 5.12        | **16.44**       | **4.25**   | 25.79       | 26.19            | 3.03       | 33.78       | **9.94**          | **2.26**   | 7.22        |
> |                              | **CVRPLTW (obj.)** | **Gap (%)** | **Time (m)** | **OCVRPL (obj.)** | **Gap (%)** | **Time (m)** | **OCVRPTW (obj.)** | **Gap (%)** | **Time (m)** | **OCVRPLTW (obj.)** | **Gap (%)** | **Time (m)** |
> | 50 Node Instance (*T*=10000)  | **16.55**        | **1.16**   | 7.32        | **6.59**        | **1.23**   | 7.43        | 10.63            | 1.14       | 14.89       | 10.64             | 1.24       | 12.13       |
> | 100 Node Instance (*T*=10000) | **26.50**        | **2.87**   | 30.94       | **9.91**        | **1.95**   | 16.59       | 17.24            | 1.83       | 35.95       | 17.26             | 1.95       | 62.24       |

---

> > ### Author Response · Authors · 2025-11-22
> >
> > ##### Table 7. AFL Performance on GPT-4o.
> >
> > |                              | CVRP (obj.)        | Gap (%)     | Time (m)     | CVRPL (obj.)      | Gap (%)     | Time (m)     | CVRPTW (obj.)      | Gap (%)     | Time (m)     | OCVRP (obj.)        | Gap (%)     | Time (m)     |
> > | ---------------------------- | ----------------- | ---------- | ----------- | ---------------- | ---------- | ----------- | ----------------- | ---------- | ----------- | ------------------ | ---------- | ----------- |
> > | 50 Node Instance (*T*=10000)  | 10.64             | 2.60       | 2.21        | 11.01            | 3.97       | 13.06m      | 16.45             | 2.62       | 12.62       | 6.87               | 5.53       | 3.07        |
> > | 100 Node Instance (*T*=10000) | 16.05             | 2.75       | 4.96        | 16.64            | 5.52       | 34.25       | 26.88             | 5.74       | 25.44       | 10.36              | 6.47       | 7.38        |
> > |                              | **CVRPLTW (obj.)** | **Gap (%)** | **Time (m)** | **OCVRPL (obj.)** | **Gap (%)** | **Time (m)** | **OCVRPTW (obj.)** | **Gap (%)** | **Time (m)** | **OCVRPLTW (obj.)** | **Gap (%)** | **Time (m)** |
> > | 50 Node Instance (*T*=10000)  | 16.79             | 2.63       | 8.54        | 6.81             | 4.61       | 7.13        | 10.91             | 3.81       | 11.89       | 10.88              | 3.52       | 11.29       |
> > | 100 Node Instance (*T*=10000) | 26.81             | 4.08       | 35.48       | 10.33            | 6.28       | 13.42       | 17.68             | 4.43       | 27.39       | 17.66              | 4.31       | 58.86       |
> >
> > ##### Table 8. Gap Comparison across Different Prompting Strategies.
> > *Note: "-" indicates cases where the code generated by the prompt strategy is not executable or the resulting routes are infeasible.*
> > | Gap (%)                                | CVRP     | CVRPL      | CVRPTW    | OCVRP     | CVRPLTW   | OCVRPL    | OCVRPTW   | OCVRPLTW  |
> > | ------------------------------------- | -------- | --------- | -------- | -------- | -------- | -------- | -------- | -------- |
> > | Standard Prompt (50 Node Instance)    | 11.76    | -         | 5.11     | 10.60    | 4.58     | 14.59    | -        | -        |
> > | Self-Refine (50 Node Instance)        | 5.69     | 8.13      | -        | -        | -        | 4.61     | 13.99    | 2.37     |
> > | Self-Debug (50 Node Instance)         | 11.76    | -         | 5.11     | 10.60    | 4.58     | 14.59    | 43.39    | 12.57    |
> > | Self-Verification (50 Node Instance)  | 7.61     | -         | 6.80     | 5.71     | -        | -        | 3.54     | -        |
> > | AFL (50 Node Instance)                 | **5.01** | **7.18**  | **2.87** | **4.30** | **4.34** | **4.61** | **1.24** | **2.12** |
> > | Standard Prompt (100 Node Instance)   | 12.42    | -         | 8.14     | 13.77    | 7.38     | 23.56    | -        | -        |
> > | Self-Refine (100 Node Instance)       | 8.83     | 12.12     | -        | -        | -        | 6.79     | 18.90    | 6.02     |
> > | Self-Debug (100 Node Instance)        | 12.42    | -         | 8.14     | 13.77    | 7.38     | 23.56    | 51.39    | 14.37    |
> > | Self-Verification (100 Node Instance) | 9.47     | -         | 8.64     | 9.25     | -        | -        | 5.96     | -        |
> > | AFL (100 Node Instance)                | **6.66** | **10.08** | **4.64** | **6.58** | **7.14** | **6.68** | **2.54** | **4.02** |

---

> ### Author Response · Authors · 2025-11-22
>
> #### **W2. Trade-off of Sacrificing Time and Accuracy**
>
> Thanks for your comment. For VRP problems, the usefulness of a solution fundamentally depends on its feasibility. As shown in Table 5 of the main paper, AFL consistently **achieves near-100% feasibility across all variants,** ensuring that the generated solutions remain valid even in challenging settings. In scenarios requiring rapid feedback, a solution produced within a short time (i.e., with a small $T$) may already be sufficient, as it is feasible even if its objective value is not fully optimized.
>
> Importantly, **the code generated earlier by AFL can be reused** directly in such real-time environments, as it is produced for the entire problem variant based on a single representative instance. This enables fast solution derivation without the need to rerun the multi-agent generation pipeline. (We provide single instance's runtime at Table 1.3.) Moreover, when new or more complex problem structures arise, **AFL is able to produce customized algorithms significantly faster than manual development (< ~30 mins),** offering a practical advantage in time-sensitive or evolving operational contexts. **We additionally provide Figure.4 in the Appendix to explore the trade of performance and efficiency.**
>
> Further improving the model’s efficiency is also part of our future plan. We intend to enhance efficiency by exploring multi-objective evolutionary strategies to evolve the generated algorithms, jointly optimizing both performance and computational efficiency.
>
> ##### Table 1.3 The Runtime on Single Instance.
>
> | Time(s)     | CVRP | CVRPL | CVRPTW | OCVRP | CVRPLTW | OCVRPL | OCVRPTW | OCVRPLTW |
> | ----------- | ---- | ----- | ------ | ----- | ------- | ------ | ------- | -------- |
> | 50(T=500)   | 0.08 | 0.19  | 0.45   | 0.20  | 0.43    | 0.26   | 0.50    | 0.82     |
> | 50(T=2000)  | 0.35 | 0.96  | 2.05   | 0.54  | 1.72    | 1.01   | 2.60    | 3.42     |
> | 50(T=10000) | 1.92 | 6.53  | 8.81   | 2.02  | 8.39    | 1.43   | 10.56   | 16.39    |
> #### **General Comment: Regarding Novelty**
>
> Recent work in the VRP domain has produced many prompt-engineering–based methods to address the CO challenges, and several of these have been published at top venues (as listed in Table 10). However, despite their contributions, all existing approaches share a fundamental limitation: **they cannot reliably ensure that the generated code is executable or that the produced solutions satisfy VRP feasibility constraints.** This bottleneck has significantly limited the practical adoption and further development of LLM-based VRP solvers.
>
> Our work provides a substantial step forward by introducing an agentic framework that explicitly judges and repairs LLM outputs.  This design enables AFL to **guarantee code executability and solution feasibility,** capabilities not achieved by prior prompt-engineering methods, while still delivering strong performance across both standard and complex VRP variants, as shown in Table 11. The results demonstrate that AFL outperforms existing approaches in solution quality. Additionally, our model can handle more complex variants of the VRP (**up to 60 variants**, including 40 newly extended ones shown in Table 12), whereas previous works are limited to only a few simpler variants. *The ability to transform raw VRP instances into fully feasible solutions in an automated, self-contained manner significantly reduces the domain expertise required to tackle complex optimization problems, and represents a novel and practical contribution to both the VRP and LLM-for-optimization communities.*

---

> > ### Author Response · Authors · 2025-11-22
> >
> > ##### Table 10. Published Prompt-Engineering Based Studies in the VRP Domain
> >
> > | Study                                                        | Venue        | Method Summary                                               |
> > | ------------------------------------------------------------ | ------------ | ------------------------------------------------------------ |
> > | Evolution of heuristics: Towards efficient automatic algorithm design using large language model | ICML 2024     | Design a population-based frame work with fixed templates for heuristic evolution |
> > | ReEvo: Large language models as hyper-heuristics with reflective evolution. | Neurips 2024 | Combine evolutionary search with LLM reflections to provide verbal feedback and enhance search efficiency based on EOH |
> > | Self-guiding exploration for combinatorial problems          | NeurIPS 2024 | Using COT to eliminate the need for predefined modules or solvers by directly generating code end-to-end |
> > | DRoC: Elevating large language models for complex vehicle routing via decomposed retrieva lof constraints. | ICLR 2025     | Design Retrive method to guide LLM produceing code that invokes OR-Tools to solve complex VRPs. |
> >
> > ##### Table 11. Gap comparison on benchmark instances
> > *Note: "-" indicates cases where the code generated by the prompt strategy is not executable or the resulting routes are infeasible.*
> > | Gap (%) | TSPLib (50-200 Node) | TSPLib (200-500 Node) | TSPLib (500-1000 Node) | CVRPLib (100-200 Node) | CVRPLib (200-500 Node) | CVRPLib (500-1000 Node) | CVRPL (50 Node) | CVRPL (100 Node) |
> > | ------ | ------------------------- | -------------------------- | --------------------------- | -------------------------- | -------------------------- | --------------------------- | -------------- | --------------- |
> > | SGE    | 109.59                    | 287.53                     | 660.36                      | –                          | –                          | –                           | –              | –               |
> > | DRoC   | 3.02                      | 3.96                       | 4.22                        | 3.93                       | 8.35                       | –                           | 6.80           | 8.31            |
> > | ReEvo  | 5.18                      | 9.13                       | 14.78                       | 8.77                       | 14.88                      | 19.81                       | -              | -               |
> > | AFL    | **1.28**                  | **2.68**                   | **2.98**                    | **1.93**                   | **5.20**                   | **6.66**                    | **5.57**       | **6.79**        |

---

> > > ### Author Response · Authors · 2025-11-22
> > >
> > > ##### Table 12. Comparison results on additional 40 standard 100-node benchmark instances.
> > >
> > > | Gap (%)         | CVRPB         | CVRPBL         | CVRPBTW        | OCVRPB        | CVRPBLTW       | OCVRPBL        | OCVRPBTW      | OCVRPBLTW      | MDCVRP          | MDCVRPL          |
> > > | -------------- | ------------- | -------------- | -------------- | ------------- | -------------- | -------------- | ------------- | -------------- | --------------- | ---------------- |
> > > | OR-Tools       | 3.85          | 3.43           | 1.60           | 2.32          | 2.77           | 2.36           | 0.76          | 0.77           | 5.27            | 5.24             |
> > > | RF-POMO        | 4.47          | 5.70           | 2.96           | 4.82          | 3.28           | 4.83           | 2.34          | 2.34           | 30.10           | 37.19            |
> > > | AFL (T =10000) | **2.50**      | 4.26           | 2.38           | 7.54          | 4.48           | **2.32**       | **0.68**      | **0.65**       | 4.21            | **3.87**         |
> > > | **Gap (%)**     | **MDCVRPTW**  | **MDCVRPBL**   | **MDCVRPBTW**  | **MDOCVRPB**  | **MDCVRPBLTW** | **MDOCVRPBL**  | **MDCVRPB**   | **MDOCVRP**    | **MDCVRPLTW**   | **MDOCVRPLTW**   |
> > > | OR-Tools       | 1.55          | 4.62           | 1.69           | 2.33          | 1.70           | 2.13           | 5.01          | 2.33           | 1.58            | 0.70             |
> > > | RF-POMO        | 38.16         | 34.80          | 38.80          | 25.48         | 41.00          | 25.49          | 30.10         | 28.52          | 40.22           | 35.43            |
> > > | AFL (T =10000) | **1.45**      | 4.97           | 2.45           | **1.73**      | 3.58           | **1.61**       | **4.63**      | **2.12**       | 4.81            | **0.54**         |
> > > | **Gap (%)**     | **MDOCVRPTW** | **MDOCVRPL**   | **MDOCVRPBTW** | **CVRPMB**    | **CVRPMBL**    | **CVRPMBTW**   | **OCVRPMB**   | **CVRPMBLTW**  | **OCVRPMBL**    | **MDOCVRPBLTW**  |
> > > | OR-Tools       | 0.74          | 2.33           | 11.79          | 10.27         | 11.90          | 17.48          | 17.54         | 17.76          | 17.54           | 11.79            |
> > > | RF-POMO        | 35.43         | 28.55          | 44.70          | 10.90         | 11.12          | 11.94          | 20.31         | 11.89          | 20.32           | 44.69            |
> > > | AFL (T =10000) | **0.51**      | **2.01**       | **1.85**       | **4.65**      | **0.94**       | **3.49**       | **1.43**      | **1.59**       | 1.45            | **0.62**         |
> > > | **Gap (%)**     | **OCVRPMBTW** | **OCVRPMBLTW** | **MDCVRPMB**   | **MDCVRPMBL** | **MDCVRPMBTW** | **MDOCVRPLTW** | **MDCVRPLTW** | **MDOCVRPMBL** | **MDOCVRPMBTW** | **MDOCVRPMBLTW** |
> > > | OR-Tools       | 13.78         | 13.78          | 14.37          | 14.23         | 16.12          | 15.83          | 1.58          | 15.78          | 11.79           | 11.79            |
> > > | RF-POMO        | 9.72          | 9.71           | 41.78          | 44.05         | 49.27          | 42.41          | 40.22         | 42.37          | 44.70           | 44.69            |
> > > | AFL (T =10000) | **0.62**      | **0.65**       | **4.40**       | **2.89**      | **3.11**       | **1.04**       | 4.81          | **1.17**       | **1.85**           | **0.62**         |
> > > ----
> > > We thank the reviewer for these helpful comments. If any parts of our revised explanations are still unclear or insufficiently detailed, we would be very happy to provide additional clarification or further refine the manuscript. Please feel free to point out any specific aspects you would like us to elaborate on.

---

> > > > ### Comment · Reviewer_KZcf · 2025-11-26
> > > >
> > > > I appreciate the authors' effort. My main concern has been resolved, and I will maintain my positive rating.

---

> > > > > ### Author Response · Authors · 2025-11-26
> > > > >
> > > > > Thank you very much for your follow-up. We are glad to hear that your main concern has been resolved, and we sincerely appreciate your positive rating and support!

---

### Official Review · Reviewer_9utD · 2025-10-29

**Soundness:** 2
**Presentation:** 3
**Contribution:** 3
**Rating:** 6
**Confidence:** 2

**Summary:**

This manuscript proposes an Agentic Framework with LLMs (AFL) for solving complex VRPs with full automation and self-containment. AFL decomposes the VRP-solving pipeline into three subtasks and employs four specialized agents to ensure cross-functional consistency and logical soundness. The framework directly extracts domain knowledge from VRPLib-format raw inputs, generates executable code without handcrafted modules, and derives feasible solutions. Extensive experiments on 20 VRP variants demonstrate that the performance of AFL is comparable to state-of-the-art solvers and significant outperformance over existing LLM-based baselines.

**Strengths:**

AFL achieves end-to-end automation from raw VRP instances to solutions by decomposing the pipeline into manageable subtasks and leveraging specialized agents. This eliminates human intervention and external solver dependencies, addressing key limitations of prior LLM-based approaches.

AFL is validated on 20 diverse VRP variants, including standard benchmarks (CVRP, VRPTW), practical electric VRPs (ECVRP, ECVRPTW), and other combinatorial problems (ATSP, ACVRP, SOP). It consistently delivers competitive performance, demonstrating broad applicability to complex real-world routing scenarios.

**Weaknesses:**

While AFL achieves competitive performance, it still lags behind state-of-the-art solvers (e.g., HGS-PyVRP)

The performance of AFL heavily relies on the LLM's ability to generate accurate problem descriptions and code. Potential biases or inaccuracies in LLM outputs may propagate through the pipeline, affecting the final solution quality.

**Questions:**

Do you plan to integrate more advanced heuristic strategies (e.g., evolutionary search) to narrow the performance gap with specialized SOTA solvers? If so, how to ensure the framework remains automated and generalizable?

Can AFL perform on extremely large VRP instances (e.g., 5000+ customers)? If not, are there any constraints? if so, are there any  optimization strategies to improve computational efficiency?

Have you tested AFL with different LLMs (e.g., GPT-4o, Claude 3)? How do variations in LLM capabilities affect the framework's code reliability and solution quality?

Can AFL be extended to handle dynamic VRPs (e.g., real-time customer additions, traffic condition changes)? If yes, what modifications are needed to the current agentic pipeline?

---

> ### Author Response · Authors · 2025-11-22
>
> We appreciate your insightful observations regarding the contributions of AFL, particularly the system’s ability to deliver full end-to-end automation and its broad applicability across diverse VRP variants. The comments effectively capture the core strengths we aimed to demonstrate and have helped further refine the presentation of our work. Below, we provide detailed responses to your comments and suggestions.
>
> #### **W1 & Q1. Lag behind HGS-PyVRP, Integration of Advanced Heuristics**
>
> We thank you for the attention to the statements in our main text:
> > Lines 378–380: *"our objective is not to surpass SOTA solvers on conventional VRPs, which reflect decades of expert effort, but to develop a fully automated and self-contained framework for tackling complex VRPs."*
>
> > Lines 533–535: *"The main limitation lies in performance, which do not yet surpass state-of-the-art solvers specifically designed for well-studied problems such as CVRP, a trade-off we consider acceptable given AFL’s automation and generality."*
>
> While AFL may not yet match HGS-PyVRP on classic benchmarks, it is designed for practical applicability, where problem structures often deviate from widely used default rules in classical VRP datasets (e.g., fixed depots, single visit per customer, and fixed vehicle capacity). In particular, AFL can naturally handle practical scenarios such as electric vehicle routing problems, which can not be solved by HGS-PyVRP. The corresponding results are reported in Section 4.2 of the main paper. As this work represents an early exploration of using agentic LLMs to tackle complex VRPs, we anticipate that future research will continue to improve performance, for example, by incorporating more advanced heuristic strategies, as the reviewer suggested.
>
> Specifically, we plan to incorporate more advanced heuristic strategies, such as evolutionary or population-based search, as part of our future work. AFL is naturally flexible in this regard: integrating a different heuristic does **not require modifying the framework** itself, but **only adjusting the algorithm code requirements** provided during the code-generation stage. For example, adopting an HGS would simply require specifying components such as genetic operators, mutation, and local search within the algorithm code requirements, while the multi-agent workflow, verification mechanisms, and overall automation pipeline would remain unchanged or minimal modification. At present, this direction is still under exploration, but we view it as a promising extension and intend to investigate it in future iterations of AFL.
> #### **W2 & Q3. The performance of AFL relies on the LLM's ability to generate accurate problem descriptions and code. How about different LLMs?**
>
> We value your insightful suggestion, which is highly relevant for assessing the robustness of AFL across different LLM backbones. Following this recommendation, we conducted additional experiments using **Claude-Sonnet-4-20250514** (as Sonnet 3 API has been officially discontinued) and **GPT-4o**. The results, summarized in Tables 6 and 7, show the performance and reliability of AFL under different LLM configurations.
>
> Regarding the impact of varying LLM capabilities, our additional experiments (Tables 6 and 7) show that the underlying **model capacity does influence the solution quality** produced by AFL. Stronger LLMs tend to yield higher-fidelity implementations of the required heuristics, leading to better objective values. Overall, we observe an ordering of  Claude-Sonnet-4 > GPT-4.1  > GPT-4o in terms of overall performance, including code generation and understanding of human intent. Although different LLMs exhibit varying capabilities and thus achieve different performance levels, our framework can enhance the implementation of the required heuristics within a single LLM, as shown in Table 8. The results demonstrate that our method outperforms both standard prompting and alternative prompt strategies, indicating that our framework effectively strengthens an LLM’s ability to implement the targeted heuristics.
>
> In contrast, **code reliability is largely unaffected by the choice of LLM**. Across all tested models, AFL consistently produces executable code and feasible solutions. This demonstrates that the multi-agent design, effectively stabilizes the pipeline even when using weaker LLMs, validating the robustness and generality of our framework.
>
> These additional results and insights have been incorporated into Section 4.4 and Table 7 of the main paper, as well as Section C.10 and Section 16 of the Appendix in the revised version.

---

> > ### Author Response · Authors · 2025-11-22
> >
> > ##### Table 6. AFL Performance on Claude-Sonnet-4-20250514.
> > *Note: **Bold** numbers indicate results that surpass the GPT-4.1 baseline used in the main paper.*
> > |  | CVRP (obj.)       | Gap (%)     | Time (m)     | CVRPL (obj.)      | Gap (%)     | Time (m)     | CVRPTW (obj.)      | Gap (%)     | Time (m)     | OCVRP (obj.)        | Gap (%)     | Time (m)     |
> > | -- | ---- | -- | ---- | --- | --- | -- | ------ | --- | ----------- | ----------------- | ---------- | ----------- |
> > | 50 Node Instance (*T*=10000)  | **10.54**        | **1.64**   | 2.60        | **10.91**       | **3.02**   | 6.85        | 16.28            | 1.56       | 8.50        | **6.60**          | **1.38**   | 2.95        |
> > | 100 Node Instance (*T*=10000) | **15.90**        | **1.79**   | 5.12        | **16.44**       | **4.25**   | 25.79       | 26.19            | 3.03       | 33.78       | **9.94**          | **2.26**   | 7.22        |
> > |                              | **CVRPLTW (obj.)** | **Gap (%)** | **Time (m)** | **OCVRPL (obj.)** | **Gap (%)** | **Time (m)** | **OCVRPTW (obj.)** | **Gap (%)** | **Time (m)** | **OCVRPLTW (obj.)** | **Gap (%)** | **Time (m)** |
> > | 50 Node Instance (*T*=10000)  | **16.55**        | **1.16**   | 7.32        | **6.59**        | **1.23**   | 7.43        | 10.63            | 1.14       | 14.89       | 10.64             | 1.24       | 12.13       |
> > | 100 Node Instance (*T*=10000) | **26.50**        | **2.87**   | 30.94       | **9.91**        | **1.95**   | 16.59       | 17.24            | 1.83       | 35.95       | 17.26             | 1.95       | 62.24       |
> >
> > ##### Table 7. AFL Performance on GPT-4o.
> >
> > |                              | CVRP (obj.)        | Gap (%)     | Time (m)     | CVRPL (obj.)      | Gap (%)     | Time (m)     | CVRPTW (obj.)      | Gap (%)     | Time (m)     | OCVRP (obj.)        | Gap (%)     | Time (m)     |
> > | ---------------------------- | ----------------- | ---------- | ----------- | ---------------- | ---------- | ----------- | ----------------- | ---------- | ----------- | ------------------ | ---------- | ----------- |
> > | 50 Node Instance (*T*=10000)  | 10.64             | 2.60       | 2.21        | 11.01            | 3.97       | 13.06m      | 16.45             | 2.62       | 12.62       | 6.87               | 5.53       | 3.07        |
> > | 100 Node Instance (*T*=10000) | 16.05             | 2.75       | 4.96        | 16.64            | 5.52       | 34.25       | 26.88             | 5.74       | 25.44       | 10.36              | 6.47       | 7.38        |
> > |                              | **CVRPLTW (obj.)** | **Gap (%)** | **Time (m)** | **OCVRPL (obj.)** | **Gap (%)** | **Time (m)** | **OCVRPTW (obj.)** | **Gap (%)** | **Time (m)** | **OCVRPLTW (obj.)** | **Gap (%)** | **Time (m)** |
> > | 50 Node Instance (*T*=10000)  | 16.79             | 2.63       | 8.54        | 6.81             | 4.61       | 7.13        | 10.91             | 3.81       | 11.89       | 10.88              | 3.52       | 11.29       |
> > | 100 Node Instance (*T*=10000) | 26.81             | 4.08       | 35.48       | 10.33            | 6.28       | 13.42       | 17.68             | 4.43       | 27.39       | 17.66              | 4.31       | 58.86       |
> >
> > ##### Table 8. Gap Comparison across Different Prompting Strategies.
> > *Note: "-" indicates cases where the code generated by the prompt strategy is not executable or the resulting routes are infeasible.*
> > | Gap (%)                                | CVRP     | CVRPL      | CVRPTW    | OCVRP     | CVRPLTW   | OCVRPL    | OCVRPTW   | OCVRPLTW  |
> > | ------------------------------------- | -------- | --------- | -------- | -------- | -------- | -------- | -------- | -------- |
> > | Standard Prompt (50 Node Instance)    | 11.76    | -         | 5.11     | 10.60    | 4.58     | 14.59    | -        | -        |
> > | Self-Refine (50 Node Instance)        | 5.69     | 8.13      | -        | -        | -        | 4.61     | 13.99    | 2.37     |
> > | Self-Debug (50 Node Instance)         | 11.76    | -         | 5.11     | 10.60    | 4.58     | 14.59    | 43.39    | 12.57    |
> > | Self-Verification (50 Node Instance)  | 7.61     | -         | 6.80     | 5.71     | -        | -        | 3.54     | -        |
> > | AFL (50 Node Instance)                 | **5.01** | **7.18**  | **2.87** | **4.30** | **4.34** | **4.61** | **1.24** | **2.12** |
> > | Standard Prompt (100 Node Instance)   | 12.42    | -         | 8.14     | 13.77    | 7.38     | 23.56    | -        | -        |
> > | Self-Refine (100 Node Instance)       | 8.83     | 12.12     | -        | -        | -        | 6.79     | 18.90    | 6.02     |
> > | Self-Debug (100 Node Instance)        | 12.42    | -         | 8.14     | 13.77    | 7.38     | 23.56    | 51.39    | 14.37    |
> > | Self-Verification (100 Node Instance) | 9.47     | -         | 8.64     | 9.25     | -        | -        | 5.96     | -        |
> > | AFL (100 Node Instance)                | **6.66** | **10.08** | **4.64** | **6.58** | **7.14** | **6.68** | **2.54** | **4.02** |

---

> > > ### Author Response · Authors · 2025-11-22
> > >
> > > #### **Q2. Large VRP Instance**
> > >
> > > AFL is capable of handling extremely large VRP instances. Our current approach generates a general heuristic from a small instance (e.g., a 50-node instance obtained by trimming the target instance’s data, which serves as a minimal example preserving all relevant information such as constraints, problem characteristics, and format specifications). The generated solver is then applied directly to large-scale cases. As shown in Table 9, **AFL achieves competitive performance on CVRPLib-XXL with more than 16,000 customers**.
> > >
> > > **How to improve computational efficiency?** During code generation, we enforce a practical constraint: the algorithm must complete the solution derivation within 10 seconds for the given instance. If this limit is exceeded, the system triggers a timeout, then the Revision Agent (RA) and Error Analysis Agent (EAA) are trigered to refine the algorithm for improved efficiency. Looking ahead, we will further enhance efficiency by exploring multi-objective evolutionary strategies to evolve the generated algorithms, jointly optimizing both performance and computational efficiency.
> > >
> > > We are grateful for this excellent suggestion, which allows us to more clearly discuss AFL’s ability to handle large-scale VRP instances. We have added the corresponding explanation and experimental results to the revised manuscript, including the Section C.12 and Table 18 in Appendix in revised paper.
> > >
> > > ##### Table 9. Performance of AFL on large-scale instances from CVRPLib-XXL.
> > >
> > > | Gap (%) | **L1 (3k)** | **L2 (4k)** | **A1 (6k)** | **A2 (7k)** | **G1 (10k)** | **G2 (11k)** | **B1 (15k)** | **B2 (16k)** |
> > > | ------ | ---------- | ---------- | ---------- | ---------- | ----------- | ----------- | ----------- | ----------- |
> > > | POMO   | 75.30      | 78.16      | 112.27     | 159.22     | _           | _           | _           | _           |
> > > | LEHD   | 14.04      | 26.30      | 18.90      | 26.40      | 27.23       | 38.45       | 35.94       | 40.76       |
> > > | Our    | **8.12**   | **13.34**  | **8.60**   | **13.62**  | **8.02**    | **14.85**   | **7.01**    | **16.56**   |
> > >
> > > #### **Q4. Dynamic VRPs**
> > >
> > > This is an excellent and practically meaningful direction, and we greatly appreciate the reviewer’s suggestion. We will leave this for future work. AFL can indeed be extended to handle dynamic VRPs. Building on the current multi-agent architecture and multi-subtask pipeline, **an additional subtask and a corresponding agent can be introduced to process dynamic events** such as real-time customer arrivals or changes in travel conditions. When a dynamic update occurs, the new agent would analyze the current solution state and, based on the previously generated solver, replan the remaining route accordingly. This modification preserves the automation and generalizability of the framework while enabling AFL to adapt to real-time changes.
> > >
> > > ----
> > > We appreciate the reviewer’s careful reading and constructive suggestions. Should there remain any ambiguities or areas that could benefit from further clarification, we are more than willing to address them and make additional revisions. We welcome any further questions or guidance you may have.

---

> > > > ### Comment · Reviewer_9utD · 2025-11-25
> > > >
> > > > Thanks for your clarification and additional experiments. I see your efforts on the automatic pipeline, but the performance still does not surpass existing solvers for all tasks. I will keep my score as it is already positive and raise my confidence.

---

> > > > > ### Author Response · Authors · 2025-11-26
> > > > >
> > > > > We sincerely appreciate your response and fully understand your perspective. Improving the solution quality while maintaining a fully automatic and feasible pipeline is indeed one of our main goals. At the same time, we would like to highlight that our current framework represents a meaningful step forward for LLM-based VRP research.
> > > > >
> > > > > * Compared with previous LLM-based methods, our agentic framework solves a critical bottleneck in this area: the lack of reliable code executability and solution feasibility.
> > > > > * In terms of performance, our method achieves clear improvements over existing LLM-based approaches such as SGE (NeurIPS 2024), DRoC (ICLR 2025) and ReEvo (NeurIPS 2024).
> > > > > * Regarding comparisons with traditional solvers, we fully acknowledge that these methods benefit from **decades of algorithmic refinement**. Yet, given the high level of automation and the ability to handle complex and realistic VRP variants within **approximately 30 minutes**, the performance achieved by AFL is already competitive and encouraging. More importantly, our framework supports a wider range of realistic and diverse scenarios, such as the electric vehicle routing variants (Table 4 in the paper), which **traditional solvers like HGS-PyVRP cannot handle**.
> > > > >
> > > > > We appreciate your positive score and the increased confidence in our work. Your comments also highlight valuable directions for future exploration, such as integrating LLMs with advanced heuristics like HGS. We agree that this is an ambitious and promising direction, and likely substantial enough to warrant a dedicated and standalone investigation. Thank you again for your constructive feedback and support.

---

### Official Review · Reviewer_QnWc · 2025-11-01

**Soundness:** 3
**Presentation:** 3
**Contribution:** 3
**Rating:** 4
**Confidence:** 3

**Summary:**

This paper proposes an LLM-based agent framework (AFL) for end-to-end solving of the Vehicle Routing Problem (VRP). The framework proceeds from the original VRPLIB instance to executable solver code and feasible solutions, without requiring manually written modules or external solvers. The entire workflow is decomposed into three subtasks: problem description, code generation, and solution derivation, executed by four dedicated agents: a Generating Agent (GA), a Judging Agent (JA), a Revising Agent (RA), and an Error Analysis Agent (EAA). These agents iteratively generate, inspect, and repair code under instance-specific constraints. At the core of the system lies a unified destroy (insert improvement heuristic), combined with simulated annealing for solution acceptance and rigorous constraint verification (capacity, time windows, and energy).

Experimental results show that AFL achieves 0% runtime error rate (RER) and 100% success rate (SR) across 17 VRP variants, and produces objective values competitive with traditional solvers and state-of-the-art LLM baselines such as SGE and DRoC.

**Strengths:**

(1) The paper cleanly explains the three subtasks and the four roles (GA/JA/RA/EAA) and how they interact (including buffer reuse and EAA-driven debugging loops). In addition, the overview figure and tables mapping constraints to VRPLIB fields aid comprehension.

(2) The proposed multi-agent, verification-centric code generation framework substantially reduces execution failures in LLM-based systems. It demonstrates strong reliability and scalability across a wide range of VRP variants, suggesting that the approach could serve as a promising blueprint for extending agentic LLM frameworks to other combinatorial optimization domains.

**Weaknesses:**

(1) Although the paper reports objective gaps (e.g., compared to the best-known or HGS reference algorithms), a more detailed runtime analysis, including per-stage timing breakdowns and hardware/memory configurations, would clarify the computational footprint and reproducibility of AFL.

(2) AFL relies on standardized VRPLIB input and Euclidean coordinates. Its robustness to schema variations (e.g., non-VRPLIB formats), data noise or missing fields, and non-Euclidean or network-based distance metrics remains unclear.

(3) The experimental comparisons primarily focus on SGE and DRoC (and reference classical solvers via gap reports). Given AFL’s agentic architecture, it would be beneficial to include an additional baseline that leverages tool-augmented debugging or unit-test generation within a single-agent setup, in order to isolate the specific impact of the multi-agent design.

**Questions:**

Please See Weaknesses

---

> ### Author Response · Authors · 2025-11-22
>
> We greatly appreciate your positive assessment of the clarity of our framework and the effectiveness of its multi-agent, verification-driven design. Your constructive suggestions are highly valuable and have provided us with insightful guidance for improving the presentation of our work. Below, we provide detailed responses to your concerns and suggestions.
>
> #### **W1:   Detailed Runtime Analysis and Configurations.**
> **Detailed Runtime Analysis.** Thank you for your thoughtful suggestion. We first provide the runtime breakdown during **testing phrase** in Table 1.1. These results, as noted in Lines 382–383, measure the combined runtime for Problem Description and Solution Derivation over 1,000 test instances using the already generated codes.
>
> ##### Table 1.1. Runtime Analysis of Testing Phrase on Different VRP Variants.
>
> | Time (m)             | CVRP | CVRPL | CVRPTW | OCVRP | CVRPLTW | OCVRPL | OCVRPTW | OCVRPLTW |
> | ------------------- | ---- | ---- | ----- | ---- | ------ | ----- | ------ | ------- |
> | Problem Description | 0.10 | 0.12 | 0.11  | 0.10 | 0.15   | 0.17  | 0.15   | 0.20    |
> | Solution Derivation | 2.00 | 6.81 | 9.21  | 2.10 | 8.75   | 5.58  | 11.00  | 17.07   |
>
> We addtionally provide a detailed runtime breakdown of **solver generation phrase** for the three subtasks: Problem Description, Code Generation, and Solution Derivation, across eight representative VRP variants (see Table 1.2 below). These results reflect the time required to generate executable solver code and obtain a feasible solution for a single instance, where the generated code can be reused within the same VRP variant. As shown in Table 1.2, **the code can be generated within approximately 30 minutes, which is notably shorter than the time typically required for human expert design**, highlighting the model’s strong efficiency in producing high-quality, executable, and feasible solver code within a short time.
>
> ##### Table 1.2. Runtime Analysis of Solver Generation Phrase on Different VRP Variants.
>
> | Time (m)             | CVRP  | CVRPL  | CVRPTW | OCVRP  | CVRPLTW | OCVRPL | OCVRPTW | OCVRPLTW |
> | ------------------- | ----- | ----- | ----- | ----- | ------ | ----- | ------ | ------- |
> | Problem Description | 0.10  | 0.13  | 0.12  | 0.10  | 0.16   | 0.16  | 0.15   | 0.20    |
> | Code Generation     | 12.28 | 12.88 | 18.38 | 23.94 | 27.43  | 28.20 | 24.85  | 30.50   |
> | Solution Derivation | 0.003 | 1.05  | 0.009 | 0.005 | 1.10   | 1.21  | 0.010  | 1.42    |
>
> **Configurations and Reproducibility.** All experiments were conducted using the OpenAI API on a server with an AMD EPYC 7702P CPU and 64 GB RAM, without GPU acceleration. We guarantee that all code and data will be publicly released after publication to ensure transparency and reproducibility.
>
> We have incorporated all these results in the revised paper (i.e., Table 10 and Section C.4 in Appendix) to improve clarity and completeness.

---

> ### Author Response · Authors · 2025-11-22
>
> #### **W2. VRPLIB Input and Euclidean Coordinates**
>
> We appreciate for this valuable comment. We agree that evaluating AFL’s robustness to different input schemas, data noise and distance metrics is important. To address this, we have conducted additional experiments.
>
> **Different Input Format.** As VRPLIB is the standard format in the VRP domain, we further evaluated AFL on **JSON and CSV inputs** to assess its robustness to alternative data representations commonly used in industrial pipelines and benchmark integrations. As shown in Tables 2 and 3, the model consistently maintains strong performance and solution feasibility across different input formats. All of these results are supplied in the paper Section C.11 and Table 17 in Appendix in the revised paper for futher donmentrate the robustness of AFL.
>
> ##### Table 2. Results of AFL Testing on JSON Format Input.
>
> |                              | CVRP (obj.)       | Gap (%)     | Time (m)     | CVRPL (obj.)      | Gap (%)     | Time (m)     | CVRPTW (obj.)      | Gap (%)     | Time (m)     | OCVRP (obj.)        | Gap (%)     | Time (m)     |
> | ---------------------------- | ---------------- | ---------- | ----------- | --------------- | ---------- | ----------- | ---------------- | ---------- | ----------- | ----------------- | ---------- | ----------- |
> | 50 Node Instance (*T*=10000)  | 10.68            | 2.99       | 2.69        | 10.86           | 2.55       | 7.37        | 16.26            | 1.43       | 12.97       | 6.71              | 3.07       | 2.07        |
> | 100 Node Instance (*T*=10000) | 16.36            | 3.71       | 4.99        | 16.42           | 4.12       | 29.20       | 26.17            | 2.95       | 21.22       | 10.22             | 5.04       | 5.38        |
> |                              | **CVRPLTW (obj.)** | **Gap (%)** | **Time (m)** | **OCVRPL (obj.)** | **Gap (%)** | **Time (m)** | **OCVRPTW (obj.)** | **Gap (%)** | **Time (m)** | **OCVRPLTW (obj.)** | **Gap (%)** | **Time (m)** |
> | 50 Node Instance (*T*=10000)  | 16.63            | 1.65       | 8.02        | 6.73            | 3.38       | 6.78        | 10.58            | 0.67       | 14.73       | 10.55             | 0.38       | 19.62       |
> | 100 Node Instance (*T*=10000) | 26.52            | 2.95       | 31.64       | 10.26           | 5.56       | 16.46       | 17.19            | 1.54       | 55.47       | 17.15             | 1.30       | 82.31       |
>
> ##### Table 3. Results of AFL Testing on CSV Format Input.
>
> |                              | CVRP (obj.)       | Gap (%)     | Time (m)     | CVRPL (obj.)      | Gap (%)     | Time (m)     | CVRPTW (obj.)      | Gap (%)     | Time (m)     | OCVRP (obj.)        | Gap (%)     | Time (m)     |
> | ---------------------------- | ---------------- | ---------- | ----------- | --------------- | ---------- | ----------- | ---------------- | ---------- | ----------- | ----------------- | ---------- | ----------- |
> | 50 Node Instance (*T*=10000)  | 10.69            | 3.09       | 2.61        | 10.75           | 1.51       | 4.73        | 16.19            | 1.00       | 14.87       | 6.68              | 2.61       | 2.80        |
> | 100 Node Instance (*T*=10000) | 16.40            | 4.99       | 5.50        | 16.31           | 3.42       | 20.72       | 26.07            | 2.56       | 26.11       | 10.14             | 4.21       | 6.17        |
> |                              | **CVRPLTW (obj.)** | **Gap (%)** | **Time (m)** | **OCVRPL (obj.)** | **Gap (%)** | **Time (m)** | **OCVRPTW (obj.)** | **Gap (%)** | **Time (m)** | **OCVRPLTW (obj.)** | **Gap (%)** | **Time (m)** |
> | 50 Node Instance (*T*=10000)  | 16.61            | 1.52       | 8.99        | 6.71            | 3.07       | 7.96        | 10.60            | 0.86       | 10.26       | 10.56             | 0.48       | 16.65       |
> | 100 Node Instance (*T*=10000) | 26.48            | 2.80       | 32.10       | 10.11           | 4.01       | 18.24       | 17.25            | 1.89       | 50.99       | 17.16             | 1.36       | 69.19       |
>
> **Data Noise and Missing Field.** Based on your valuable suggestion, we further explored how to handle data noise and missing fields. We fully agree that this is an important aspect of improving AFL’s robustness. As an initial attempt, we guided the LLM to **automatically generate code for detecting and correcting noisy or incomplete input data**. The preliminary results, summarized in Table 4, demonstrate that the approach is promising, though further refinement will be part of our future work.

---

> ### Author Response · Authors · 2025-11-22
>
> ##### Table 4. Detection rates of data noise and missing fields.
>
> *Note:"50-Node Instance (LLM)" refers to using the LLM directly to detect faults in the 50-node instances, while ‘50-Node Instance (Ours)’ refers to our method, where the LLM is guided to generate code that automatically analyzes the instance and produces a detailed fault report, which then directs the LLM’s final fault-detection process.*
>
> | Detection Rate (%)     | CVRP      | CVRPL    | CVRPTW     | OCVRP      | CVRPLTW    | OCVRPL     | OCVRPTW    | OCVRPLTW   |
> | --------------------- | --------- | ------- | --------- | --------- | --------- | --------- | --------- | --------- |
> | 50 Node Instance (LLM) | 87.25     | 69.50   | 66.50     | 91.88     | 59.61     | 65.25     | 69.50     | 56.59     |
> | 50 Node Instance (Our) | **99.49** | **100** | **95.00** | **98.98** | **99.51** | **99.50** | **98.00** | **96.10** |
>
> **Non-Euclidean Distance Metrics.** We thank you for highlighting this important point. We acknowledge the importance of evaluating robustness under non-Euclidean distance metrics. As discussed in Section 4.6 (Broad Applicability) and further detailed in Tables 13–15 in the Appendix, we evaluate AFL on **ATSP, ACVRP, and SOP**, all of which involve **non-Euclidean and asymmetric distance structures.** These experiments confirm that AFL performs reliably under such conditions. We will further clarified this aspect in the main text to make the distinction more explicit.
> #### **W3. Additional Baselines**
>
> We thank you for this insightful suggestion. To better isolate the impact of AFL’s multi-agent design, we have added several single-agent, tool-augmented baselines:
>
> - **Self-Verification** – a single-agent variant in which the LLM repeatedly verifies and repairs its own code based on code generation requirements.
> - **Self-Debug** – a single-agent variant where the LLM relies on an integrated debugging module to detect and correct errors.
> - **SGE + Debug** – an single-agent enhanced version of SGE in which we incorporate an additional debugging mechanism to improve code correctness.
>
> The results are summarized in Table 5, where "-" denotes cases in which the generated code failed to execute or produced infeasible solutions. We observe that AFL consistently outperforms all three baselines:
> * Self-Verification achieves better performance through iterative self-checking of code generation requirements, but its execution reliability remains limited.
> * Self-Debug provides better code executability and solution feasibility than Self-Verification, yet still falls short of AFL’s overall robustness and performance.
> * Although adding a debug module, SGE still struggles with constraint awareness due to its original design, resulting in very poor solution feasibility.
>
> These results **reinforce the importance of AFL’s multi-agent architecture**, particularly the coordination between JA, RA, and EAA, in achieving high executability, feasibility, and overall solution quality.
> ##### Table 5. Additional Single-Agent Baselines
> *Note: "-" indicates cases where the code generated by the prompt strategy is not executable or the resulting routes are infeasible.*
> | Gap (%)                                | CVRP     | CVRPL      | CVRPTW    | OCVRP     | CVRPLTW   | OCVRPL    | OCVRPTW   | OCVRPLTW  |
> | ------------------------------------- | -------- | --------- | -------- | -------- | -------- | -------- | -------- | -------- |
> | Self-Verification (50 Node Instance)  | 7.61     | -         | 6.80     | 5.71     | -        | -        | 3.54     | -        |
> | Self-Debug (50 Node Instance)         | 11.76    | -         | 5.11     | 10.60    | 4.58     | 14.59    | 43.39    | 12.57    |
> | SGE + Debug (50 Node Instance)         | -        | -         | -        | -        | -        | -        | -        | -        |
> | AFL (50 Node Instance)                 | **5.01** | **7.18**  | **2.87** | **4.30** | **4.34** | **4.61** | **1.24** | **2.12** |
> | Self-Verification (100 Node Instance) | 9.47     | -         | 8.64     | 9.25     | -        | -        | 5.96     | -        |
> | Self-Debug (100 Node Instance)   | 12.42    | -         | 8.14     | 13.77    | 7.38     | 23.56    | 51.39    | 14.37    |
> | SGE + Debug (100 Node Instance)        | -        | -         | -        | -        | -        | -        | -        | -        |
> | AFL (100 Node Instance)    | **6.66** | **10.08** | **4.64** | **6.58** | **7.14** | **6.68** | **2.54** | **4.02** |
>
> We sincerely appreciate the reviewer’s great suggestion, and we have incorporated all corresponding results into the revised paper, including Sections 4.4 and Tables 7.
>
> ----
> We sincerely appreciate the reviewer’s thoughtful evaluation and constructive feedback. If there are still aspects of our explanation that appear ambiguous or incomplete, we would be glad to clarify them in more detail. We welcome any further questions or suggestions and are fully committed to improving the manuscript wherever needed.

---

### Author Response · Authors · 2025-11-22
**General Response**

We thank all reviewers for their thoughtful evaluations and the encouraging recognition of AFL’s contributions. Across the reviews, the strengths highlighted include:
* AFL’s full end-to-end automation from raw VRP instances to feasible solutions (Reviewer `9utD`, `KZcf`, `KgjX`);
* Clear decomposition of the pipeline into three subtasks and the effective coordination of specialized agents, including verification and error-analysis mechanisms (Reviewer `QnWc`, `KZcf`);
* Substantial improvement in reliability brought by the verification-centric multi-agent design, which reduces execution failures in LLM-based systems (Reviewer `QnWc`, `KZcf`);
* AFL’s broad applicability demonstrated across diverse VRP variants, including complex and real-world formulations such as Electric VRPs, ACVRP, ATSP, and SOP (Reviewer `9utD`, `KZcf`).
* Ambition and potential impact of developing a generalizable and self-contained LLM framework for combinatorial optimization (Reviewer `KgjX`).

The reviewers’ comments and questions are highly constructive and have greatly helped us strengthen the paper. Motivated by these suggestions, we expanded our experiments and analyses, which led to several new insights. Below, we provide a global summary of these findings together with the corresponding experimental results.

* AFL runs efficiently, and its solver code can be generated in about 30 minutes, much faster than human design.
    * Table 1.1-1.3 Runtime Analysis of Testing Phrase, Solver Generation and single instance on Different VRP Variants
* AFL can accept multiple input formats, such as JSON and CSV, not only VRPLib; and robust to data noise.
    * Table 2-3. Results of AFL Testing on JSON and CSV Format Input.
    * Table 4. Detection rates of data noise and missing fields.
* LLM model capacity affects solution quality, and AFL can boost the LLM’s solution quality; code reliability under the AFL framework is insensitive to the choice of LLM.
    * Table 6-7. AFL Performance on Claude-Sonnet-4-20250514 and GPT-4o.
    * Table 8. Gap Comparison across Different Prompting Strategies.
* AFL multi-agetic framework can surpass different prompt strategies and LLM-based methods, also enhance *evolving-basic-heuristics* based LLM model
    * Table 8. Gap Comparison across Different Prompting Strategies.
    * Table 11. Gap comparison on benchmark instances
    * Table 12. Comparison results on additional 40 standard 100-node benchmark instances.
    * Table 14. AFL Application on ReEvo

---

### Meta-Review · Area_Chair_4F29 · 2026-01-06

**Summary:**

This paper introduces AFL, an agentic framework that achieves end-to-end automation for complex Vehicle Routing Problems (VRPs) by decomposing the workflow into subtasks and coordinating specialized agents. While the framework shows high reliability and successfully generates solutions for diverse VRP variants without external solvers, initial reviews raised several concerns. Reviewers primarily questioned the incremental nature of the methodological novelty, the gap in solution quality when compared to specialized state-of-the-art algorithms, and the lack of a comprehensive runtime and computational analysis.

**Reviewer Concerns:**

The rebuttal phase successfully resolved the primary technical concerns of Reviewers QnWc, 9utD, and KZcf, particularly regarding the depth of the runtime analysis and the framework's generalization capabilities across various problem benchmarks. While some discussion remains regarding the general prevalence of multi-agent architectures in the LLM field and the performance gap relative to specialized solvers, the reviewers largely agreed that the authors' detailed clarifications mitigated these issues.

**Reviewer Scores:**

Reviewer QnWc would likely increase the score to a positive level, reflecting the resolution of their initial technical doubts. Reviewer 9utD would likely maintain the positive score and increase the confidence level, while Reviewer KZcf would likely maintain their positive rating. Given that the authors have successfully addressed the core criticisms and demonstrated the practical utility of the framework, the Area Chair AC recommends the acceptance of the manuscript.

---

### Decision · Program_Chairs · 2026-01-26

Accept (Poster)